

# 1 FLUXNET-CH4: A global, multi-ecosystem dataset and analysis
# 2 of methane seasonality from freshwater wetlands

Kyle B. Delwiche[1], Sara Helen Knox[2], Avni Malhotra[1], Etienne Fluet-Chouinard[1], Gavin
McNicol[1], Sarah Feron[1,3], Zutao Ouyang[1], Dario Papale[4,5], Carlo Trotta[5], Eleonora Canfora[5],
You-Wei Cheah[6], Danielle Christianson[6], M. Carmelita R. Alberto[7], Pavel Alekseychik[8], Mika
Aurela[9], Dennis Baldocchi[10], Sheel Bansal[11], David P. Billesbach[12], Gil Bohrer[13]
Rosvel Bracho[14], Nina Buchmann[15], David I. Campbell[16], Gerardo Celis[17], Jiquan Chen[18],
Weinan Chen[19], Housen Chu[20], Higo J. Dalmagro[21], Sigrid Dengel[6], Ankur R. Desai[22], Matteo
Detto[23], Han Dolman[24], Elke Eichelmann[25], Eugenie Euskirchen[26], Daniela Famulari[27], Thomas
Friborg[28], Kathrin Fuchs[29], Mathias Goeckede[30], Sébastien Gogo[31], Mangaliso J. Gondwe[32],
Jordan P. Goodrich[16], Pia Gottschalk[33], Scott L. Graham[34], Martin Heimann[30], Manuel
Helbig[35,36], Carole Helfter[37], Kyle S. Hemes[10,38], Takashi Hirano[39], David Hollinger[40], Lukas
Hörtnagl[15], Hiroki Iwata[41], Adrien Jacotot[31], Joachim Jansen[42], Gerald Jurasinski[43], Minseok
Kang[44], Kuno Kasak[45], John King[46], Janina Klatt[47], Franziska Koebsch[43], Ken W. Krauss[48],
Derrick Y.F. Lai[49], Ivan Mammarella[50], Giovanni Manca[51], Luca Belelli Marchesini[52], Jaclyn
Hatala Matthes[53], Trofim Maximon[54], Lutz Merbold[55], Bhaskar Mitra[56], Timothy H. Morin[57],
Eiko Nemitz[37], Mats B. Nilsson[58], Shuli Niu[19], Walter C. Oechel[59], Patricia Y. Oikawa[60],
Keisuke Ono[61], Matthias Peichl[58], Olli Peltola[9], Michele L. Reba[62], Andrew D. Richardson[63,64],
William Riley[6], Benjamin R. K. Runkle[65], Youngryel Ryu[66], Torsten Sachs[33], Ayaka Sakabe[67],
Camilo Rey Sanchez[10], Edward A. Schuur[68], Karina V.R. Schäfer[69], Oliver Sonnentag[70], Jed P.
Sparks[71], Ellen Stuart-Haëntjens[72], Cove Sturtevant[73], Ryan C. Sullivan[74], Daphne J. Szutu[10],
Jonathan E. Thom[75], Margaret S. Torn[6], Eeva-Stiina Tuittila[76], Jessica Turner[77], Masahito
Ueyama[78], Alex C. Valach[10], Rodrigo Vargas[79], Andrej Varlagin[80], Alma Vazquez-Lule[79],
Joseph G. Verfaillie[10], Timo Vesala[50], George L. Vourlitis[81], Eric J. Ward[48], Christian Wille[33],
Georg Wohlfahrt[82], Guan Xhuan Wong[83], Zhen Zhang[84], Donatella Zona[59,85], Lisamarie
Windham-Myers[86], Benjamin Poulter[87], Robert B. Jackson[1,38,88]
[1] Department of Earth System Science, Stanford University, Stanford, California
[2] Department of Geography, The University of British Columbia, Vancouver, British Columbia, Canada
[3] Department of Physics, University of Santiago de Chile, Santiago, Chile
[4] Dipartimento per la Innovazione nei Sistemi Biologici, Agroalimentari e Forestali, Università degli Studi della
Tuscia, Largo dell'Universita, Viterbo, Italy e Forestali, Universita
[5] euroMediterranean Center on Climate Change CMCC, Lecce, Italy
[6] Earth and Environmental Sciences Area, Lawrence Berkeley National Lab, Berkeley, California
[7] International Rice Research Institute
[8] Natural Resources Institute Finland (LUKE), Helsinki, Finland
[9] Finnish Meteorological Institute, PO Box 501, 00101 Helsinki, Finland
[10] Department of Environmental Science, Policy and Management, University of California, Berkeley, CA, USA
[11] U.S. Geological Survey, Northern Prairie Wildlife Research Center, 8711 37th St Southeast, Jamestown, ND
58401 USA
[12] University of Nebraska-Lincoln, Department of Biological Systems Engineering, Lincoln, NE 68583, USA
[13] Department of Civil, Environmental & Geodetic Engineering, Ohio State University
[14] School of Forest Resources and Conservation, University of Florida, Gainesville FL, 32611
[15] Department of Environmental Systems Science, Institute of Agricultural Sciences, ETH Zurich, 8092 Zurich,
Switzerland
[16] School of Science, University of Waikato, Hamilton, New Zealand





[17] Agronomy Department, University of Florida, Gainesville FL, 32601
[18] Department of Geography, Michigan State University
[19] Institute of Geographic Sciences and Natural Resources Research, Chinese Academy of Sciences, Beijing 100101,
PR China.
[20] Climate and Ecosystem Sciences Division, Lawrence Berkeley National Lab, Berkeley, CA 94702, USA
[21] Universidade de Cuiaba, Cuiaba, Mato Grosso, Brazil
[22] Dept of Atmospheric and Oceanic Sciences, University of Wisconsin-Madison, Madison, WI 53706 USA
[23] Department of Ecology and Evolutionary Biology, Princeton University, Princeton NJ, USA
[24] Department of Earth Sciences, Vrije Universiteit, Amsterdam, Netherlands
[25] School of Biology and Environmental Science, University College Dublin, Ireland
[26] University of Alaska Fairbanks, Institute of Arctic Biology, Fairbanks, AK, USA
[27] C NR - institute for Mediterranean Agricultural and Forest Systems, Piazzale Enrico Fermi, 1  Portici (Napoli)
Italy
[28] University of Copenhagen, Department of Geosciences and Natural Resource Management
[29] Institute of Meteorology and Climate Research - Atmos. Environ. al Research, Karlsruhe Institute of Technology
(KIT Campus Alpin), 82467 Garmisch-Partenkirchen, Germany
[30] Max Planck Institute for Biogeochemistry, Jena, Germany
[31] ISTO, Université d'Orléans, CNRS, BRGM, UMR 7327, 45071, Orléans, France
[32] Okavango Research Institute, University of Botswana, Maun, Botswana.
[33] GFZ German Research Centre for Geosciences, Telegrafenberg, 14473 Potsdam, Germany
[34] Manaaki Whenua - Landcare Research, Lincoln, NZ
[35] Université de Montréal, Département de géographie, Université de Montréal, Montréal, QC H2V 0B3,
[36] Canada & Dalhousie University, Department of Physics and Atmospheric Science, Halifax, NS B2Y 1P3, Canada
[37] UK Centre for Ecology and Hydrology, Edinburgh, UK
[38] Woods Institute for the Environment, Stanford University, Stanford, California
[39] Research Faculty of Agriculture, Hokkaido University, Sapporo, Japan
[40] Northern Research Station, USDA Forest Service, Durham, NH 03824, USA
[41] Department of Environmental Science, Faculty of Science, Shinshu University
[42] Stockholm University, Department of Geological Sciences
[43] University of Rostock, Rostock, Germany
[44] National Center for Agro Meteorology, Seoul, South Korea
[45] Department of Geography, University of Tartu, Vanemuise st 46, Tartu, 51410, Estonia
[46] Department of Forestry and Environmental Resources, North Carolina State University, Raleigh, NC, USA
[47] Vegetation Ecology, Institute of Ecology and Landscape, Department Landscape Architecture, Weihenstephan-
Triesdorf University of Applied Sciences, Am Hofgarten 1, 85354 Freising, Germany
[48] USGS Wetland and Aquatic Research Center, Lafayette LA
[49] Department of Geography and Resource Management, The Chinese University of Hong Kong, Shatin, New
Territories, Hong Kong SAR, China
[50] Institute for Atmospheric and Earth System Research/Physics, Faculty of Science, University of Helsinki,
Helsinki, Finland
[51] European Commission, Joint Research Centre (JRC), Ispra, Italy.
[52] Dept. of Sustainable Agro-Ecosystems and Bioresources, Research and Innovation Centre, Fondazione Edmund
Mach, San Michele all'Adige , Italy
[53] Department of Biological Sciences, Wellesley College, Wellesley, MA 02481, USA
[54] Institute for Biological Problems of the Cryolithozone, RAS, Yakutsk, REp. Yakutia.
[55] Mazingira Centre, International Livestock Research Institute (ILRI), Old Naivasha Road, PO Box 30709, 00100
Nairobi, Kenya
[56] Northern Arizona University, School of Informatics, Computing and Cyber Systems
[57] Environmental Resources Engineering, SUNY College of Environmental Science and Forestry
[58] Dept. of Forest Ecology and Management, Swedish University of Agricultural Sciences, 901 83 Umeå, Sweden
[59] Dept. Biology, San Diego State University, San Diego, CA 92182, USA
[60] Department of Earth and Environmental Sciences, Cal State East Bay, Hayward CA 94542 USA
[61] National Agriculture and Food Research Organization, Tsukuba, Japan
[62] USDA-ARS Delta Water Management Research Unit, Jonesboro, Arkansas 72401, United States
[63] School of Informatics, Computing & Cyber Systems, Northern Arizona University, Flagstaff, AZ 86011, USA
[64] Center for Ecosystem Science and Society, Northern Arizona University, Flagstaff, AZ 86011, USA



[65] Department of Biological & Agricultural Engineering, University of Arkansas, Fayetteville, Arkansas 72701,
United States
[66] Department of Landscape Architecture and Rural Systems Engineering, Seoul National University, South Korea
[67] Hakubi center, Kyoto University, Kyoto, Japan
[68] Department of Biological Sciences, Northern Arizona University, Flagstaff, AZ, USA
[69] Dept of Earth and Environmental Science, Rutgers University Newark, NJ
[70] Université de Montréal, Département de géographie, Université de Montréal, Montréal, QC H2V 0B3, Canada
[71] Department of Ecology and Evolution, Cornell
[72] USGS California Water Science Center, 6000 J Street, Placer Hall, Sacramento, CA, 95819
[73] National Ecological Observatory Network, Battelle, 1685 38th St Ste 100, Boulder, Colorado, 80301, USA
[74] Environmental Science Division, Argonne National Laboratory, Lemont, IL, USA
[75] Space Sciences and Engineering Center, University of Wisconsin-Madison, Madison, WI 53706 USA
[76] School of Forest Sciences, University of Eastern Finland, Joesnuu, Finland
[77] Freshwater and Marine Science, University of Wisconsin-Madison
[78] Graduate School of Life and Environmental Sciences, Osaka Prefecture University
[79] Department of Plant and Soil Sciences, University of Delaware, Newark, DE, USA
[80] A.N. Severtsov Institute of Ecology and Evolution, Russian Academy of Sciences
[81] California State University San Marcos, San Marcos, CA, USA
[82] University of Innsbruck, Department of Ecology, Sternwartestr. 15, 6020 Innsbruck, AUSTRIA
[83] Sarawak Tropical Peat Research Institute, Sarawak, Malaysia
[84] Department of Geographical Sciences, University of Maryland, College Park, MD 20740, USA
[85] Department of Animal and Plant Sciences, University of Sheffield, Western Bank, Sheffield, S10 2TN, United
Kingdom
[86] USGS Water Mission Area, 345 Middlefield Road, Menlo Park, CA, 94025
[87] Biospheric Sciences Laboratory, NASA Goddard Space Flight Center, Greenbelt, Maryland
[88] Precourt Institute for Energy, Stanford University, Stanford, California

*Correspondence to*: Kyle B. Delwiche (delwiche@stanford.edu)

**Abstract.** Methane ($CH_4$) emissions from natural landscapes constitute roughly half of global $CH_4$ contributions to
the atmosphere, yet large uncertainties remain in the absolute magnitude and the seasonality of emission quantities
and drivers. Eddy covariance (EC) measurements of $CH_4$ flux are ideal for constraining ecosystem-scale $CH_4$
emissions, including their seasonality, due to quasi-continuous and high temporal resolution of flux measurements,
coincident measurements of carbon, water, and energy fluxes, lack of ecosystem disturbance, and increased
availability of datasets over the last decade. Here, we 1) describe the newly published dataset, FLUXNET-$CH_4$
Version 1.0, the first global dataset of $CH_4$ EC measurements (available at https://fluxnet.org/data/fluxnet-ch4-
community-product/). FLUXNET-$CH_4$ includes half-hourly and daily gap-filled and non gap-filled aggregated $CH_4$
fluxes and meteorological data from 79 sites globally: 42 freshwater wetlands, 6 brackish and saline wetlands, 7
formerly drained ecosystems, 7 rice paddy sites, 2 lakes, and 15 uplands. Then, we 2) evaluate FLUXNET-$CH_4$
representativeness for freshwater wetland coverage globally, because the majority of sites in FLUXNET-$CH_4$ Version
1.0 are freshwater wetlands and because freshwater wetlands are a substantial source of total atmospheric $CH_4$
emissions; and 3) provide the first global estimates of the seasonal variability and seasonality predictors of freshwater
wetland $CH_4$ fluxes. Our representativeness analysis suggests that the freshwater wetland sites in the dataset cover
global wetland bioclimatic attributes (encompassing energy, moisture, and vegetation-related parameters) in arctic,
boreal, and temperate regions, but only sparsely cover humid tropical regions. Seasonality metrics of wetland $CH_4$
emissions vary considerably across latitudinal bands. In freshwater wetlands (except those between 20° S to 20° N)
the spring onset of elevated $CH_4$ emissions starts three days earlier, and the $CH_4$ emission season lasts 4 days longer,
for each degree C increase in mean annual air temperature. On average, the onset of increasing $CH_4$ emissions lags
soil warming by one month, with very few sites experiencing increased $CH_4$ emissions prior to the onset of soil





warming. In contrast, roughly half of these sites experience the spring onset of rising CH$_4$ emissions prior to the spring
increase in gross primary productivity (GPP). The timing of peak summer CH$_4$ emissions does not correlate with the
timing for either peak summer temperature or peak GPP. Our results provide seasonality parameters for CH$_4$
modeling, and highlight seasonality metrics that cannot be predicted by temperature or GPP (i.e., seasonality of CH$_4$
peak). The FLUXNET-CH4 dataset provides an open-access resource for CH$_4$ flux synthesis, has a range of
applications, and is unique in that it includes coupled measurements of important CH$_4$ drivers such as GPP and
temperature. Although FLUXNET-CH4 could certainly be improved by adding more sites in tropical ecosystems and
by increasing the number of site-years at existing sites, it is a powerful new resource for diagnosing and understanding
the role of terrestrial ecosystems and climate drivers in the global CH$_4$ cycle. All seasonality parameters are available
at https://doi.org/10.5281/zenodo.4408468. Additionally, raw FLUXNET-CH4 data used to extract seasonality
parameters can be downloaded from https://fluxnet.org/data/fluxnet-ch4-community-product/, and a complete list of
the 79 individual site data DOIs is provided in Table 2 in the Data Availability section of this document.

## 1 Introduction

Methane (CH$_4$) has a global warming potential that is 28 times larger than carbon dioxide (CO$_2$) on a 100-
174 year time scale (Myhre et al., 2013), and its atmospheric concentration has increased by >1000 ppb since 1800
(Etheridge et al., 1998). While atmospheric CH$_4$ concentrations are substantially lower than those of CO$_2$, CH$_4$'s
higher effectiveness at absorbing longwave radiation means that CH$_4$ has contributed 20-25% as much radiative
forcing as CO$_2$ since 1750 (Etminan et al., 2016). Despite its importance to global climate change, natural CH$_4$ sources
and sinks remain poorly constrained, and with uncertain attribution to the various biogenic and anthropogenic sources
(Saunois et al., 2016, 2020). Bottom-up and top-down estimates differ by 154 TG/yr (745 vs 591 TG/yr, respectively),
with much of this difference arising from natural sources (Saunois et al., 2020). Vegetated wetlands and inland water
bodies account for most natural CH$_4$ emissions, as well as the majority of uncertainty in bottom-up emissions estimates
(Saunois et al., 2016). Better diagnosis and prediction of terrestrial CH$_4$ sources to the atmosphere requires high
frequency and continuous measurements of CH$_4$ exchanges across a continuum of ecological time (hours to years)
and space (meters to kilometers) scales.

Tower-based eddy covariance (EC) measurements provide ecosystem-scale CH$_4$ fluxes at high temporal
resolution across years, are coupled with measurements of key CH$_4$ drivers such as temperature, water and recent
substrate input (inferred from CO$_2$ flux), and thus help constrain bottom-up CH$_4$ budgets and improve CH$_4$ predictions.
Although EC towers began measuring CO$_2$ fluxes in the late 1970s (Desjardins 1974; Anderson et al., 1984), and
some towers began measuring CH$_4$ in the 1990s (Verma et al., 1992), most CH$_4$ flux EC measurements began within
the last decade. Given that many EC CH$_4$ sites are relatively new, the flux community has only recently compiled
them for global synthesis efforts (e.g., Chang et al., in review) and is still working to standardize CH$_4$ flux
measurements and establish gap-filling protocols (Nemitz et al., 2018; Knox et al., 2019). Furthermore, the growth of
EC networks for CH$_4$ fluxes has sometimes taken place in a relatively *ad hoc* fashion, often at sites that were already
measuring CO$_2$ fluxes or where higher CH$_4$ fluxes were expected, potentially introducing bias. The representativeness
and spatial distribution of CO$_2$ flux tower networks has been assessed to evaluate its ability to upscale fluxes regionally
(Hargrove et al., 2003; Hoffman et al., 2013; Papale et al., 2015; Villarreal et al., 2018, 2019) and globally (Jung et
al., 2009; 2020; Kumar et al., 2016). However, a relatively sparse coverage of CH$_4$ flux towers prompts the question
of how well the current observation network provides a sufficient sampling of global or ecosystem-specific bioclimatic
conditions.



Broad-scale wetland $CH_4$ seasonality estimates, such as when fluxes increase, peak, and decrease and the
predictors of seasonality, remain relatively unconstrained across wetlands globally. These key seasonality metrics
vary considerably across high-emitting systems such as wetlands and other aquatic systems (Desjardins, 1974; Dise,
1992; Melloh and Crill 1996; Wik et al., 2013; Zona et al., 2016; Treat et al., 2018). Few continuous $CH_4$ flux datasets
across representative site-years make it difficult to establish trends in seasonal dynamics, though monthly or annually
aggregated estimates of $CH_4$ fluxes from different seasons do exist for high latitudes (Zona et al., 2016; Treat et al.,
2018). Seasonal variability in wetland $CH_4$ fluxes is expected to be driven by changes in air and soil temperature, soil
moisture (including water table dynamics), and recent carbon substrate availability, which influence the rates of $CH_4$
production and consumption (Lai, 2009; Bridgham et al., 2013; Dean et al., 2018). Temperature has widely been found
to strongly affect $CH_4$ flux (Chu et al., 2014; Yvon-Durocher et al., 2014; Sturtevant et al., 2016), but the relationship
is complex (Chang et al., 2020) and varies seasonally (Koebsch et al., 2015; Helbig et al., 2017).  Methane flux is also
driven by inundation depth since anoxic conditions are typically necessary for methanogenesis (Lai, 2009; Bridgham
et al., 2013), though $CH_4$ production under bulk-oxic conditions has been observed (Angle et al., 2017). Substrate
availability influences $CH_4$ production potential and is linked with gross primary productivity (GPP) because recent
photosynthate fuels methanogenesis though this relationship can vary by ecosystem type, plant functional type and
biome (Megonigal et al., 1999; Chanton et al., 2008; Hatala et al., 2012; Lai et al., 2014; Malhotra and Roulet, 2015;
Sturtevant et al., 2016). In process models, the seasonality of $CH_4$ emissions from wetlands globally is primarily
constrained by inundation (Poulter et al., 2017), with secondary within-wetland influences from temperature and
availability of carbon (C) substrates (Melton et al., 2013; Castro-Morales et al., 2018). Bottom-up and top-down global
$CH_4$ estimates continue to disagree on total $CH_4$ flux magnitudes and seasonality, including the timing of annual peak
emissions (Spahni et al., 2011; Saunois et al., 2020). Thus, the variability and predictors of wetland $CH_4$ seasonality
globally remain a knowledge gap that high-frequency and long-term EC data can help fill.

Here, we 1) describe Version 1.0 of the FLUXNET-CH4 dataset (available at https://fluxnet.org/data/fluxnet-
ch4-community-product/). Version 1.0 of the dataset expands and formalizes the publication of data scattered among
regional flux networks as described previously in Knox et al., 2019. FLUXNET-CH4 includes half-hourly and daily
gap-filled and non gap-filled aggregated $CH_4$ fluxes and meteorological data from 79 sites globally: 42 freshwater
wetlands, 6 brackish and saline wetlands, 7 formerly drained ecosystems, 7 rice paddy sites, 2 lakes, and 15 upland
ecosystems. Since the majority of sites in FLUXNET-CH4 Version 1.0 are freshwater wetlands, and freshwater
wetlands are a substantial source of total atmospheric $CH_4$ emissions, we use the subset of data from freshwater
wetlands to then; 2) evaluate the representativeness of freshwater wetland coverage in the dataset relative to wetlands
globally; and 3) provide the first assessment of global variability and predictors of freshwater wetland $CH_4$ flux
seasonality. We quantify a suite of $CH_4$ seasonality metrics and evaluate temperature and GPP (a proxy for recent
substrate input) as predictors of seasonality across four latitudinal bands (northern, temperate, subtropical, and
tropical). Due to a lack of high-temporal resolution water table data at all sites, our analyses are unable to evaluate the
critical role of water table on $CH_4$ seasonality. Here we provide parameters for better understanding and modeling
seasonal variability in freshwater wetland $CH_4$ fluxes and generate new hypotheses and data resources for future
syntheses.

**2. Methods**

**2.1 FLUXNET-CH4 dataset**

**2.1.1  History and data description**

The FLUXNET-CH4 dataset was initiated by the Global Carbon Project (GCP) in 2017 to better constrain
the global $CH_4$ budget (https://www.globalcarbonproject.org/methanebudget/index.htm).  Beginning with a kick off
meeting in May 2018 in Washington DC, hosted by Stanford University, we coordinated with the AmeriFlux
Management Project, the European Ecosystem Fluxes Database, and the ICOS Ecosystem Thematic Centre (ICOS-





ETC) in order to avoid duplication of efforts, as most sites are part of different regional networks (albeit with different
data products). We have collected and standardized data for FLUXNET-CH4 with assistance from the regional flux
networks, AmeriFlux's "Year of Methane", FLUXNET, the EU's Readiness of ICOS for Necessities of Integrated
Global Observations (RINGO) project, and a USGS Powell Center working group. FLUXNET-CH4 is a community-
led project, so while we developed it with assistance from FLUXNET, we do not necessarily use standard FLUXNET
data variables, formats, or methods.

FLUXNET-CH4 includes gap-filled half-hourly $CH_4$ fluxes and meteorological variables. Gaps in
meteorological variables (TA - air temperature, SW_IN - incoming shortwave radiation, LW_IN - incoming longwave
radiation, VPD - vapor pressure deficient, PA - pressure, P - precipitation, WS - wind speed) were filled with the
ERA-Interim (ERA-I) reanalysis product (Vuichard and Papale, 2015). We used the REddyProc package (Wutzler et
al., 2018) to filter flux values with low friction velocity ($u_*$), based on relating nighttime $u_*$, to fill gaps in $CO_2$, latent
heat, and sensible heat fluxes, and to partition net $CO_2$ fluxes into gross primary production (GPP) and ecosystem
respiration (RECO) using both the daytime (Lasslop et al., 2010) and nighttime (Reichstein et al., 2005) approaches
in REddyProc. $CH_4$ flux data gaps were filled using artificial neural network (ANN) methods first described in Knox
et al. (2015) and in Knox et al. (2019), and summarized here in Sect. 2.1.2. Gap-filled data for gaps exceeding two
259 months are provided and flagged for quality. Please see Table B1 for variable description and units, as well as quality
flag information. For the seasonality analysis in this paper we excluded data from gaps exceeding two months, and
we encourage future users of FLUXNET-CH4 to critically evaluate gap-filled values from long data gaps before
including them in analyses (Dengel et al., 2013; Kim et al., 2020).

In addition to half-hourly data, the FLUXNET-CH4 Version 1.0 release also contains a full set of daily mean
values for all parameters except wind direction and precipitation. Daily precipitation is included as the daily sum of
the half-hourly data, and daily average wind direction is not included.

### 2.1.2 Gap-filling methods and uncertainty estimates

As described in Knox et al. (2015) and in Knox et al. (2019), the ANN routine used to gap-fill the $CH_4$ data
was optimized for generalizability and representativeness. To avoid biasing the ANN toward environmental conditions
with typically better data coverage (e.g., summer-time and daytime measurements), the explanatory data were divided
into a maximum of 15 clusters using a k-means clustering algorithm. Data used to train, test, and validate the ANN
were proportionally sampled from these clusters. For generalizability, the simplest ANN architecture with good
performance (<5% gain in model accuracy for additional increases in architecture complexity) was selected for 20
extractions of the training, test, and validation data. Within each extraction, each tested ANN architecture was
reinitialized 10 times, and the initialization with the lowest root-mean-square-error was selected to avoid local minima.
The median of the 20 predictions was used to fill each gap. A standard set of variables available across all sites were
used to gap-fill $CH_4$ fluxes (Dengel et al., 2013), which included the previously mentioned meteorological variables
TA, SW_IN, WS, PA, and sine and cosine functions to represent seasonality. These meteorological variables were
selected since they are relevant to $CH_4$ exchange and were gap-filled using the ERA-I reanalysis data. Other variables
related to $CH_4$ flux (e.g., water table depth and soil temperature) were not included as explanatory variables as they
were not available across all sites or had large gaps that could not be filled using the ERA-I reanalysis data (Knox et
al., 2019). The ANN gap- filling was performed using MATLAB (MathWorks 2018, version 9.4.0).

While the median of the 20 predictions was used to fill each gap, the spread of the predictions was used to
provide a measure of uncertainty resulting from the ANN gap-filling procedure. Specifically, for gap-filled values,
the combined annual gap-filling and random uncertainty was calculated from the variance of the cumulative sums of
the 20 ANN predictions (Knox et al., 2015; Anderson et al., 2016; Oikawa et al., 2017). The (non-cumulative) variance
of the 20 ANN predictions was also used to provide gap-filling uncertainty for each half-hourly gap-filled value
included in the dataset. While this is useful for data-model comparisons, it cannot be used to estimate cumulative



annual gap-filling error because gap-filling error is not random, which is why the cumulative sums of the 20 ANN
predictions are used to estimate annual gap-filling error.

Random errors in EC fluxes follow a double exponential (Laplace) distribution with the standard deviation
varying with flux magnitude (Richardson et al., 2006; Richardson et al., 2012). For half-hourly $CH_4$ flux
measurements, random error was estimated using the residuals of the median ANN predictions, providing a
conservative "upper limit" estimate of the random flux uncertainty (Moffat et al., 2007; Richardson et al., 2008). The
annual cumulative uncertainty at 95% confidence was estimated by adding the cumulative gap-filling and random
measurement uncertainties in quadrature (Richardson and Hollinger, 2007; Anderson et al., 2016). Annual
uncertainties for individual site-years are provided in Table B7. Throughout this paper, we include uncertainties on
individual site years when discussing single years of data. In sites with multiple years of data, we report the standard
deviation of the multiple years.

### 299 2.1.3 Dataset structure and site metadata

To enable data use by the broader flux community, we have partnered with regional flux networks and
FLUXNET to provide standardized and gap-filled EC $CH_4$ data. FLUXNET-CH4 Version 1.0 contains two comma-
separated data files per site at half-hourly and daily resolutions. Half-hourly and daily aggregations are available for
download at https://fluxnet.org/data/fluxnet-ch4-community-product/, along with a file containing select site
metadata. Each site has a unique FLUXNET-CH4 DOI. All site data are available under CC BY 4.0
(https://creativecommons.org/licenses/by/4.0/) copyright license.

Metadata (Table B2) include site coordinates, ecosystem classification based on site literature,
presence/absence and dominance for specific vegetation types, and DOI link, as well as calculated data such as annual
and quarterly flux values. FLUXNET-CH4 Version 1.0 sites were classified based on site-specific literature as fen,
bog, swamp, marsh, salt marsh, lake, mangrove, rice paddy/field, wet tundra, upland, or drained ecosystems that
previously could have been wetlands, seasonally flooded pastures, or agricultural areas. To the extent possible, we
followed classification systems of previous wetland $CH_4$ syntheses (Olefeldt et al., 2013; Turetsky et al., 2014; Treat
et al., 2018). Drained systems are former wetlands that have subsequently been drained but may maintain a relatively
shallow water table. Upland ecosystems are further divided into alpine meadows, grasslands, needleleaf forests, mixed
forest, crops, tundra, and urban. Freshwater wetland classifications follow hydrological definitions of bog
(ombrotrophic), fen (minerotrophic), wet tundra, marshes and swamps, and were designated as per primary literature
on the site. For all sites, vegetation was classified for presence or absence of brown mosses, *Sphagnum* mosses,
ericaceous shrubs, trees (of any height) and aerenchymatous species (mostly Order Poales but includes exceptions).
These categories closely follow Treat et al., (2018), except that aerenchymatous species had to be expanded beyond
Cyperaceae to incorporate wetlands globally. Presence/absence of vegetation groups was designated based on species
lists in primary literature from the site. Out of the vegetation groups present, the dominant (most abundant) group is
also reported and is based on data from a survey of lead site investigators.

In addition to the variable description table (Table B1) and the site metadata (Table B2), we provide several
more tables to complement our analysis. Table B3 includes the climatic data for the representativeness analysis.
Seasonality parameters for $CH_4$ flux, air temperature, soil temperature (for sites with multiple probes, Table B4
includes parameters from the probe closest to the ground surface), and GPP are provided in Table B4, with the full set
of soil temperature parameters from all probes in Table B5. Table B6 contains the soil temperature probe depths.
Table B7 contains the annual flux and uncertainty. All Appendix B tables are also available at
https://doi.org/10.5281/zenodo.4408468.



### 2.1.4 Annual fluxes

Annual fluxes were calculated from gap-filled data for site-years with data gaps shorter than two consecutive months, or for sites above 20° N where >2 month data gaps occurred outside of the highest $CH_4$-emission months of May 1 through October 31. Since we did not sum gap-filled values for >2 month gaps during the winter, annual sums from these years will be an underestimate since winter fluxes can be important (Zona et al., 2016; Treat et al., 2018). Several sites had less than one year of data, and we report gap-filled flux annual sums for sites with between six months and one year of data (BW-Gum = 228 days, CH-Oe2 = 200 days, JP-Swl = 210 days, US-EDN = 182 days). While these sums will be an underestimate of annual flux since they do not span a full year (and we therefore do not use them in the seasonality analysis), their relative magnitude can still be informative. For example, site JP-SWL is a lake site, and even with less than a year of data the summed flux of 66 g C m$^{-2}$ is relatively high (Taoka et al., 2020). In addition to sites with short time series, the annual sum for site ID-Pag represents 365 days spanning June 2016 to June 2017.

### 2.1.5 Subset analysis on freshwater wetland $CH_4$ flux

In addition to the FLUXNET-CH4-wide description of site class distributions and annual fluxes, we also include a subset analysis on freshwater wetlands, given that it is the dominant ecosystem type in our dataset and an important global CH4 source (Saunois et al., 2016). First, we analyze freshwater wetland representativeness, and subsequently the seasonality of their $CH_4$ emissions. Freshwater wetlands included in the seasonality and representativeness analysis are indicated in Table B2, column "IN_SEASONALITY_ANALYSIS".

### 2.2 Wetland representativeness

### 2.2.1 Principal Component Analysis

To understand how the distribution of FLUXNET-CH4 Version 1.0 sites compares with the global wetland distribution, we evaluated the representativeness of the FLUXNET-CH4 Version 1.0 wetland sites in the entire global wetland cover along four bioclimatic gradients. Only freshwater wetland sites were included in this analysis, with coastal sites excluded because, due to a lack of global gridded datasets, salinity could not be included as an environmental variable despite being an important control on $CH_4$ production (Bartlett et al., 1987; Poffenbarger et al., 2011). The four bioclimatic variables used were: mean annual air temperature (MAT), latent heat flux (LE), enhanced vegetation index (EVI), and water index (SRWI; data sources in Table B3). EVI is a more direct measurement than GPP from global gridded products and is considered a reasonable proxy for GPP (Sims et al., 2006). Thus, we used EVI instead of GPP. Together, these environmental variables account for or are proxies for key controls of $CH_4$ production, oxidation at the surface, and transport (Bridgham et al., 2013). We use a principal components analysis (PCA) to visualize the site distribution across the four environmental drivers at once. For this analysis, we consider the annual average bioclimatic conditions over 2003-2015. In the PCA output, we evaluate the coverage of the 42 freshwater sites over 0.25° grid cells containing >5% wetland mean cover in Wetland Area and Dynamics for Methane Modeling (WAD2M; Zhang et al., In Review) for the same time period.

### 2.2.2 Global Dissimilarity and Constituency Analysis

To further identify geographical gaps in the coverage of the FLUXNET-CH4 Version 1.0 network, we quantified the dissimilarity of global wetlands from the tower network, using a similar approach to that taken for $CO_2$ flux towers (Kumar et al., 2016; Meyer and Pebesma 2020). We calculated the 4-dimensional Euclidean distance from the four bioclimatic variables between every point at the land surface to every tower location at the FLUXNET-CH4 network. We then divided these distances by the average distance between towers to produce a dissimilarity index. Dissimilarity scores <1 represent areas whose nearest tower is closer than the average distance among towers, while



areas with scores >1 are more distant. Lastly, we identified the importance of an individual tower in the network by
estimating the geographical area to which it is most analogous in bioclimate space. We divided the world's land
surface according to closest towers in bioclimatic space. The area to which each tower is nearest is defined as the
tower's constituency.

**2.3 Wetland CH$_4$ seasonality**

To examine freshwater wetland CH$_4$ seasonality across the global range of sites in FLUXNET-CH4
Version 1.0, we extracted seasonality parameters for CH$_4$, temperature, and GPP using Timesat, a software package
designed to analyse seasonality of environmental systems (Jönsson and Eklundh, 2002; Jönsson and Eklundh, 2004;
Eklundh and Jönsson, 2015). Timesat calculates a range of seasonality parameters, including baseline flux, peak
flux, and the slope of spring flux increase and fall decrease (Fig. 1). We also calculate parameters such as amplitude
( "e" - baseline, which is the average of "a" and "b", in Fig. 1), and relative peak timing ( ( "g" - "f" ) / ("h"  - "f") in
Fig. 1). Timesat uses a double-logistic fitting function to create a series of localized fits centered on data minima and
maxima. Localized fits are minimized using a merit function and the Levenberg-Marquardt method (Madsen et al.,
2004; Nielsen, 1999). These localized fits are then merged using a global function to create a smooth fit over the
full time interval., To fit CH$_4$ time-series in Timesat, we used gap-filled data after removing gaps exceeding two
387  months. We do not report Timesat parameters when large gaps occur during CH$_4$ emissions spring increase, peak, or
fall decrease.

We estimate 'start of elevated emissions season' when CH$_4$ emissions begin to increase in the spring ( "f"
in Fig. 1), and 'end of elevated emissions season' when the period of elevated flux ends in the fall ("h" in Fig. 1), as
the intercept between the Timesat fitted baseline parameter and shoulder-season slope (similar to Gu et al., 2009).
To extract seasonality parameters with Timesat, sites need a sufficiently pronounced seasonality, a sufficiently long
time period, and minimal data gaps (we note that while Timesat is capable of fitting two peaks per year, all the
freshwater wetland sites have a single annual peak). We excluded site-years in restored wetlands when wetlands
were still under construction. We were able to fit 36 freshwater wetland sites using Timesat, with 141 site-years of
data, using the double-logistic fitting method which followed site data well (representative examples in Fig. 2). For
extratropical sites in the Southern Hemisphere, we shifted all data by 182 days so that maximum solar insolation
seasonality would be congruent across the globe.

We also used Timesat to extract seasonality metrics for GPP, partitioned using the daytime-based approach
(Lasslop et al., 2010) (GPP_DT), air temperature (TA), and soil temperature (TS_1, TS_2, etc). For sites where
winter soil temperatures fall significantly below 0 °C, Timesat fits a soil temperature "start of elevated season" date
to periods when the soil is still frozen. In order for Timesat to define the soil temperature seasonality within the
thawed season, we converted all negative soil temperatures to zero (simply removing these values results in too
many missing values for Timesat to fit). Many sites have more than one soil temperature probe, so we extracted
separate seasonality metrics from each individual probe (although we used the metrics from the shallowest
temperature probe in our analysis). Tables B4 contain the Timesat seasonality parameters used in the seasonality
analysis. We did not include water table depth in the seasonality analysis because many sites either lack water table
depth measurements or have sparse data.



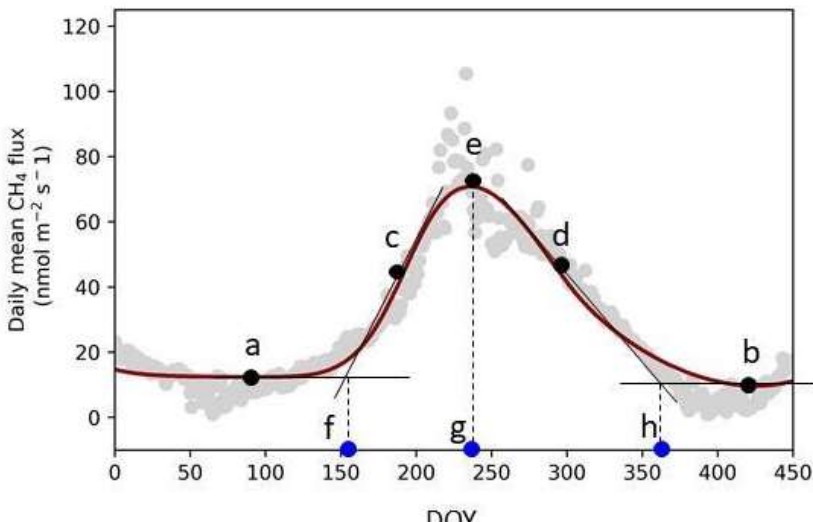

**Figure 1: TIMESAT parameter description. (a) and (b) base values (Timesat reports the average of these two values), (c)**
**and (d) slopes of seasonal curves (lines drawn between 20% and 80% of the amplitude), (e) peak value, and day of year**
**(DOY) for the start (f), peak (g), and end (h) of the elevated CH₄ emissions season. Data points are the mean daily gap-**
**filled CH₄ fluxes from site JP-Bby in 2015.**

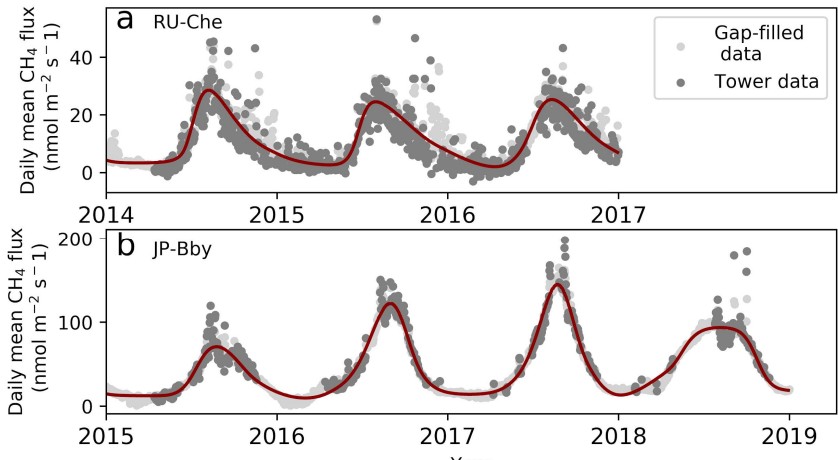

**Figure 2: Examples of Timesat fits for two FLUXNET-CH4 sites, (a) RU-Che and (b) JP-BBY. Flux data showing daily**
**average flux tower data, with several high outliers excluded to improve the plot (dark gray), gap-filled values (light gray),**



**and Timesat-fitted curve (dark red line) for sites JP-BBY and RU-Che. Timesat captures peak shape as well as size (note different scale on y-axes).**

We regressed the $CH_4$ seasonality parameters from Timesat against a range of temperature, moisture, and GPP (proxy for recent carbon input available as substrate) metrics using linear mixed-effect modeling with the *lmer* command (with site as a random effect) from the R (R Core Team 2018, version 3.6.2) package lmerTest. For these regressions we present the marginal $R^2$ outputs from *lmer*, which represent the variance explained only by the fixed effects. Mixed-effect modeling was necessary to account for the non-independence between measurements taken at the same site during different years (Zona et al., 2016; Treat et al., 2018). We also compared how seasonality metrics varied across latitudinal bands by dividing sites into northern ( > 60° N), temperate (between 40° N and 60° N), subtropical (absolute value between 20° and 40° latitude, with site NZ-KOP being the only Southern hemisphere site), and tropical (absolute value below 20°). Site-year totals for the northern, temperate, subtropical, and tropical bands were $n$ = 57, 36, 39, and 9, respectively. We used the Kruskal-Wallis test to establish whether groups (either across quarters or across latitudes) were from similar distributions, and the post hoc multiple comparison "Dwass, Steel, Critchlow, and Fligner" procedure for inter-group comparisons. Kruskal-Wallis and post-hoc tests were implemented in Python Version 3.7.4, using stats from scipy for Kruskal-Wallis and posthoc_dscf from scikit_posthocs.

In addition to comparing $CH_4$ flux seasonality across latitudinal bands and to the seasonality of potential drivers, we also compared quarterly flux sums by dividing data into quarterly periods: January/February/March (JFM), April/May/June (AMJ), July/August/September (JAS), and October/November/December (OND). For the sake of simplicity, we chose to compare quarterly periods rather than site-specific growing/non-growing season periods so that all time periods would be the same length. Quarterly sums were computed from the gap-filled $CH_4$ fluxes when the longest continuous data gap within the quarter did not exceed 30 days, leading to site-year counts of 67, 92, 95, 72 for JFM, AMJ, JAS, and OND, respectively. We compared quarterly fluxes across latitudinal bands both for the total flux, and for the quarterly percentage of the annual flux. Quarterly statistics were also conducted with the Kruskal-Wallis test and the post hoc multiple comparison "Dwass, Steel, Critchlow, and Fligner" procedure implemented in Python. Quarterly values are provided in Table B2, and the sum of mean quarterly flux does not always equal mean annual flux because some quarters either do not have data, or have data gaps that exceed 30 days.

**3. Results and Discussion**

**3.1 FLUXNET-CH4 dataset**

**3.1.1 Dataset description**

Version 1.0 of the FLUXNET-CH4 dataset contains 79 unique sites, 293 total site-years of data, and 201 site-years with sufficient data to estimate annual $CH_4$ emissions. A previous synthesis paper, published prior to the public data release of FLUXNET-CH4 Version 1.0, had 60 unique sites and 139 site-years with annual $CH_4$ emissions estimates (Knox et al., 2019). Freshwater wetlands make up the majority of sites (n = 42), and the dataset also includes five salt marshes and one mangrove wetland. Notable additions to FLUXNET-CH4 Version 1.0 from the previous unpublished dataset used in Knox et al., (2019) include six tropical sites (between 20°S and 20°N), including one site in South America, two sites in southern Africa, and three sites in Southeast Asia. The 15 upland sites include six needleleaf forests, three crop sites (excluding rice), two alpine meadows, one grassland, one mixed forest, one tundra, and one urban site. The drained sites represent former wetlands that have been artificially drained for use as grasslands





(*n* = 3) or croplands (*n* = 3).  FLUXNET-CH4 Version 1.0 sites span the globe, though are concentrated in North
America and Europe (Fig. 3).  Table B2 includes characteristics of all sites in the dataset.

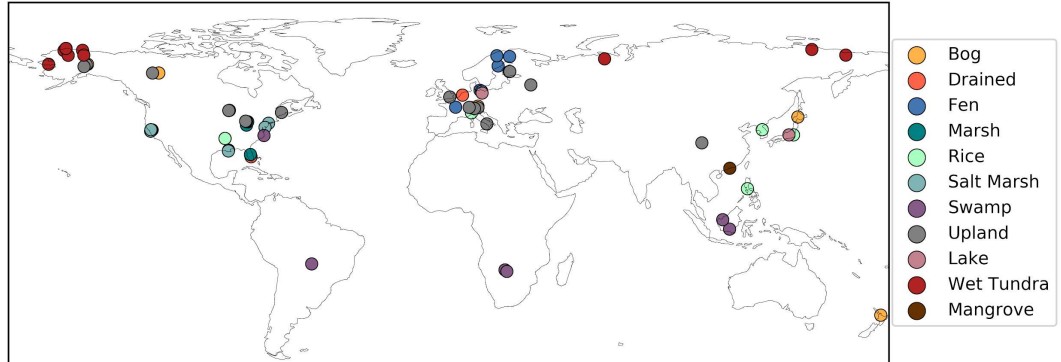

**Figure 3. Global map of FLUXNET-CH4 Version 1.0 site locations colored by site type. The bog and upland site in the Northwest Territories of Canada have been slightly offset from each other so that both are visible.**

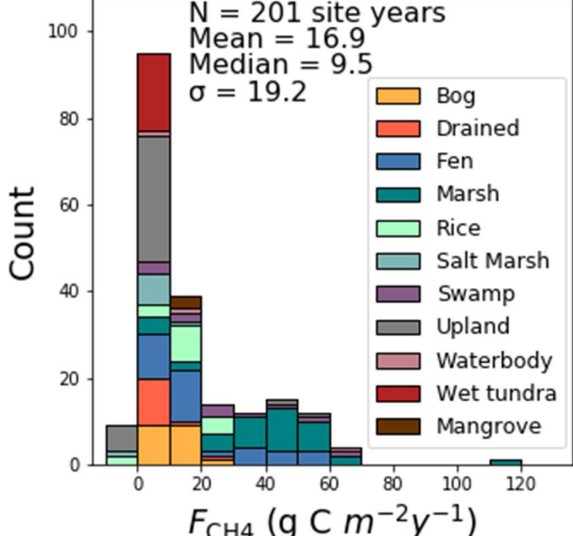

**Figure 4. Histogram of annual CH4 fluxes (g C m$^{-2}$ yr$^{-1}$) grouped by site type.**

Sites represent a range of ecosystem types, latitudes, median fluxes, and seasonality patterns (Table 1).
Across all FLUXNET-CH4 Version 1.0 sites, mean average annual flux is positively skewed with a median flux of
9.5 g C m$^{-2}$ yr$^{-1}$, a mean flux of 16.9 g C m$^{-2}$ yr$^{-1}$, and numerous annual fluxes exceeding 60 g C m$^{-2}$ yr$^{-1}$.  The
addition of 19 sites from the 60 sites aggregated in Knox et al., (2019) therefore do not significantly change the
distribution of annual CH$_4$ fluxes. Marshes and swamps have the highest median flux, and upland, salt marsh, and





tundra sites have the lowest (Fig. 4). Lake emissions are highly variable due to one high-flux lake site (JP-SWL).
Flux data at many sites show strong seasonality in CH$_4$ emissions, but data coverage is also lower outside the
growing season. Data coverage is lowest during the JFM quarter (on average 20% of half-hourly time periods
contain flux data) reflecting the predominance of Northern hemisphere sites and the practical difficulties in
maintaining EC tower sites during colder winter months (Table 1). Bogs, fens, and marshes have pronounced
seasonality, with fluxes being highest in the AMJ and JAS quarters. In contrast, CH$_4$ fluxes from uplands, drained
sites, and salt marshes are more uniform and low year-round.

**Table 1: Summary table of sites grouped by ecosystem class reporting annual mean flux (Ann_Flux) and standard**
**deviation from inter-annual variability (Ann_Flux_SD), site-years of data, % data cover per quarter, and median (med.)**
**flux across site class.**

| | # of Sites | # of Site-Years | Ann_Flux g C m$^{-2}$ year$^{-1}$ | Ann_Flux_SD g C m$^{-2}$ year$^{-1}$ | JFM coverage (%) | AMJ coverage (%) | JAS coverage (%) | OND coverage (%) | JFM flux (med.) | AMJ flux (med.) | JAS flux (med.) | OND flux (med.) |
|---|---|---|---|---|---|---|---|---|---|---|---|---|
| Salt marsh | 5 | 10 | 2.9 | 4.7 | 7 | 42 | 50 | 37 | 1.5 | 1.7 | 2.1 | 1.6 |
| Wet tundra | 11 | 39 | 3.8 | 1.8 | 8 | 28 | 40 | 18 | 0.4 | 2.6 | 8.1 | 3.2 |
| Upland | 15 | 47 | 4.0 | 10.5 | 23 | 35 | 39 | 28 | 1.2 | 0.5 | 1.4 | 0.8 |
| Drained | 7 | 20 | 6.3 | 7.1 | 22 | 39 | 39 | 29 | 4.6 | 3.6 | 5.1 | 3.6 |
| Bog | 7 | 32 | 10.5 | 6.4 | 8 | 27 | 37 | 18 | 7.2 | 11.0 | 24.8 | 9.5 |
| Mangrove | 1 | 3 | 11.1 | 0.5 | 46 | 28 | 30 | 41 | 3.2 | 7.2 | 22.5 | 14.1 |
| Rice | 7 | 20 | 14.4 | 8.8 | 16 | 37 | 45 | 27 | 3.2 | 11.9 | 43.1 | 4.2 |
| Fen | 8 | 40 | 20.5 | 16.0 | 29 | 43 | 40 | 30 | 2.8 | 14.2 | 26.0 | 6.4 |
| Swamp | 6 | 15 | 26.4 | 19.9 | 24 | 34 | 29 | 19 | 14.7 | 24.9 | 31.0 | 24.4 |
| Lake | 2 | 4 | 28.2 | 33.4 | 15 | 13 | 27 | 36 | 0.2 | 47.6 | 90.2 | 40.3 |
| Marsh | 10 | 42 | 40.8 | 20.7 | 22 | 43 | 53 | 30 | 13.5 | 55.0 | 85.8 | 36.1 |

### 3.1.2 Freshwater wetland CH$_4$ characteristics

The FLUXNET-CH4 Version 1.0 dataset contains 42 freshwater wetlands that span 37°S to 69°N, including
bogs, fens, wet tundra, marshes, and swamps, and a range of annual CH$_4$ emission rates (Fig. 4). The majority of
wetlands in our dataset emit 0-20 g C m$^{-2}$ yr$^{-1}$, with 10 emitting 20-60 g C m$^{-2}$ yr$^{-1}$, and one more than 60 g C m$^{-2}$ yr$^{-1}$.
Differences in annual flux among wetland types is partially driven by temperature (which is often linked to site type),
with mean annual air temperature explaining 51% of the variance between sites (Fig. 5, exponential relationship). The
global relationship between annual methane emissions and temperature can be described using a Q$_{10}$ relationship
where Q$_{10}$ = R2/R1$^{((T2-T1)/10)}$, with R2 and R1 being the methane emission rates at temperatures T2 and T1, respectively
(temperature in degrees C). The Q$_{10}$ based on Fig. 5 data is 2.57. Annual CH$_4$ flux is not correlated with mean annual
water table depth in FLUXNET-CH4 Version 1.0, unlike in Knox set al (2019; which used a subset of the FLUXNET-
CH4 Version 1.0 sites) where CH$_4$ flux was correlated with water table depth only for sites with water table below
ground for 90% of measured days  (r2 = 0.31, p<0.05, n = 27 site years, Knox et al., 2019).   Freshwater wetland
seasonality is further described in Sect. 3.3.

Figure 5: Relationship between mean annual wetland flux (g C m$^{-2}$ yr$^{-1}$) and mean annual air temperature (°C, logarithmic scale) for each freshwater wetland site, with wetland type indicated by symbol. The difference in emissions across site types is partially driven by difference in mean annual temperature. Markers represent individual site means, with vertical error bars representing the standard deviation of interannual variability.

### 3.1.3 Non-wetland CH$_4$ characteristics

Upland agricultural sites are characterized by a lack of seasonal pattern in CH$_4$ emissions, relatively low flux,
and some negative daily flux (i.e., CH$_4$ uptake) averages.  All of the upland non-agricultural sites in FLUXNET-CH4



Version 1.0 are net (albeit weak) CH$_4$ sources except for the needleleaf forest site US-Ho1, which has mean annual
flux of -0.1 ± 0.1 g C m$^{-2}$ yr$^{-1}$ (see Table B2 for site acronyms and metadata). The average agricultural site emissions
are 1.3 ± 0.8 g C m$^{-2}$ yr$^{-1}$ and non-agricultural site emissions are 1.6 ± 1.2 g C m$^{-2}$ yr$^{-1}$ across sites.

Rice sites (n = 7) have average annual emissions across all sites of 16.7 ± 7.7 g C m$^{-2}$ yr$^{-1}$ and are
characterized by strong seasonal patterns, with either one or more CH$_4$ emission peaks per year depending on the
number of rice seasons and field water management. One peak is typically observed during the reproductive period
for the continuously flooded sites with one rice season (i.e., US-HRC, JP-MSE) (Iwata et al., 2018; Runkle et al.,
2019; Hwang et al., 2020). For sites with only one rice season but with single or multiple drainage and re-flooding
periods, a secondary peak may appear before the reproductive peak (i.e., KR-CRK, IT-Cas, and US-HRA; Meijide et
al., 2011; Runkle et al., 2019; Hwang et al., 2020). Two reproductive peaks appear for sites with two rice seasons (i.e.
PH-RiF), and each reproductive peak may be accompanied by a secondary peak due to drainage events (Alberto et al.,
2015). Even sites with one, continuously flooded rice season may experience a second peak if the field is flooded
during the fallow season to provide habitat for migrating birds (e.g. US-Twt; Knox et al., 2016).

The dataset has one year of urban data from site UK-LBT in London, England. UK-LBT observes CH$_4$ fluxes
from a 190 m tall communications tower in the center of London, and had a mean annual flux of 46.5 ± 5.6 g C m-2
527  yr$^{-1}$. This flux is more than twice as high as the mean annual flux across all FLUXNET-CH4 Version 1.0 sites, 16.9
528  g C m$^{-2}$ yr$^{-1}$. The London site has higher CH$_4$ emissions in the winter compared to summer, which is attributed to a
seasonal increase in natural gas usage (Helfter et al., 2016.)

**3.1.4 Non-freshwater wetland CH$_4$ characteristics**

Three of the five saltwater wetlands in FLUXNET-CH4 Version 1.0 (US-Edn, US-MRM, and US-Srr) have
a very low mean annual flux (see Table B7 for individual site-year flux sums and associated uncertainty) and minimal
seasonality. Two other FLUXNET-CH4 Version 1.0 saltwater sites (US-La1, US-StJ) have significantly higher
fluxes, with annual sums of 12.6 ± 0.6 and 9.6 ± 1.0 g C m-2 yr$^{-1}$ respectively, while the mangrove site HK-MPM has
annual mean fluxes of 11.1 ± 0.5 g C m$^{-2}$ yr$^{-1}$. This range of CH$_4$ fluxes across different saltwater ecosystems could
be valuable for exploring the effect of salinity and different biogeochemical pathways of CH$_4$ production and transport
of CH$_4$ (Bartlett et al., 1987; Poffenbarger et al., 2011). Saltwater wetlands along the coast have unique CH$_4$ dynamics
attributable to the presence of abundant electron acceptors, most importantly sulphates, which inhibit methanogenesis
(Pattnaik et al., 2000; Mishra et al., 2003; Weston et al., 2006), but at low concentrations can have no effect (Chambers
et al., 2011) or even increase methanogenesis (Weston et al., 2011). In fact, estuarine wetlands with moderate salinity
can still be significant sources of CH$_4$ (Liu et al., 2020). Even under sulfate-rich conditions, high CH$_4$ production can
be found via methylotrophic methanogenesis (Seyfferth et al., 2020) or because the processes of sulfate reduction and
methanogenesis are spatially separated (Koebsch et al., 2019).

**3.2 Wetland Representativeness**

We evaluated the representativeness of freshwater wetland sites in the FLUXNET-CH4 Version 1.0 dataset
against wetlands globally. Specifically, we asked how representative the bioclimatic conditions of our sites are,
relative to bioclimatic conditions in all wetlands globally. Parameters defining bioclimatic conditions are selected
from those known to affect CH$_4$ production, consumption, and transport processes (e.g., energy, moisture, substrate
availability, and vegetation). When evaluating bioclimatic variables individually, the distribution across the network
was significantly different from the global distribution (alpha > 0.05; two-tailed Kolmogorov-Smirnov tests; see Table
B3).
When considering the four bioclimatic variables, MAT, LE, EVI and SRWI in a PCA, we found that our
tower network generally samples the bioclimatic conditions of global wetland cover, but some noticeable gaps remain

(Fig. 6). Three clusters of the world's wetland-dense regions are identified, but are not equally sampled by the network.
A cluster of low temperature wetlands is sampled by a large number of high-latitude sites. The other two wetland
clusters are not as well sampled: a high temperature and LE cluster is represented only by two towers (ID-Pag and
MY-MLM), while drier and temperate and subtropical wetlands including large swathes of the Sahel only have a site
in Botswana as their closest tower.

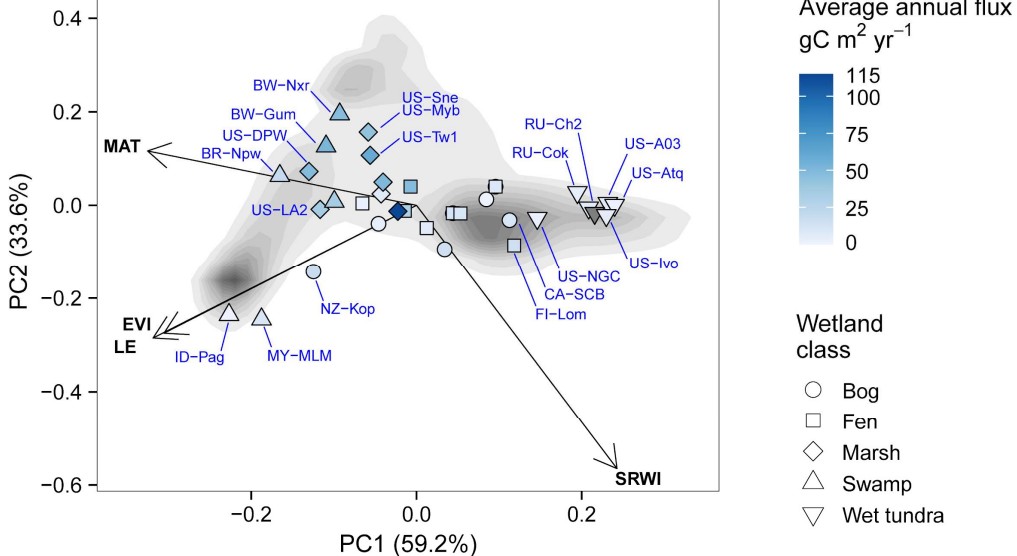

**Figure 6: Principal Component Analysis displaying the distribution of freshwater wetland sites (blue points) along the two**
**main principal components together accounting for 91.9% of variance. Tower sites are represented as the larger points of**
**shapes representing wetland type and color shade representing the annual CH$_4$ flux (greyed points represent sites for which**
**<6 months of flux data was available to estimate annual budget). The size of wetland points are made larger for visual**
**clarity and site codes are labelled in blue. The background shades of grey represent the density of land pixels (excluding**
**Greenland and Antarctica) that have a >5% wetland fraction according to the WAD2M map (Zhang et al., In Review).**
**Loading variables are represented by the arrows: mean annual temperature (MAT), simple ratio water index (SRWI),**
**latent heat flux (LE) and enhanced vegetation index (EVI).**

Evaluating the bioclimatic dissimilarity of global wetlands to the FLUXNET-CH4 Version 1.0 network
shows the least captured regions are in the tropics and mountainous regions (Fig. 7A). Sparse coverage in the tropics
also means that the few existing towers occupy a critical place in the network, particularly as tropical wetlands are the
largest CH$_4$ emitters (Bloom et al., 2017; Poulter et al., 2017). Highly dissimilar wetlands are limited in extent and
distributed across all latitudes, but the average dissimilarity is higher in temperate and tropical latitudes (Fig. 7B). To
evaluate the importance of individual towers in the network, we estimated the geographical area to which it is most
analogous in bioclimate-space (Fig. 7C). We found that some towers have disproportionately large constituencies (i.e.,
wetland areas that share the same closest bioclimatic analog tower). Towers in Indonesia (ID-Pag), Brazilian Pantanal
(BR-Npw), and Botswana floodplains (BW-Nxr) represent the closest climate analog for much of the tropics (678,
300 & 284 thousand km$^2$ respectively) while CA-SCB represents a vast swath (291 thousand km$^2$) of boreal/arctic
regions (Fig. 7D).

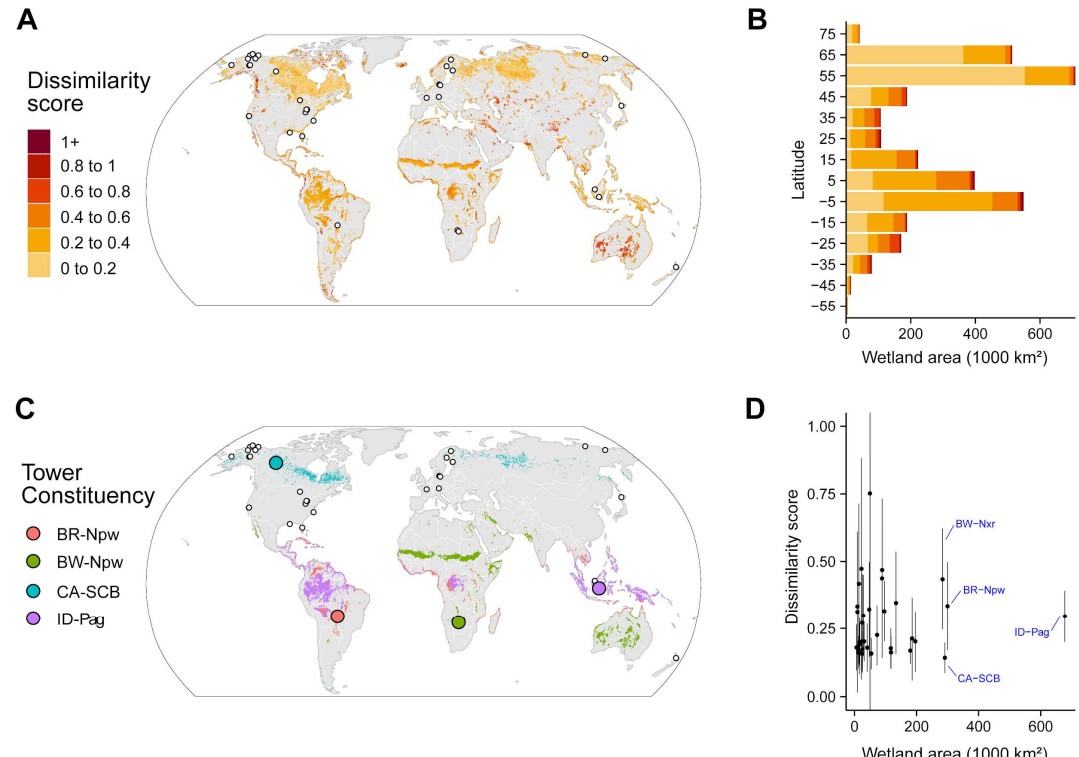

**Figure 7: (A) Distance in bioclimatic space between global land surface and the FLUXNET-CH4 Version 1.0 tower network.**
**The Euclidean distance was computed on the five bioclimatic variables and was then standardized by the average distance**
**within-network. Most of the land surface has a dissimilarity score lower than 1, meaning these areas are closer than the**
**average tower distance (lower dissimilarity score means a similar bioclimate to that represented by towers in the network).**
**However, this pattern reflects more the sparsity of the tower network than a similarity of the land surface to the network.**
**Areas with <5% coverage by wetlands were excluded to focus on wetland-dense regions. (B) Latitudinal distribution of**
**dissimilarity score, (C) Map of the four largest tower constituencies, (D) Scatterplot of wetland area in each tower**
**constituency plotted against the average dissimilarity score (point) and +/- standard deviation (error bar).**

Our assessment of wetland CH$_4$ tower coverage determines the ability of our dataset to represent global
wetland distributions and highlights some clear representation gaps in the network - tropical, humid, and mountainous
regions. Other geographic regions such India, China and Australia, where towers exist but are not included in the
current network should be prioritized when expanding the network, even though they are not among the most distant
areas to the current network. Similar representativeness assessments have been developed for CO$_2$ tower networks to
identify gaps and priorities for expansion (Jung et al., 2009; Kumar et al., 2016). To improve the geographic coverage
of the network for representing global-scale fluxes, locations for new tower sites can be targeted to cover bio-
climatically distant areas from the current network (Villarreal et al., 2019). Candidate regions for expansion that are
both high CH$_4$ emitting and under-sampled are: African Sahel, Amazon basin, Congo basin, South-East Asia.
Moreover, sites should be established in other ecosystem types, especially lakes and reservoirs (Bastviken et al., 2011;
Deemer et al., 2016; Matthews et al., 2020) in most climatic zones in order to capture CH$_4$ fluxes from these
ecosystems. Understanding the representativeness of the network is essential when inferring general patterns of flux
magnitude, seasonality, and drivers from the tower data (Villarreal et al., 2018). We produced a first-order
representativeness of average bioclimatic conditions, but temporal representativeness (across seasons, climate
anomalies and extreme events) is particularly needed given the episodic nature of CH$_4$ fluxes (Chu et al., 2017;
Mahecha et al., 2017; Göckede et al., 2019).



608   Assessing representation of wetland $CH_4$ sites is complicated by the fact that wetlands occupy only a fraction
609 of most landscapes (except wetland dense regions such as Siberian Lowlands, Hudson Bay Lowlands, Congo basin,
610 etc.) and that not all relevant factors affecting $CH_4$ production and consumption could be considered in our analysis.
611 For instance, our assessment of representation did not consider wetland types as such maps are limited (Gallant, 2015).
612 The attribution of representativeness is further complicated by the fact that many EC tower locations are subject to
613 small-scale variability within the field of view, or footprint, of the sensor. Consequently, the individual time steps
614 within EC flux time series may represent a mixture of different wetland types, or different fractions of wetland
615 contribution to the total flux, varying with wind direction, atmospheric stability, or season. This further complicates
616 upscaling efforts. Additionally, this representativeness analysis did not apply weights to the drivers to reflect their
617 varying influence on $CH_4$ flux. Such weights can be included in future versions as they are generated by a cross-
618 validated machine learning approach (Jung et al., 2020). Future efforts will include the dissimilarity index from this
619 analysis as a metric of extrapolation in a $CH_4$ flux upscaling effort.

### 621 3.3 Freshwater wetland flux seasonality

622 We used seasonality parameters extracted by Timesat to describe typical seasonal patterns in freshwater wetland
623 $CH_4$ fluxes, and to compare them with seasonality in soil temperature and GPP.  Of the 42 freshwater wetland sites
624 in FLUXNET-CH4 Version 1.0, 36 had sufficient data series to extract seasonality parameters.

### 625 3.3.1 Seasonal flux comparisons by latitudinal bands

626   $CH_4$ flux and seasonality varied substantially across latitudinal bands (northern, temperate, subtropical, and
627 tropical) (Fig. 8).  Annual fluxes for temperate, and subtropical sites were significantly higher than for northern sites
628 ($8.7 \pm 5.0$, $29.7 \pm 25.2$, $40.1 \pm 14.6$, and $24.5 \pm 20.7$  g C m$^{-2}$ yr$^{-1}$ for northern, temperate, subtropical, and tropical
629 respectively, p<0.0001 using Kruskal Wallis and post hoc comparisons; Fig. 8a), and tropical sites were similar to all
630 other latitudinal bands likely because of their small sample size.  The ratio of seasonal amplitude to peak flux provides
631 a measure of the relative seasonal increase in emissions compared with baseline, where a ratio of zero indicates no
632 seasonal change in amplitude, a ratio of one indicates the off-season flux is zero, and values over one means the off-
633 season baseline $CH_4$ fluxes were negative (i.e., uptake). Average amplitude to peak flux ratios were similar across all
634 latitudinal bands ($0.9 \pm 0.1$, $0.9 \pm 0.1$, $0.9 \pm 0.1$, $1.0 \pm 0.7$, for northern, temperate, subtropical, and tropical,
635 respectively; Fig. 8b). The spring increase in $CH_4$ emissions begins later in northern sites compared with temperate
636 and subtropical sites (end of May versus April, respectively, p=0.001; Fig. 8c), while tropical sites vary widely in
637 elevated emission season start date. Northern sites also have shorter $CH_4$ flux season lengths ($138 \pm 24$ days) compared
638 to temperate sites ($162 \pm 32$ days), and both are shorter than subtropical sites ($209 \pm 43$ days; p<0.0001; Fig. 8d).  On
639 average, $CH_4$ flux peaks earlier for temperate sites compared to northern (p = 0.008) and subtropical sites (p = 0.02;
640 mid to late July compared with early August; Fig. 8e), while tropical sites again vary widely.  Given their unique
641 seasonality, and low number of site-years (n = 9), tropical systems are discussed separately in Sect. 3.3.3, and not
642 included in the comparisons in the remainder of this section.  While our results on $CH_4$ seasonality corroborate
643 expected trends for these latitudinal bands, they provide some of the first estimates of $CH_4$ seasonality parameters and
644 ranges across a global distribution of sites.

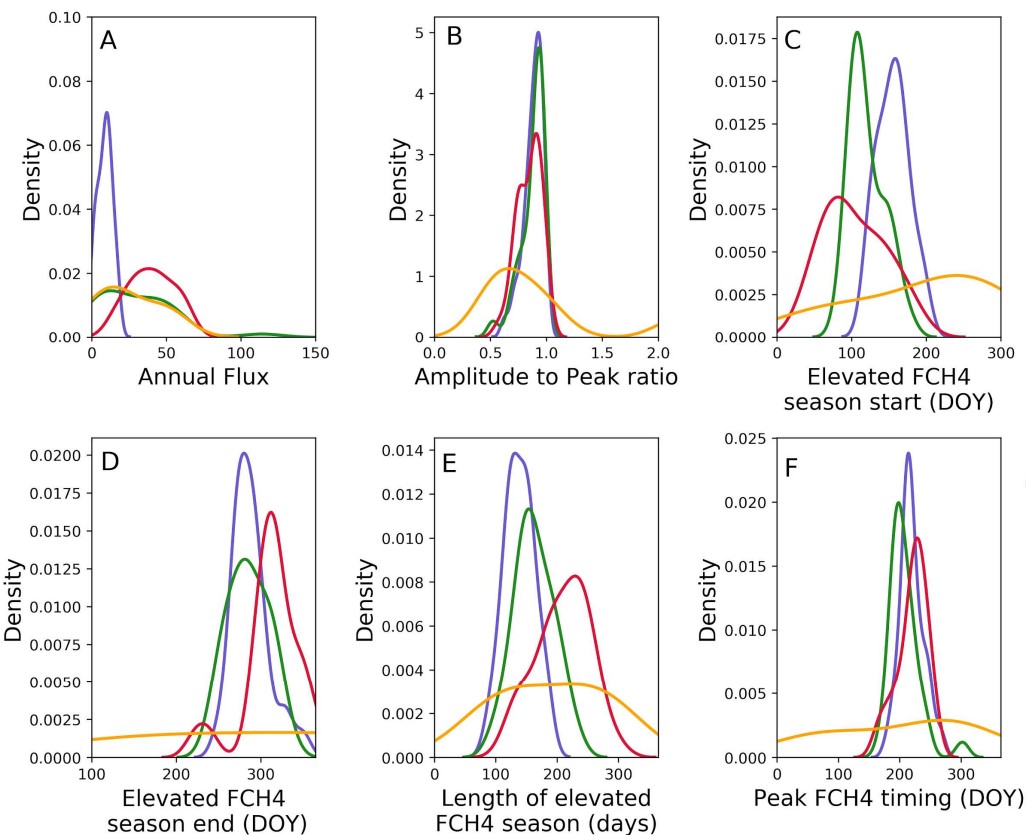

**Figure 8: (a) Annual flux (g C m⁻² yr⁻¹), (b) Ratio of seasonal amplitude to seasonal peak, where values of 0 indicate uniform annual flux, values of one indicate zero off-season fluxes, and values exceeding one indicate negative off-season fluxes, (c) CH₄ flux elevated emissions season start, (d) Length of CH₄ flux season, and (e) peak timing metrics for high, mid, and lower latitude wetlands (purple, green, red, respectively) plotted using the kernel density function. Latitudinal bands are as follows: northern ( > 60° ), temperate (between 40° and 60° ), subtropical (between 20° and 40°), and tropical (< 20° ), though the site-year totals vary between these groups (n = 57, n = 36, n = 39, and n = 9 respectively). All total flux values and elevated season start values are positive, and the apparent continuation of the data distribution into negative values is an artifact of the kernel density function. Southern Hemisphere sites below 20° S were shifted by 182 days.**

We found that latitudinal groups show strong differences in absolute flux across quarters, and narrower differences in percentage of annual flux (Fig. 9a versus 9b). Thus, the AMJ quarter has a similar contribution to the annual flux across latitudes, regardless of the absolute annual flux. Methane fluxes (Fig. 9a) are highest during JAS for northern, temperate, and subtropical sites and highest in AMJ and JAS for temperate sites (p<0.01). Though CH₄ fluxes in northern sites are most commonly measured during warm summer months (Sachs et al., 2010; Parmentier et al., 2011), fluxes in JFM and OND (50% of the yearly duration) on average make up $18.1 \pm 3.6\%$, $15.3 \pm 0.1\%$, and $31.2 \pm 0.1\%$ (northern, temperate, subtropical, respectively) of annual emissions. This pattern indicates that a substantial fraction of annual CH₄ fluxes occurs during cooler months. The fraction would be even higher if we added April, May, and September emissions to the northern (> 60 °N) sites, as done in (Zona et al., 2016), where > 50% of emissions were found to come from non-growing season months. The contribution of non-growing season CH₄ emissions to annual fluxes has previously been described for arctic and boreal regions (Zona et al., 2016; Treat et al.,

2018) and our analysis suggests comparable contributions in temperate and subtropical systems for the same quarterly
periods.

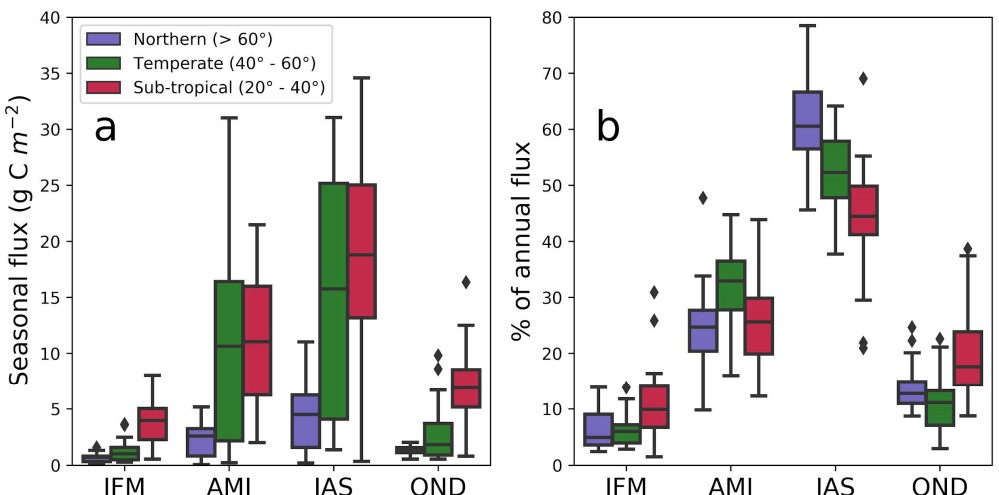

**Figure 9: (A) Quarterly contribution to total annual flux in g C m⁻², and (B) percentage of annual flux. Sites were divided**
**into northern (> 60° N), temperate (40° N - 60° N), and subtropical (20° N - 40° N).  Quarters with continuous data gaps**
**exceeding 30 days were excluded.  We used the following quarterly periods:  January/February/March (JFM),**
**April/May/June (AMJ), July/August/September (JAS), and October/November/December (OND).  Tropical sites are**
**discussed separately in Sect. 3.3.3 because of their unique seasonality and low number of sites.**

**3.3.2 Predictors of CH₄ flux phenology**

The start of the elevated CH₄ flux season, and how long the elevated flux season lasts, correlates strongly
with mean annual air temperature (Fig. 10; p<0.0001 for each).  Methane flux begins to increase roughly two
685 months earlier in the warmest systems (mean annual temperature > 20 °C) compared to the coldest (mean annual
temperature near -10 °C), though several of the warmer sites have high variability. Our data suggest that the CH₄
season starts 2.8 ± 0.5 days earlier for every degree Celsius increase in mean annual temperature (Fig. 10a).  In
contrast, the end of the CH₄ emission season is not correlated with mean annual temperature but a positive trend
exists despite high variability in warmest and coldest sites (Fig. 10b). The high variability seen in the end of CH₄
season at northern sites is important to note and would likely be better resolved by incorporating other seasonality or
phenological characteristics, such as moisture, active layer depth, and plant community composition. Plants with
aerenchymatous tissue, for example, influence the timing of plant-mediated CH₄ flux and are a key source of
uncertainty while predicting CH₄ seasonality for northern wetlands (Xu et al., 2016). Despite the relative lack of
trend with season end date, the season length is still positively correlated with mean annual temperature, with the
warmest sites having roughly three more months of seasonally elevated CH₄ emissions than the coldest sites (Fig.
10c).  Methane season length increases 3.6 ± 0.6 days for every °C increase in mean annual temperature (note that
these relationships are correlations, and we cannot disentangle causality with this analysis).  Temperature is highly
correlated with other parameters (i.e., radiation, days of snow cover, etc.), so CH₄ flux is also likely to correlate with
other environmental parameters.

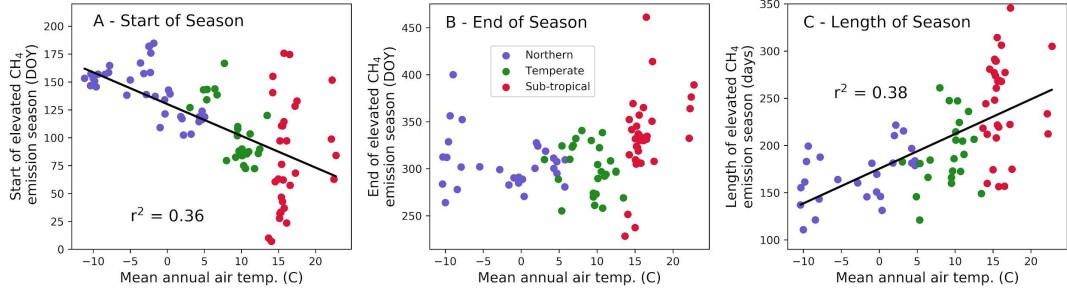

**Figure 10.  The (a) start of the elevated CH$_4$ emission season (y = -2.8x + 130, with 'x' in °C and 'y' in day of year), (b) the end of the emission season, and (c) the length of the emission season with mean annual site air temperature (y = 3.6x + 176.6,  with 'x' in °C and 'y' in days). Each point represents a site-year of data and all reported r$^2$ are significant to p < 0.0001).   Tropical sites are discussed separately in Sect. 3.3.3.**

Although the spring onset of increasing CH$_4$ emissions correlates with mean annual air temperature, on average it lags the spring increase in the shallowest soil temperatures by $31 \pm 40$ days (Fig. 11), with very few instances of CH4 emissions beginning before seasonal soil temperatures increase (and by $20 \pm 50$ days for the deepest temperature probes).  In contrast, for roughly half of the sites, CH$_4$ emission increases prior to seasonal GPP (a proxy for fresh substrate availability) increases.  This suggests that the initiation of increased CH$_4$ fluxes at the beginning of the season is not limited by availability of substrate derived from recent photosynthate, especially in cooler climates. Additionally, the onset of CH$_4$ fluxes tends to occur closer to the onset of soil temperature increase for cooler temperature sites (sites with later start dates tend to be cooler; Fig. 11a). This result is likely attributable to the direct influence of increased temperature on microbial processes as well as the indirect influences of snow melt, both via release of CH$_4$ from the snowpack as well as a higher water table leading to more CH$_4$ production (Hargreaves et al., 2001; Tagesson et al., 2012; Mastepanov et al., 2013; Helbig et al., 2017).  These observed trends hold for the entire temperature or GPP range of freshwater wetland sites, but are not necessarily applicable within individual latitudinal bands.

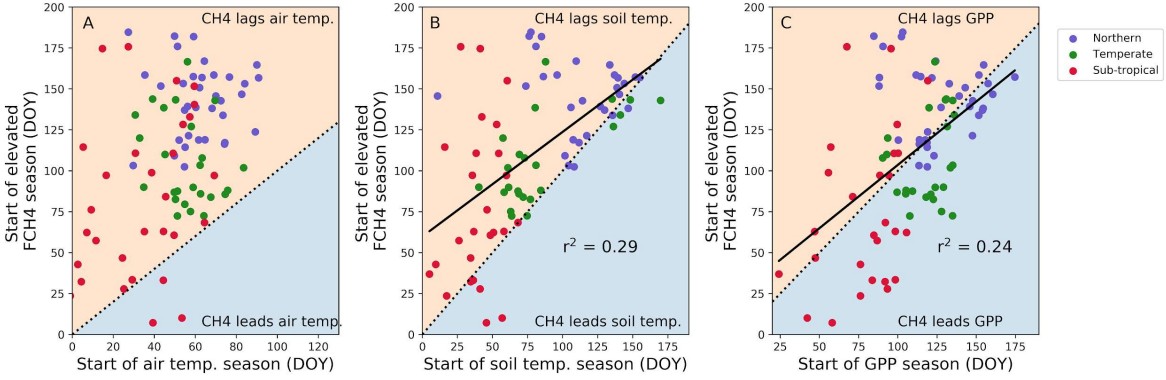

**Figure 11. Relationship between the onset of the onset of the CH$_4$ emission season to (a) the beginning of the air warming,**
**(b) soil warming at the shallowest probe depth per site, and (c) gross primary productivity (GPP) increase for the subset**
**of sites with soil temperature data.  Each point represents a site-year of data.  Dashed lines represent a 1:1 relationship,**





solid lines are regression fits. **On average, the CH$_4$ emission season lags the soil temperature increase by 31 ± 40 days,**
**and is more synchronous with GPP.**

In contrast with the CH$_4$ season-start timing, the timing of the CH$_4$ peak does not correlate with either the
timing of the soil temperature peak or the GPP peak (Fig. A1). For 63% of the sites, the average timing of peak CH$_4$
emissions lags the soil temperature peak, and at 83% of the sites average peak CH$_4$ lags peak GPP (Fig. A1).
Although there is no simple relationship between absolute CH$_4$ peak timing and the environmental drivers we
investigated, there is a correlation ($p = 0.0005$) between the relative timing of peak CH$_4$ compared to season onset
(calculated as described in Section 2.3) and mean annual air temperature (Fig. 12a). For cooler sites, the peak of
seasonal CH$_4$ emissions occurs closer to the onset of the CH$_4$ emission season than the end of the season, resulting in
an asymmetrical seasonal flux shape that is illustrated in Fig. 2a. Soil temperature also peaks earlier in the season
for cooler wetlands, though the relationship is not as pronounced ($p = 0.009$, Fig. 12b). In contrast, GPP peaks later
in the season for cooler wetlands ($p = 0.009$, Fig. 12c). Previous work on Arctic sites (sites US-Ivo, US-Beo, US-
Atq, US-Bes, and RU-CH2) has highlighted the asymmetrical annual CH$_4$ peak, with higher fall emissions being
attributed to the "zero curtain" period when soil below the surface remains thawed for an extended period of time
due to snow insulation (Zona et al., 2016; Kittler et al., 2017). Furthermore, soils can stay above the "zero curtain"
range for an extended time into the fall and winter (Helbig et al., 2017), which may also be caused by snow
insulation. The rapid onset of emissions in the spring following snowmelt could be attributed to the release of
accumulated CH$_4$ (Friborg et al., 1997), and other high latitude sites have seen similarly sharp increases in CH$_4$
emissions at snowmelt (Dise, 1992, Windsor, 1992). However, not all studies in high latitudes have observed
asymmetrical CH$_4$ emission peaks, pointing to the inherent complexity of these ecosystems (Rinne et al., 2007;
Tagesson et al., 2012).

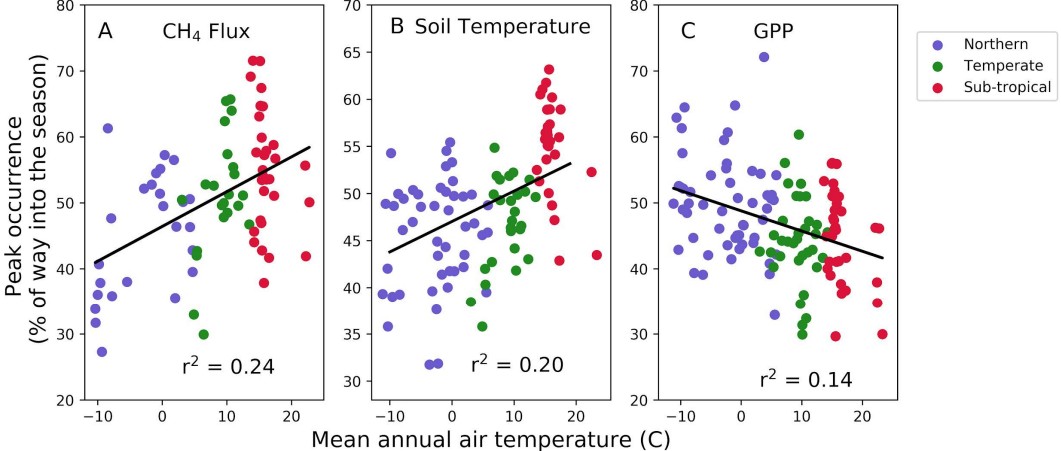

**Figure 12. Site-year peak CH$_4$ emission (a) and peak soil temperature (b) occur earlier in the season for sites with lower**
**mean annual temperatures. (c) GPP tends to peak earlier in the season for warmer sites, though the trend is weak. All r$^2$**
**values are significant at $p < 0.001$. Each point represents a site-year of data.**

**3.3.3 Uniqueness of tropical wetlands**

Tropical wetlands typically do not experience the large swings in temperature and GPP that contribute to
CH$_4$ flux seasonality in temperate and northern sites. Indeed, the relatively constant high temperatures and high
GPP in tropical ecosystems may lead to the lower ratio between seasonal amplitude and peak CH$_4$ flux compared
with temperate and northern sites (Fig. 8b). Tropical flux sites have historically been under-studied, leading to a





lack of synthesized information about these ecosystems. FLUXNET-CH4 Version 1.0 has five tropical wetland sites
(latitude between 20° S and 20° N), and one tropical rice site, representing 13 site-years of data. The tropical sites
are especially insightful as they provide the first estimates of $CH_4$ fluxes from large seasonal floodplain systems
found in the tropics.

We find a broad range of annual $CH_4$ fluxes across tropical sites in FLUXNET-CH4 Version 1.0. Annual
$CH_4$ flux emissions from two Southeast Asian flooded peat forests were relatively low, $0.01 \pm 0.1$ and $9.5 \pm 0.6$ g C
$m^{-2} yr^{-1}$ for ID-PAG and MY-MLM, respectively, which is consistent with annual $CH_4$ fluxes measured at another
peat forest in Indonesia (Deshmukh et al., 2020). In contrast, mean annual flux for a seasonally flooded swamp in
the Brazilian Pantanal region (BR-NPW) was over twice as high as MY-MLM, at $19.2 \pm 2.5$ g C $m^{-2} yr^{-1}$. Similarly
high annual fluxes were observed at the two Botswana swamp sites in the Okavango Delta ($51.7 \pm 10.6$ and $47.3 \pm$
$3.7$ g C $m^{-2} yr^{-1}$ for BW-GUM and BW-NXR, respectively), one of which is seasonally inundated and surrounded by
grassland (BW-NXR) and the other is a permanently flooded lagoon covered in a floating papyrus mat (BW-GUM).
The relatively low fluxes found at the two Southeast Asian peat forest sites indicate that these ecosystems may be
smaller $CH_4$ sources than expected, given their location in the humid tropics. Even the higher-emitting tropical sites
in Brazil and Botswana are still well within the range of annual flux typical in cooler latitudes (Fig. 1).

In addition to having highly variable flux magnitudes, the tropical sites differ from each other in their
seasonality. Methane flux hits a minimum around July for two sites (BW-GUM, latitude 18.965 °S and MY-MLM,
latitude 1.46 °N), while $CH_4$ flux increases through July and the subsequent months for the other Botswana site, BW-
NXR (latitude 19.548 °S). Site ID-Pag (latitude 2.32 °S) has minimal seasonality, whereas the flooded forest site in
Brazil (BR-NPW, latitude 16.49 °S ) has near-zero fluxes from approximately July to January, and consistently high
fluxes for the remainder of the year. The rice site PH-RiF (latitude 14.14 °N) has two annual $CH_4$ flux peaks, which
is consistent with some other rice sites and likely reflects management practices. Baseline flux values also differ, with
the two Botswana sites having the highest off-season fluxes (29 and 133 nmol $m^{-2} s^{-1}$ for BW-NXR and BW-GUM,
respectively, estimated by Timesat), MY-MLM having an intermediate baseline flux (16 nmol $m^{-2} s^{-1}$, estimated by
Timesat), and the remainder of the sites having essentially zero flux at baseline. While more tropical wetland data will
be needed to extract broad scale conclusions about these ecosystems, the six tropical sites in FLUXNET-CH4 provide
an important starting point for synthesis studies and highlight tropical wetland $CH_4$ variability.

**4.0 Data Availability**

Half-hourly and daily aggregations are available for download at https://fluxnet.org/data/fluxnet-ch4-
community-product/, along with a table containing site metadata compiled from Table B2. Variable descriptions and
units are provided in Table B1, and at https://fluxnet.org/data/fluxnet-ch4-community-product/. Each site has a unique
FLUXNET-CH4 DOI as listed in Table B2. All site data are available under the CC BY 4.0
(https://creativecommons.org/licenses/by/4.0/) copyright policy. The individual site DOIs are provided below in
Table 2. All seasonality parameters used in these analyses are available at https://doi.org/10.5281/zenodo.4408468.

**Table 2: Site identification (SITE_ID), data DOI, and DOI reference for each FLUXNET-CH4 site.**

| SITE_ID | DOI | DOI_REFERENCE |
|---------|-----|---------------|
| AT-Neu | 10.18140/FLX/1669365 | Wohlfahrt et al., 2020. |
| BR-Npw | 10.18140/FLX/1669368 | Vourlitis et al., 2020. |

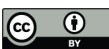



| BW-Gum | 10.18140/FLX/1669370 | Helfter, 2020a. |
|---|---|---|
| BW-Nxr | 10.18140/FLX/1669518 | Helfter, 2020b. |
| CA-SCB | 10.18140/FLX/1669613 | Sonnentag and Helbig, 2020a. |
| CA-SCC | 10.18140/FLX/1669628 | Sonnentag and Helbig, 2020b. |
| CH-Cha | 10.18140/FLX/1669629 | Hörtnagl et al., 2020a. |
| CH-Dav | 10.18140/FLX/1669630 | Hörtnagl et al., 2020b. |
| CH-Oe2 | 10.18140/FLX/1669631 | Hörtnagl, et al., 2020c. |
| CN-Hgu | 10.18140/FLX/1669632 | Niu and Chen, 2020. |
| DE-Dgw | 10.18140/FLX/1669633 | Sachs et al, 2020a. |
| DE-Hte | 10.18140/FLX/1669634 | Koebsch and Jurasinski, 2020. |
| DE-SfN | 10.18140/FLX/1669635 | Klatt et al., 2020. |
| DE-Zrk | 10.18140/FLX/1669636 | Sachs et al., 2020b. |
| FI-Hyy | 10.18140/FLX/1669637 | Mammarella et al. 2020. |
| FI-Lom | 10.18140/FLX/1669638 | Aurela et al., 2020. |
| FI-Si2 | 10.18140/FLX/1669639 | Vesala et al., 2020a. |
| FI-Sii | 10.18140/FLX/1669640 | Vesala et al., 2020b |
| FR-LGt | 10.18140/FLX/1669641 | Jacotot et al., 2020. |
| HK-MPM | 10.18140/FLX/1669642 | Lai and Liu, 2020. |
| ID-Pag | 10.18140/FLX/1669643 | Sakabe et al., 2020. |
| IT-BCi | 10.18140/FLX/1669644 | Magliulo et al., 2020. |
| IT-Cas | 10.18140/FLX/1669645 | Manca and Goded, 2020. |
| JP-BBY | 10.18140/FLX/1669646 | Ueyama et al., 2020. |
| JP-Mse | 10.18140/FLX/1669647 | Iwata, 2020a. |
| JP-SwL | 10.18140/FLX/1669648 | Iwata, 2020b. |
| KR-CRK | 10.18140/FLX/1669649 | Ryu et al., 2020. |
| MY-MLM | 10.18140/FLX/1669650 | Tang et al., 2020. |
| NL-Hor | 10.18140/FLX/1669651 | Dolman et al., 2020a. |
| NZ-Kop | 10.18140/FLX/1669652 | Campbell and Goodrich, 2020. |
| PH-RiF | 10.18140/FLX/1669653 | Albert and Wassmann, 2020. |
| RU-Ch2 | 10.18140/FLX/1669654 | Goeckede, 2020. |
| RU-Che | 10.18140/FLX/1669655 | Merbold et al., 2020. |
| RU-Cok | 10.18140/FLX/1669656 | Dolman et al., 2020b. |
| RU-Fy2 | 10.18140/FLX/1669657 | Varlagin, 2020. |
| SE-Deg | 10.18140/FLX/1669659 | Nilsson and Peichl, 2020. |
| UK-LBT | 10.18140/FLX/1670207 | Helfter, 2020c. |
| US-A03 | 10.18140/FLX/1669661 | Billesbach and Sullivan, 2020a. |
| US-A10 | 10.18140/FLX/1669662 | Billesbach and Sullivan, 2020b. |
| US-Atq | 10.18140/FLX/1669663 | Zona and Oechel, 2020a. |
| US-Beo | 10.18140/FLX/1669664 | Zona and Oechel, 2020b. |
| US-Bes | 10.18140/FLX/1669665 | Zona and Oechel, 2020c. |
| US-Bi1 | 10.18140/FLX/1669666 | Rey-Sanchez et al., 2020a. |





| US-Bi2 | 10.18140/FLX/1669667 | Rey-Sanchez et al., 2020b. |
|--------|----------------------|----------------------------|
| US-BZB | 10.18140/FLX/1669668 | Euskirchen and Edgar, 2020a. |
| US-BZF | 10.18140/FLX/1669669 | Euskirchen and Edgar, 2020b. |
| US-BZS | 10.18140/FLX/1669670 | Euskirchen and Edgar, 2020c. |
| US-CRT | 10.18140/FLX/1669671 | Chen and Chu, 2020a. |
| US-DPW | 10.18140/FLX/1669672 | Hinkle and Bracho, 2020. |
| US-EDN | 10.18140/FLX/1669673 | Oikawa, 2020. |
| US-EML | 10.18140/FLX/1669674 | Schuur, 2020. |
| US-Ho1 | 10.18140/FLX/1669675 | Richardson and Hollinger, 2020. |
| US-HRA | 10.18140/FLX/1669676 | Runkle et al., 2020. |
| US-HRC | 10.18140/FLX/1669677 | Reba et al., 2020. |
| US-ICs | 10.18140/FLX/1669678 | Euskirchen et al., 2020d. |
| US-Ivo | 10.18140/FLX/1669679 | Zona and Oechel, 2020d. |
| US-LA1 | 10.18140/FLX/1669680 | Holm et al., 2020a. |
| US-LA2 | 10.18140/FLX/1669681 | Holm et al., 2020b. |
| US-Los | 10.18140/FLX/1669682 | Desai and Thom, 2020a. |
| US-MAC | 10.18140/FLX/1669683 | Sparks, 2020. |
| US-MRM | 10.18140/FLX/1669684 | Schafer, 2020. |
| US-Myb | 10.18140/FLX/1669685 | Matthes et al., 2020. |
| US-NC4 | 10.18140/FLX/1669686 | Noormets et al., 2020. |
| US-NGB | 10.18140/FLX/1669687 | Torn and Dengel, 2020a. |
| US-NGC | 10.18140/FLX/1669688 | Torn and Dengel, 2020b. |
| US-ORv | 10.18140/FLX/1669689 | Bohrer and Morin, 2020a. |
| US-OWC | 10.18140/FLX/1669690 | Bohrer et al., 2020b. |
| US-PFa | 10.18140/FLX/1669691 | Desai and Thom, 2020b. |
| US-Snd | 10.18140/FLX/1669692 | Detto et al., 2020. |
| US-Sne | 10.18140/FLX/1669693 | Short et al., 2020. |
| US-Srr | 10.18140/FLX/1669694 | Windham-Myers et al., 2020. |
| US-StJ | 10.18140/FLX/1669695 | Vazquez-Lule and Vargas, 2020. |
| US-Tw1 | 10.18140/FLX/1669696 | Valach et al., 2020a. |
| US-Tw3 | 10.18140/FLX/1669697 | Chamberlain et al., 2020. |
| US-Tw4 | 10.18140/FLX/1669698 | Eichelmann et al., 2020. |
| US-Tw5 | 10.18140/FLX/1669699 | Valach et al., 2020b. |
| US-Twt | 10.18140/FLX/1669700 | Knox et al., 2020. |
| US-Uaf | 10.18140/FLX/1669701 | Iwata et al., 2020c. |
| US-WPT | 10.18140/FLX/1669702 | Chen and Chu, 2020b. |



**5.0 Conclusions**

The breadth and scope of $CH_4$ flux data in the FLUXNET-CH4 Version 1.0 dataset make it possible to
study the global patterns of $CH_4$ fluxes, particularly for global freshwater wetlands which release a substantial
fraction of atmospheric $CH_4$. We provide the first global estimates of flux patterns and predictors in $CH_4$ seasonality
using freshwater wetland data. In the seasonality analysis, we find that, on average, the seasonal increase in $CH_4$
emissions begins about three months earlier and lasts about four months longer at the warmest sites compared with
the coolest sites. We also find that the beginning of the $CH_4$ emission season lags the beginning of seasonal soil
warming by approximately one month, with almost no instances of $CH_4$ increasing before temperature increases.
Additionally, roughly half the sites have $CH_4$ emissions increasing prior to GPP increase; highlighting the
importance of substrate vs temperature limitations on wetland $CH_4$ emissions. Furthermore, relative to warmer
climates, wetland $CH_4$ emissions in cooler climates increase faster in the warming season and decrease slower in the
cooling season. This phenomenon has previously been noted on a regional scale and we show that it persists at the
global scale. Constraining the seasonality of $CH_4$ fluxes on a global scale can help improve the accuracy of global
wetland models.

FLUXNET-CH4 is an important new resource for the research community, but critical data gaps and
opportunities remain. The current FLUXNET-CH4 Version 1.0 is biased towards sites in boreal and temperate
regions, which influence the relationships presented in our analyses. Tropical ecosystems are estimated to account
for 64% of potential natural $CH_4$ emissions (<30° N, Saunois et al., 2020) but only account for 13% of the
FLUXNET-CH4 Version 1.0 sites in the dataset. Unsurprisingly, tropical sites in our network do not represent the
range of bioclimatic wetland conditions present in the tropics. Therefore, while maintaining flux towers in tropical
ecosystems is challenging, it is necessary to further constrain the global $CH_4$ cycle. Coastal wetlands are also poorly
represented in FLUXNET-CH4 even though there is evidence of substantial $CH_4$ emissions from these ecosystems,
so better representation across salinity gradients is warranted. Lastly, the average time series for FLUXNET-CH4
Version 1.0 is relatively short, only 3.7 site-years on average compared with 7.2 for $CO_2$ sites in FLUXNET
(Pastorello et al., 2020). Adding additional site-years of data from existing sites, as a complement to adding new
sites, will increase the community's ability to explain interannual variability in $CH_4$ emission and seasonality.
Nevertheless, FLUXNET-CH4 is an important and unprecedented resource with which to diagnose and understand
drivers of the global $CH_4$ cycle.

**Author contribution**

Kyle B. Delwiche oversaw the data release, performed the seasonality analysis, gathered metadata, and
prepared the manuscript with contributions from all co-authors. Sara Helen Knox gathered and standardized the
data, and gap-filled the $CH_4$ flux data. Avni Malhotra prepared the manuscript and gathered metadata. Etienne
Fluet-Chouinard did the representativeness analysis and prepared the manuscript. Gavin McNicol gathered data and
prepared the manuscript. Robert B. Jackson oversaw the data collection, processing, analysis, and release. Danielle
Christianson and You-Wei Cheah oversaw the FLUXNET-CH4 dataset release on fluxnet.org. Dario Papale,
Eleonora Canfora, and Carlo Trotta did the data collection, curation, and processing for a majority of the non-
American sites. Remaining co-authors contributed eddy-covariance data to FLUXNET-CH4 Version 1.0 dataset
and/or participated in editing the manuscript.

**Competing interests**

The authors declare that they have no conflict of interest.



### Acknowledgements

**We acknowledge primary support from the Gordon and Betty Moore Foundation (Grant GBMF5439, "Advancing Understanding of the Global Methane Cycle"; Stanford University) and from the John Wesley Powell Center for Analysis and Synthesis of the U.S. Geological Survey ("Wetland FLUXNET Synthesis for Methane" working group). Benjamin R. K. Runkle was supported by National Science Foundation Award 1752083. Ankur R. Desai acknowledges support of the DOE AmeriFlux Network Management Project. Masahito Ueyama was supported by ArCS II (JPMXD1420318865) and JSPS KAKENHI (20K21849). Dario Papale and Nina Buchmann acknowledge the support of the RINGO (GA 730944) H2020 EU project. Nina Buchmann and Kathrin Fuchs acknowledge the SNF project M4P (40FA40_154245/1) and InnoFarm (407340_172433). Nina Buchmann acknowledges support from the SNF for ICOS-CH Phases 1 and 2 (20FI21_148992, 20FI20_173691). Carlo Trotta acknowledges the support of the E-SHAPE (GA 820852) H2020 EU project. William J. Riley was supported by the US Department of Energy, BER, RGCM, RUBISCO project under contract no. DEAC02-05CH11231. Jessica Turner acknowledges support from NSF GRFP (DGE-1747503) and NTL LTER (DEB-1440297). Minseok Kang was supported by the National Research Foundation of Korea (NRF-2018 R1C1B6002917). Carole Helfter acknowledges the support of the UK Natural Environment Research Council (the Global Methane Budget project, grant number NE/N015746/1). Rodrigo Vargas acknowledges support from the National Science Foundation (1652594). Dennis Baldocchi acknowledges the California Department of Water Resources for a funding contract from the California Department of Fish and Wildlife and the United States Department of Agriculture (NIFA grant #2011-67003-30371), as well as the U.S. Department of Energy's Office of Science (AmeriFlux contract #7079856) for funding the AmeriFlux core sites. US-A03 and US-A10 are operated by the Atmospheric Radiation Measurement (ARM) user facility, a U.S. Department of Energy Office of Science user facility managed by the Biological and Environmental Research Program (doi:10.5439/1025039, doi:10.5439/1025274, doi:10.5439/1095578). Work at ANL was supported by the U.S. Department of Energy, Office of Science, Office of Biological and Environmental Research, under contract DE-AC02-06CH11357. Any use of trade, firm, or product names is for descriptive purposes only and does not imply endorsement by the U.S. Government. The CH-Dav, DE-SfN, FI-Hyy, FI-Lom, FI-Sii, FR-LGt, IT-BCi, SE-Deg and SE-Sto sites are part of the ICOS European Research Infrastructure. Oliver Sonnentag acknowledges funding by the Canada Research Chairs, Canada Foundation for Innovation Leaders Opportunity Fund, and Natural Sciences and Engineering Research Council Discovery Grant Programs for work at CA-SCC and CA-SCB. Benjamin Poulter acknowledges support from the NASA Carbon Cycle and Ecosystems Program. We thank Nathaniel Goenawan for his help with the representativeness analysis.**

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




**APPENDIX A**

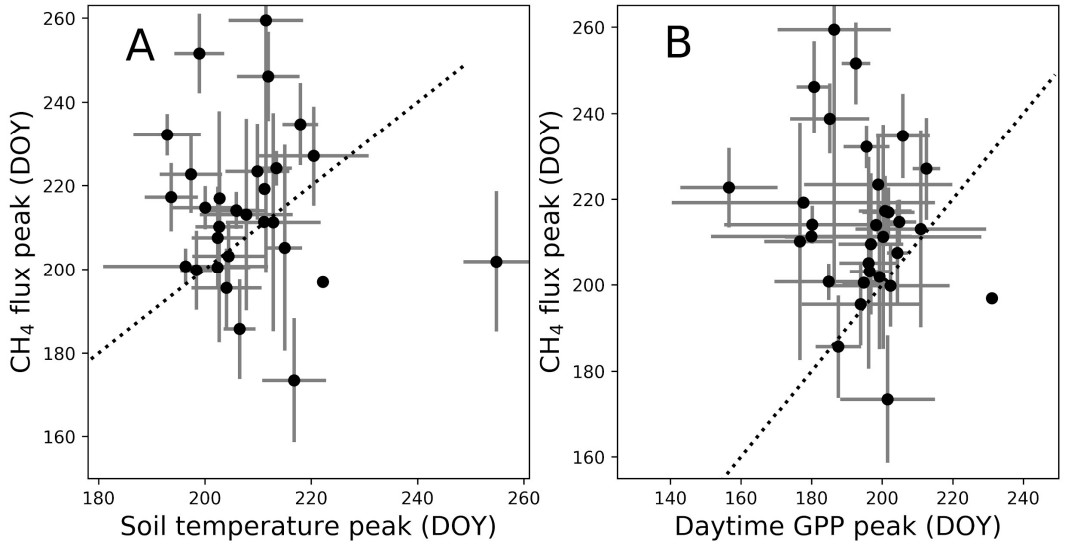

**Figure A1: Peak CH₄ flux timing versus peak GPP timing (A) and peak soil temperature timing (B). Points represent site**
**average and error bars represent standard deviations.**



**APPENDIX B**

**Table B1**: Data variable names, descriptions, and units

**FLUXNET-CH4 Data Variables**

This webpage describes data variables and file formatting for the FLUXNET-CH4 Community
Product.

**1. Data Variable: Base names**

Base names indicate fundamental quantities that are either measured or calculated/derived.
They can also indicate quantified quality information.

Table 1. Base names for data variables

| Variable | Description | Units |
|---|---|---|
| **TIMEKEEPING** | | |
| TIMESTAMP_START | ISO timestamp start of averaging period, used in half-hourly data | YYYYMMDDHHMM |
| TIMESTAMP_END | ISO timestamp end of averaging period, used in half-hourly data | YYYYMMDDHHMM |
| TIMESTAMP | ISO timestamp used in daily aggregation files | YYYYMMDD |
| **MET_RAD** | | |
| SW_IN | Shortwave radiation, incoming | W m-2 |
| SW_OUT | Shortwave radiation, outgoing | W m-2 |
| LW_IN | Longwave radiation, incoming | W m-2 |
| LW_OUT | Longwave radiation, outgoing | W m-2 |



| PPFD_IN | Photosynthetic photon flux density, incoming | µmolPhoton m-2 s-1 |
|---|---|---|
| PPFD_OUT | Photosynthetic photon flux density, outgoing | µmolPhoton m-2 s-1 |
| NETRAD | Net radiation | W m-2 |

**MET_WIND**

| USTAR | Friction velocity | m s-1 |
|---|---|---|
| WD | Wind direction | Decimal degrees |
| WS | Wind speed | m s-1 |

**HEAT**

| H | Sensible heat turbulent flux (with storage term if provided by site PI) | W m-2 |
|---|---|---|
| LE | Latent heat turbulent flux (with storage term if provided by site PI) | W m-2 |
| G | Soil heat flux | W m-2 |

**MET_ATM**

| PA | Atmospheric pressure | kPa |
|---|---|---|
| TA | Air temperature | deg C |



| VPD | Vapor Pressure Deficit | hPa |
|---|---|---|
| RH | Relative humidity, range 0-100 | % |

**MET_PRECIP**

| P | Precipitation | mm |
|---|---|---|

**PRODUCTS**

| NEE | Net Ecosystem Exchange | µmolCO2 m-2 s-1 |
|---|---|---|
| GPP | Gross primary productivity | µmolCO2 m-2 s-1 |
| RECO | Ecosystem respiration | µmolCO2 m-2 s-1 |

**GASES**

| FCH4 | Methane (CH4) turbulent flux (no storage correction) | nmolCH4 m-2 s-1 |
|---|---|---|

**MET_SOIL**

| TS | Soil temperature | deg C |
|---|---|---|
| WTD | Water table depth (negative values indicate below the surface) | m |



## 2. Data Variable: Qualifiers

Qualifiers are suffixes appended to variable base names that provide additional information about the variable. For example, the _DT qualifier in the variable label GPP_DT indicates that gross primary production (GPP) has been partitioned using the flux partitioning method from Lasslop et al. 2010.

Multiple qualifiers can be added, and they must **follow the order in which they are presented here**.

### 2.1. Qualifiers: General

General qualifiers indicate additional information about a variable.

· **_F :** Variable has been gap-filled by the FLUXNET-CH4 team.  Gaps in meteorological variables (including air temperature (TA), incoming shortwave (SW_IN) and longwave (LW_IN) radiation, vapor pressure deficit (VPD), pressure (PA), precipitation (P), and wind speed (WS) ) were filled with ERA-Interim (ERA-I) reanalysis data ((Vuichard and Papale 2015)).  Other variables were filled using the MDS approach in REddyProc (see Delwiche et al. 2020 for more details).

· **_DT :** Variable acquired using the flux partitioning method from (Lasslop et al. 2010), with values estimated by fitting the light-response curve.

· **_NT :** Variable acquired using the flux partitioning method from (Reichstein et al. 2005), with values estimated from night-time data and extrapolated to day time.

· **_RANDUNC:** Random uncertainty introduced from several different sources including errors associated with the flux measurement system (gas analyzer, sonic anemometer, data acquisition system, flux calculations), errors associated with turbulent transport, and statistical errors relating to the location and activity of the sites of flux exchange ("footprint heterogeneity") (Hollinger and Richardson 2005).

· **_ANNOPTLM :** Gap-filled variable using an artificial neural net routine from Matlab with the Levenberg-Marquardt algorithm as the training function, and parameters optimized across runs (more detail in (Sara Helen Knox et al. 2016; Sara H. Knox et al. 2019)).

· **_UNC :** Uncertainty introduced from ANNOPTLM gap-filling routine, as described in Knox et al. 2016 and Knox et al. 2019.

· **_QC :** Reports quality checks on FCH4 gap-filled data (_ANNOPTLM) based on length of data gap.  1 = data gap shorter than 2 months, 3 = data gap exceeds 2 months which could lead to poor quality gap-filled data. Nondimensional.

### 2.2. Qualifiers: Positional (_V)

Positional qualifiers are used to indicate relative positions of observations at the site. For FLUXNET-CH4, positional qualifiers are used to distinguish soil temperature probes for sites with more than one probe.  Probe depths for each positional qualifier per site are included in the





metadata file included with data download and also in Table B6 of Delwiche et al. 2020.  For
sites where the original database file release in Ameriflux, AsiaFlux, or EuroFlux contains
multiple probes at the same _V depth, we average values and report only the average for each
_V position.  The one exception to this is site US-UAF where the original positional qualifier from
the data we downloaded from Ameriflux had different depths for the same qualifier.  We still
averaged the probe data, so _V qualifiers from US-UAF represent an average of more than one
depth.

## 3.0 Missing data

Missing data are reported using -9999. Data for all days in a leap year are reported.

## 4.0 References

Hollinger, D. Y., and A. D. Richardson. 2005. "Uncertainty in Eddy Covariance Measurements
and Its Application to Physiological Models." *Tree Physiology* 25 (7): 873–85.
Knox, Sara Helen, Jaclyn Hatala Matthes, Cove Sturtevant, Patricia Y. Oikawa, Joseph
Verfaillie, and Dennis Baldocchi. 2016. "Biophysical Controls on Interannual Variability in
Ecosystem-Scale CO2and CH4exchange in a California Rice Paddy." *Journal of
Geophysical Research: Biogeosciences*. https://doi.org/10.1002/2015jg003247.
Knox, Sara H., Robert B. Jackson, Benjamin Poulter, Gavin McNicol, Etienne Fluet-Chouinard,
Zhen Zhang, Gustaf Hugelius, et al. 2019. "FLUXNET-CH4 Synthesis Activity: Objectives,
Observations, and Future Directions." *Bulletin of the American Meteorological Society* 100
(12): 2607–32.
Lasslop, Gitta, Markus Reichstein, Dario Papale, Andrew D. Richardson, Almut Arneth, Alan
Barr, Paul Stoy, and Georg Wohlfahrt. 2010. "Separation of Net Ecosystem Exchange into
Assimilation and Respiration Using a Light Response Curve Approach: Critical Issues and
Global Evaluation." *Global Change Biology*. https://doi.org/10.1111/j.1365-
2486.2009.02041.x.
Reichstein, Markus, Eva Falge, Dennis Baldocchi, Dario Papale, Marc Aubinet, Paul Berbigier,
Christian Bernhofer, et al. 2005. "On the Separation of Net Ecosystem Exchange into
Assimilation and Ecosystem Respiration: Review and Improved Algorithm." *Global Change
Biology*. https://doi.org/10.1111/j.1365-2486.2005.001002.x.
Vuichard, N., and D. Papale. 2015. "Filling the Gaps in Meteorological Continuous Data
Measured at FLUXNET Sites with ERA-Interim Reanalysis." *Earth System Science Data*.
https://doi.org/10.5194/essd-7-157-2015.




**Table B2-A: Site metadata, select data, and DOI links**

| | SITE_ID | SITE_NAME | SITE_PERSONNEL | COUNTRY | LAT | LON | DATA_DOI | YEAR_START | YEAR_END | UTC_OFFSET | ORIGINAL_DATA_SOURCE |
|---|---|---|---|---|---|---|---|---|---|---|---|
| 1 | AT-Neu | Neustift | Georg Wohlfahrt | Austria | 47.117 | 11.318 | 10.18140/FLX/1669365 | 2010 | 2012 | 1 | EuroFlux |
| 2 | BR-Npw | Northern Pantanal Wetland | George Vourlitis | Brazil | -16.498 | -56.412 | 10.18140/FLX/1669368 | 2013 | 2016 | -4 | AmeriFlux |
| 3 | BW-Gum | Guma | Carole Helfter | Botswana | -18.965 | 22.371 | 10.18140/FLX/1669370 | 2018 | 2018 | 2 | EuroFlux |
| 4 | BW-Nxr | Nxaraga | Carole Helfter | Botswana | -19.548 | 23.179 | 10.18140/FLX/1669518 | 2018 | 2018 | 2 | EuroFlux |
| 5 | CA-SCB | Scotty Creek Bog | Oliver Sonnentag, Manuel Helbig | Canada | 61.309 | -121.298 | 10.18140/FLX/1669613 | 2014 | 2017 | -7 | AmeriFlux |
| 6 | CA-SCC | Scotty Creek Landscape | Oliver Sonnentag, Manuel Helbig | Canada | 61.308 | -121.299 | 10.18140/FLX/1669628 | 2013 | 2016 | -7 | AmeriFlux |
| 7 | CH-Cha | Chamau | Nina Buchmann | Switzerland | 47.210 | 8.410 | 10.18140/FLX/1669629 | 2012 | 2016 | 1 | EuroFlux |
| 8 | CH-Dav | Davos | Nina Buchmann | Switzerland | 46.815 | 9.856 | 10.18140/FLX/1669630 | 2016 | 2017 | 1 | EuroFlux |
| 9 | CH-Oe2 | Oensingen crop | Nina Buchmann | Switzerland | 47.286 | 7.734 | 10.18140/FLX/1669631 | 2018 | 2018 | 1 | EuroFlux |
| 10 | CN-Hgu | Hongyuan | Shuli Niu, Weinan Chen | China | 32.845 | 102.590 | 10.18140/FLX/1669632 | 2015 | 2017 | 8 | EuroFlux |
| 11 | DE-Dgw | Dagowsee | Torsten Sachs | Germany | 53.151 | 13.054 | 10.18140/FLX/1669633 | 2015 | 2018 | 1 | EuroFlux |
| 12 | DE-Hte | Huetelmoor | Gerald Jurasinski | Germany | 54.210 | 12.176 | 10.18140/FLX/1669634 | 2011 | 2018 | 1 | EuroFlux |
| 13 | DE-SfN | Schechenfilz Nord | Hans Peter Schmid | Germany | 47.806 | 11.328 | 10.18140/FLX/1669635 | 2012 | 2014 | 1 | EuroFlux |
| 14 | DE-Zrk | Zarnekow | Torsten Sachs | Germany | 53.876 | 12.889 | 10.18140/FLX/1669636 | 2013 | 2018 | 1 | EuroFlux |
| 15 | FI-Hyy | Hyytiala | Timo Vesala, Ivan Mammarella | Finland | 61.847 | 24.295 | 10.18140/FLX/1669637 | 2016 | 2016 | 2 | EuroFlux |
| 16 | FI-Lom | Lompolojankka | Annalea Lohila | Finland | 67.997 | 24.209 | 10.18140/FLX/1669638 | 2006 | 2010 | 2 | EuroFlux |
| 17 | FI-Si2 | Siikaneva-2 Bog | Timo Vesala, Ivan Mammarella, Eeva-Stiina Tuittila | Finland | 61.837 | 24.197 | 10.18140/FLX/1669639 | 2012 | 2016 | 2 | EuroFlux |
| 18 | FI-Sii | Siikaneva | Timo Vesala, Ivan Mammarella, Eeva-Stiina Tuittila | Finland | 61.833 | 24.193 | 10.18140/FLX/1669640 | 2013 | 2018 | 2 | EuroFlux |
| 19 | FR-LGt | La Guette | Adrien Jacotot, Sébastien Gogo, Fatima Laggoun-Défarge, Laurent Perdereau | France | 47.323 | 2.284 | 10.18140/FLX/1669641 | 2017 | 2018 | 1 | EuroFlux |



| # | Code | Site name | PI / Contact | Country | Lat | Lon | DOI | Start | End | Records | Network |
|---|---|---|---|---|---|---|---|---|---|---|---|
| 20 | HK-MPM | Mai Po Mangrove | Derrick Lai, Jiangong Liu | Hong Kong | 22.498 | 114.029 | 10.18140/FLX/1669642 | 2016 | 2018 | 8 | EuroFlux |
| 21 | ID-Pag | Palangkaraya undrained forest | Takashi Hirano | Indonesia | -2.320 | 113.900 | 10.18140/FLX/1669643 | 2016 | 2017 | 7 | EuroFlux |
| 22 | IT-BCi | Borgo Cioffi | Vincenzo Magliulo | Italy | 40.524 | 14.957 | 10.18140/FLX/1669644 | 2017 | 2018 | 1 | EuroFlux |
| 23 | IT-Cas | Castellaro | Giovanni Manca, Ignacio Goded, Carsten Gruening, Ana Meijide | Italy | 45.070 | 8.718 | 10.18140/FLX/1669645 | 2009 | 2010 | 1 | EuroFlux |
| 24 | JP-BBY | Bibai bog | Masahito Ueyama | Japan | 43.323 | 141.811 | 10.18140/FLX/1669646 | 2015 | 2018 | 9 | AsiaFlux |
| 25 | JP-Mse | Mase rice paddy field | Akira Miyata | Japan | 36.054 | 140.027 | 10.18140/FLX/1669647 | 2012 | 2012 | 9 | AsiaFlux |
| 26 | JP-SwL | Suwa Lake | Hiroki Iwata | Japan | 36.047 | 138.108 | 10.18140/FLX/1669648 | 2016 | 2016 | 9 | AsiaFlux |
| 27 | KR-CRK | Cheorwon Rice paddy | Youngryel Ryu, Minseok Kang | Korea | 38.201 | 127.251 | 10.18140/FLX/1669649 | 2015 | 2018 | 9 | AsiaFlux |
| 28 | MY-MLM | Maludam National Park | Angela C. I. Tang, Guan Xhuan Wong, Lulie Melling | Malaysia | 1.454 | 111.149 | 10.18140/FLX/1669650 | 2014 | 2015 | 8 | AsiaFlux |
| 29 | NL-Hor | Horstermeer | Han Dolman | Netherlands | 52.240 | 5.071 | 10.18140/FLX/1669651 | 2007 | 2009 | 1 | EuroFlux |
| 30 | NZ-Kop | Kopuatai | Dave Campbell | New Zealand | -37.388 | 175.554 | 10.18140/FLX/1669652 | 2012 | 2015 | 13 | OzFlux |
| 31 | PH-RiF | Philippines Rice Institute flooded | Ma. Carmelita Alberto | Philippines | 14.141 | 121.265 | 10.18140/FLX/1669653 | 2012 | 2014 | 8 | EuroFlux |
| 32 | RU-Ch2 | Chersky reference | Matthias Goeckede | Russia | 68.617 | 161.351 | 10.18140/FLX/1669654 | 2014 | 2016 | 11 | EuroFlux |
| 33 | RU-Che | Cherski | Matthias Goeckede | Russia | 68.613 | 161.341 | 10.18140/FLX/1669655 | 2014 | 2016 | 11 | EuroFlux |
| 34 | RU-Cok | Chokurdakh | Han Dolman | Russia | 70.829 | 147.494 | 10.18140/FLX/1669656 | 2008 | 2016 | 11 | EuroFlux |
| 35 | RU-Fy2 | Fyodorovskoye dry spruce | Andrej Varlagin | Russia | 56.448 | 32.902 | 10.18140/FLX/1669657 | 2015 | 2018 | 3 | EuroFlux |
| 36 | SE-Deg | Degero | Matthias Peichl, Mats Nilsson | Sweden | 64.182 | 19.557 | 10.18140/FLX/1669659 | 2014 | 2018 | 1 | EuroFlux |
| 37 | UK-LBT | London_BT | Carole Helfter | UK | 51.522 | -0.139 | 10.18140/FLX/1670207 | 2011 | 2014 | 0 | EuroFlux |
| 38 | US-A03 | ARM-AMF3-Oliktok | Ryan Sullivan, David Cook, David Billesbach | USA | 70.495 | -149.882 | 10.18140/FLX/1669661 | 2015 | 2018 | -9 | AmeriFlux |
| 39 | US-A10 | ARM-NSA-Barrow | Ryan Sullivan, David Cook, David Billesbach | USA | 71.324 | -156.615 | 10.18140/FLX/1669662 | 2012 | 2018 | -9 | AmeriFlux |
| 40 | US-Atq | Atqasuk | Donatella Zona | USA | 70.470 | -157.409 | 10.18140/FLX/1669663 | 2013 | 2016 | -9 | AmeriFlux |

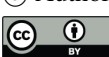



| | | | | | | | | | |
|---|---|---|---|---|---|---|---|---|---|
| 41 | US-Beo | Barrow Environmental Observatory (BEO) tower | Donatella Zona | USA | 71.281 | -156.612 | 10.18140/FLX/1669664 | 2013 | 2014 | -8 AmeriFlux |
| 42 | US-Bes | Barrow-Bes (Biocomplexity Experiment South tower) | Donatella Zona | USA | 71.281 | -156.597 | 10.18140/FLX/1669665 | 2013 | 2015 | -8 AmeriFlux |
| 43 | US-Bi1 | Bouldin Island Alfalfa | Dennis Baldocchi | USA | 38.099 | -121.499 | 10.18140/FLX/1669666 | 2016 | 2018 | -8 AmeriFlux |
| 44 | US-Bi2 | Bouldin Island corn | Dennis Baldocchi | USA | 38.109 | -121.535 | 10.18140/FLX/1669667 | 2017 | 2018 | -8 AmeriFlux |
| 45 | US-BZB | Bonanza Creek Thermokarst Bog | Eugenie Euskirchen | USA | 64.696 | -148.321 | 10.18140/FLX/1669668 | 2014 | 2016 | -9 AmeriFlux |
| 46 | US-BZF | Bonanza Creek Rich Fen | Eugenie Euskirchen | USA | 64.704 | -148.313 | 10.18140/FLX/1669669 | 2014 | 2016 | -9 AmeriFlux |
| 47 | US-BZS | Bonanza Creek Black Spruce | Eugenie Euskirchen | USA | 64.696 | -148.324 | 10.18140/FLX/1669670 | 2015 | 2016 | -9 AmeriFlux |
| 48 | US-CRT | Curtice Walter-Berger cropland | Jiquen Chen, Housen Chu | USA | 41.628 | -83.347 | 10.18140/FLX/1669671 | 2011 | 2012 | -5 AmeriFlux |
| 49 | US-DPW | Disney Wilderness Preserve Wetland | Charless Ross Hinkle, Rosvel Bracho, Scott Graham, Brian Benscoter | USA | 28.052 | -81.436 | 10.18140/FLX/1669672 | 2013 | 2017 | -5 AmeriFlux |
| 50 | US-EDN | Eden Landing Ecological Reserve | Patty Oikawa | USA | 37.616 | -122.114 | 10.18140/FLX/1669673 | 2018 | 2018 | -8 AmeriFlux |
| 51 | US-EML | Eight Mile Lake Permafrost thaw gradient, Healy Alaska. | Ted Schuur | USA | 63.878 | -149.254 | 10.18140/FLX/1669674 | 2015 | 2018 | -9 AmeriFlux |
| 52 | US-Ho1 | Howland Forest (main tower) | Andrew Richardson, David Hollinger | USA | 45.204 | -68.740 | 10.18140/FLX/1669675 | 2012 | 2018 | -5 AmeriFlux |
| 53 | US-HRA | Humnoke Farm Rice Field – Field A | Benjamin Runkle | USA | 34.585 | -91.752 | 10.18140/FLX/1669676 | 2017 | 2017 | -6 AmeriFlux |
| 54 | US-HRC | Humnoke Farm Rice Field – Field C | Benjamin Runkle | USA | 34.589 | -91.752 | 10.18140/FLX/1669677 | 2017 | 2017 | -6 AmeriFlux |
| 55 | US-ICs | Imnavait Creek Watershed Wet Sedge Tundra | Eugenie Euskirchen | USA | 68.606 | -149.311 | 10.18140/FLX/1669678 | 2014 | 2016 | -9 AmeriFlux |
| 56 | US-Ivo | Ivotuk | Donatella Zona | USA | 68.487 | -155.750 | 10.18140/FLX/1669679 | 2013 | 2016 | -9 AmeriFlux |
| 57 | US-LA1 | Pointe-aux-Chenes Brackish Marsh | Ken Krauss | USA | 29.501 | -90.445 | 10.18140/FLX/1669680 | 2011 | 2012 | -6 AmeriFlux |
| 58 | US-LA2 | Salvador WMA Freshwater Marsh | Ken Krauss | USA | 29.859 | -90.287 | 10.18140/FLX/1669681 | 2011 | 2013 | -6 AmeriFlux |
| 59 | US-Los | Lost Creek | Ankur Desai | USA | 46.083 | -89.979 | 10.18140/FLX/1669682 | 2014 | 2018 | -6 AmeriFlux |





| # | SITE_ID | SITE_NAME | SITE_PERSONNEL | COUNTRY | LAT | LON | DOI | | | | Network |
|---|---------|-----------|----------------|---------|-----|-----|-----|---|---|---|---------|
| 60 | US-MAC | MacArthur Agro-Ecology | Jed Sparks, Sam Chamberlain | USA | 27.163 | -81.187 | 10.18140/FLX/1669683 | 2013 | 2015 | -5 | AmeriFlux |
| 61 | US-MRM | Marsh Resource Meadowlands Mitigation Bank | Karina Schäfer | USA | 40.816 | -74.044 | 10.18140/FLX/1669684 | 2012 | 2013 | 5 | AmeriFlux |
| 62 | US-Myb | Mayberry Wetland | Dennis Baldocchi | USA | 38.050 | -121.765 | 10.18140/FLX/1669685 | 2010 | 2018 | -8 | AmeriFlux |
| 63 | US-NC4 | NC_AlligatorRiver | Asko Noormets | USA | 35.788 | -75.904 | 10.18140/FLX/1669686 | 2012 | 2016 | -5 | AmeriFlux |
| 64 | US-NGB | NGEE Arctic Barrow | Margaret Torn | USA | 71.280 | -156.609 | 10.18140/FLX/1669687 | 2012 | 2018 | -9 | AmeriFlux |
| 65 | US-NGC | NGEE Arctic Council | Margaret Torn | USA | 64.861 | -163.701 | 10.18140/FLX/1669688 | 2017 | 2018 | -9 | AmeriFlux |
| 66 | US-ORv | Olentangy River Wetland Research Park | Gil Bohrer | USA | 40.020 | -83.018 | 10.18140/FLX/1669689 | 2011 | 2015 | -5 | AmeriFlux |
| 67 | US-OWC | Old Woman Creek | Gil Bohrer | USA | 41.380 | -82.512 | 10.18140/FLX/1669690 | 2015 | 2016 | -5 | AmeriFlux |
| 68 | US-PFa | Park Falls/WLEF | Ankur Desai | USA | 45.946 | -90.272 | 10.18140/FLX/1669691 | 2010 | 2018 | -6 | AmeriFlux |
| 69 | US-Snd | Sherman Island | Dennis Baldocchi | USA | 38.037 | -121.754 | 10.18140/FLX/1669692 | 2010 | 2015 | -8 | AmeriFlux |
| 70 | US-Sne | Sherman Island Restored Wetland | Dennis Baldocchi | USA | 38.037 | -121.755 | 10.18140/FLX/1669693 | 2016 | 2018 | -8 | AmeriFlux |
| 71 | US-Srr | Suisun marsh - Rush Ranch | Lisamarie Windham-Myers | USA | 38.201 | -122.026 | 10.18140/FLX/1669694 | 2014 | 2017 | -8 | AmeriFlux |
| 72 | US-StJ | St Jones Reserve | Rodrigo Vargas | USA | 39.088 | -75.437 | 10.18140/FLX/1669695 | 2016 | 2016 | -5 | AmeriFlux |
| 73 | US-Tw1 | Twitchell Wetland West Pond | Dennis Baldocchi | USA | 38.107 | -121.647 | 10.18140/FLX/1669696 | 2011 | 2018 | -8 | AmeriFlux |
| 74 | US-Tw3 | Twitchell Alfalfa | Dennis Baldocchi | USA | 38.116 | -121.647 | 10.18140/FLX/1669697 | 2013 | 2014 | -8 | AmeriFlux |
| 75 | US-Tw4 | Twitchell East End Wetland | Dennis Baldocchi | USA | 38.103 | -121.641 | 10.18140/FLX/1669698 | 2013 | 2018 | -8 | AmeriFlux |
| 76 | US-Tw5 | East Pond Wetland | Dennis Baldocchi | USA | 38.107 | -121.643 | 10.18140/FLX/1669699 | 2018 | 2018 | -8 | AmeriFlux |
| 77 | US-Twt | Twitchell Island | Dennis Baldocchi | USA | 38.109 | -121.653 | 10.18140/FLX/1669700 | 2009 | 2017 | -8 | AmeriFlux |
| 78 | US-Uaf | University of Alaska, Fairbanks | Masahito Ueyama | USA | 64.866 | -147.856 | 10.18140/FLX/1669701 | 2011 | 2018 | -9 | AmeriFlux |
| 79 | US-WPT | Winous Point North Marsh | Jiquen Chen, Housen Chu | USA | 41.465 | -82.996 | 10.18140/FLX/1669702 | 2011 | 2013 | -5 | AmeriFlux |

**Column Descriptions**

SITE_ID          Site identification code as assigned by regional flux data network
SITE_NAME        Site name determined by site personnel
SITE_PERSONNEL   People associated with site FLUXNET-CH4 data
COUNTRY          Site country
LAT              Latitude
LON              Longitude



| DATA_DOI | DOI link for site FLUXNET-CH4 data |
| YEAR_START | Year data begins |
| YEAR_END | Year data ends |
| UTC_OFFSET | Site data offset from Coordinated Universal Time (in hours) |
| ORIGINAL_DATA_SOURCE | Regional network hosting the site methane data that was incorporated into FLUXNET-CH4 |





**Table B2-B: Site metadata, select data, and DOI links**

| | SITE_ID | SITE_CLASSIFICATION | UPLAND_CLASS | IGBP | KOPPEN | MOSS_BROWN | MOSS_SPHAGNUM | AERENCHYMATOUS | ERI_SHRUB | TREE | DOM_VEG | IN_SEASONALITY_ANALYSIS |
|---|---|---|---|---|---|---|---|---|---|---|---|---|
| 1 | AT-Neu | Upland | Alpine meadow | GRA | Dfb | 0 | 0 | 1 | 0 | 0 | aerenchymatous | 0 |
| 2 | BR-Npw | Swamp | | WSA | Aw | 0 | 0 | 1 | 0 | 1 | tree | 0 |
| 3 | BW-Gum | Swamp | | WET | Bsh | 0 | 0 | 1 | 0 | 1 | aerenchymatous | 0 |
| 4 | BW-Nxr | Swamp | | GRA | Bsh | 0 | 0 | 1 | 0 | 1 | aerenchymatous | 0 |
| 5 | CA-SCB | Bog | | WET | Dfc | 0 | 1 | 1 | 1 | 0 | moss_sphagnum | 1 |
| 6 | CA-SCC | Upland | Needleleaf forest | ENF | Dfc | 0 | 1 | 0 | 1 | 1 | tree | 0 |
| 7 | CH-Cha | Upland | Grassland | GRA | Cfb | 0 | 0 | 1 | 0 | 0 | aerenchymatous | 0 |
| 8 | CH-Dav | Upland | Needleleaf forest | ENF | ET | 1 | 0 | 0 | 1 | 1 | tree | 0 |
| 9 | CH-Oe2 | Upland | Crop - wheat | CRO | Cfb | 0 | 0 | 1 | 0 | 0 | aerenchymatous | 0 |
| 10 | CN-Hgu | Upland | Alpine meadow | GRA | Cwc | 0 | 0 | 1 | 0 | 0 | aerenchymatous | 0 |
| 11 | DE-Dgw | Lake | | WAT | Cfb | 0 | 0 | 0 | 0 | 0 | no vegetation | 0 |
| 12 | DE-Hte | Fen | | WET | Dfb | 0 | 0 | 1 | 0 | 0 | aerenchymatous | 1 |
| 13 | DE-SfN | Bog | | WET | Cfb | 0 | 1 | 1 | 1 | 1 | tree | 1 |
| 14 | DE-Zrk | Fen | | WET | Dfb | 0 | 0 | 1 | 0 | 0 | aerenchymatous | 1 |
| 15 | FI-Hyy | Upland | Needleleaf forest | ENF | Dfc | 1 | 1 | 0 | 1 | 1 | tree | 0 |
| 16 | FI-Lom | Fen | | WET | Dfc | 1 | 1 | 1 | 1 | 0 | aerenchymatous | 1 |
| 17 | FI-Si2 | Bog | | WET | Dfc | 0 | 1 | 1 | 1 | 1 | moss_sphagnum | 1 |
| 18 | FI-Sii | Fen | | WET | Dfc | 0 | 1 | 1 | 0 | 0 | moss_sphagnum | 1 |
| 19 | FR-LGt | Fen | | WET | Cfb | 0 | 1 | 1 | 1 | 0 | aerenchymatous | 0 |
| 20 | HK-MPM | Mangrove | | EBF | Cfa | 0 | 0 | 1 | 0 | 1 | aerenchymatous | 0 |
| 21 | ID-Pag | Swamp | | EBF | Af | 0 | 0 | 1 | 0 | 1 | tree | 0 |
| 22 | IT-BCi | Upland | Crop - corn | CRO | Csa | 0 | 0 | 1 | 0 | 0 | aerenchymatous | 0 |
| 23 | IT-Cas | Rice | | CRO | Cfa | 0 | 0 | 1 | 0 | 0 | aerenchymatous | 0 |
| 24 | JP-BBY | Bog | | WET | Dfb | 0 | 1 | 1 | 1 | 0 | aerenchymatous | 1 |
| 25 | JP-Mse | Rice | | CRO | Cfa | 0 | 0 | 1 | 0 | 0 | aerenchymatous | 0 |
| 26 | JP-SwL | Lake | | WAT | Dfb | 0 | 0 | 1 | 0 | 0 | aerenchymatous | 0 |
| 27 | KR-CRK | Rice | | CRO | Dwa | 0 | 0 | 1 | 0 | 0 | aerenchymatous | 0 |



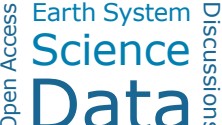
Open Access — Earth System Science Data Discussions

|  | Site | Description | Class |  | IGBP | Köppen |  |  |  |  | Vegetation |  |
|---|---|---|---|---|---|---|---|---|---|---|---|---|
| 28 | MY-MLM | Swamp |  |  | EBF | Af | 0 | 0 | 0 | 0 | 1 tree | 0 |
| 29 | NL-Hor | Drained | Grassland |  | GRA | Cfb | 0 | 1 | 0 | 0 | 0 aerenchymatous | 0 |
| 30 | NZ-Kop | Bog |  |  | EBF | Cfb | 1 | 1 | 0 | 0 | 0 aerenchymatous | 1 |
| 31 | PH-RiF | Rice |  |  | CRO | Am | 0 | 1 | 0 | 0 | 0 aerenchymatous | 0 |
| 32 | RU-Ch2 | Wet tundra |  |  | WET | Dfc | 0 | 1 | 0 | 1 | 0 aerenchymatous | 1 |
| 33 | RU-Che | Drained |  |  | WET | Dfc | 0 | 1 | 0 | 1 | 0 aerenchymatous | 0 |
| 34 | RU-Cok | Wet tundra |  |  | OSH | Dfc | 1 | 1 | 0 | 1 | 0 moss_sphagnum | 0 |
| 35 | RU-Fy2 | Upland | Needleleaf forest |  | ENF | Dfb | 1 | 0 | 0 | 1 | 1 tree | 0 |
| 36 | SE-Deg | Fen |  |  | GRA | Dfc | 1 | 1 | 0 | 1 | 0 moss_sphagnum | 1 |
| 37 | UK-LBT | Upland | Urban |  | URB | Cfb | 0 | 0 | 0 | 0 | 0 no vegetation | 0 |
| 38 | US-A03 | Wet tundra |  |  | BSV | ET | 0 | 1 | 0 | 0 | 0 moss_sphagnum | 0 |
| 39 | US-A10 | Wet tundra |  |  | BSV | ET | 0 | 1 | 0 | 0 | 0 moss_sphagnum | 0 |
| 40 | US-Atq | Wet tundra |  |  | WET | ET | 1 | 1 | 0 | 1 | 0 aerenchymatous | 1 |
| 41 | US-Beo | Wet tundra |  |  | WET | ET | 1 | 1 | 0 | 0 | 0 aerenchymatous | 1 |
| 42 | US-Bes | Wet tundra |  |  | WET | ET | 0 | 1 | 0 | 0 | 0 aerenchymatous | 1 |
| 43 | US-Bi1 | Drained | Crop - alfalfa |  | CRO | Csa | 0 | 0 | 1 | 0 | 0 aerenchymatous | 0 |
| 44 | US-Bi2 | Drained | Crop - corn |  | CRO | Csa | 0 | 0 | 1 | 0 | 0 aerenchymatous | 0 |
| 45 | US-BZB | Bog |  |  | WET | Dfd | 0 | 1 | 0 | 1 | 0 eri_shrub | 1 |
| 46 | US-BZF | Fen |  |  | WET | Dfd | 1 | 1 | 1 | 0 | 0 aerenchymatous | 1 |
| 47 | US-BZS | Upland | Needleleaf forest |  | ENF | Dfd | 1 | 0 | 0 | 0 | 1 tree | 0 |
| 48 | US-CRT | Upland | Crop - soy |  | CRO | Dfa | 0 | 1 | 0 | 0 | 0 aerenchymatous | 0 |
| 49 | US-DPW | Marsh |  |  | WET | Cwa | 0 | 1 | 0 | 1 | 0 aerenchymatous | 1 |
| 50 | US-EDN | Salt marsh |  |  | WET | Csa | 0 | 1 | 0 | 0 | 0 aerenchymatous | 0 |
| 51 | US-EML | Upland | Tundra |  | OSH | ET | 1 | 1 | 0 | 1 | 0 aerenchymatous | 0 |
| 52 | US-Ho1 | Upland | Needleleaf forest |  | ENF | Dfb | 0 | 0 | 0 | 0 | 1 tree | 0 |
| 53 | US-HRA | Rice |  |  | CRO | Cfa | 0 | 1 | 0 | 0 | 0 aerenchymatous | 0 |
| 54 | US-HRC | Rice |  |  | CRO | Cfa | 0 | 1 | 0 | 0 | 0 aerenchymatous | 0 |
| 55 | US-ICs | Wet tundra |  |  | WET | ET | 0 | 1 | 0 | 1 | 0 aerenchymatous | 1 |
| 56 | US-Ivo | Wet tundra |  |  | WET | ET | 1 | 1 | 0 | 1 | 0 aerenchymatous | 1 |
| 57 | US-LA1 | Salt marsh |  |  | WET | Cfa | 0 | 1 | 0 | 0 | 0 aerenchymatous | 0 |
| 58 | US-LA2 | Marsh |  |  | WET | Cfa | 0 | 1 | 0 | 0 | 0 aerenchymatous | 1 |
| 59 | US-Los | Fen |  |  | WET | Dfb | 0 | 1 | 0 | 1 | 1 eri_shrub | 1 |





| # | SITE_ID | SITE_CLASSIFICATION | UPLAND_CLASS | IGBP | KOPPEN | | | | MOSS_BROWN | MOSS_SPHAGNUM | | AERENCHYMATOUS |
|---|---------|---------------------|--------------|------|--------|---|---|---|---|---|---|---|
| 60 | US-MAC | Drained | Grassland | WET | Cwa | 0 | 0 | 1 | 0 | 0 | aerenchymatous | 0 |
| 61 | US-MRM | Salt marsh | | WET | Cfa | 0 | 0 | 1 | 0 | 0 | aerenchymatous | 0 |
| 62 | US-Myb | Marsh | | WET | Csa | 0 | 0 | 1 | 0 | 0 | aerenchymatous | 1 |
| 63 | US-NC4 | Swamp | | WET | Cfa | 0 | 0 | 1 | 0 | 1 | aerenchymatous | 1 |
| 64 | US-NGB | Wet tundra | | SNO | ET | 0 | 1 | 1 | 0 | 0 | moss_sphagnum | 0 |
| 65 | US-NGC | Wet tundra | | GRA | ET | 0 | 1 | 1 | 1 | 0 | eri_shrub | 0 |
| 66 | US-ORv | Marsh | | WET | Cfa | 0 | 0 | 1 | 0 | 0 | aerenchymatous | 1 |
| 67 | US-OWC | Marsh | | WET | Dfb | 0 | 0 | 1 | 0 | 0 | aerenchymatous | 1 |
| 68 | US-PFa | Upland | Mixed forest | MF | Dfb | 0 | 0 | 0 | 0 | 1 | tree | 0 |
| 69 | US-Snd | Drained | Grassland | GRA | Csa | 0 | 0 | 1 | 0 | 0 | aerenchymatous | 0 |
| 70 | US-Sne | Marsh | | GRA | Csa | 0 | 0 | 1 | 0 | 0 | aerenchymatous | 1 |
| 71 | US-Srr | Salt marsh | | WET | Csa | 0 | 0 | 1 | 0 | 0 | aerenchymatous | 0 |
| 72 | US-StJ | Salt marsh | | WET | Cfa | 0 | 0 | 1 | 0 | 0 | aerenchymatous | 0 |
| 73 | US-Tw1 | Marsh | | WET | Csa | 0 | 0 | 1 | 0 | 0 | aerenchymatous | 1 |
| 74 | US-Tw3 | Drained | Crop - alfalfa | CRO | Csa | 0 | 0 | 1 | 0 | 0 | aerenchymatous | 0 |
| 75 | US-Tw4 | Marsh | | WET | Csa | 0 | 0 | 1 | 0 | 0 | aerenchymatous | 1 |
| 76 | US-Tw5 | Marsh | | WET | Csa | 0 | 0 | 1 | 0 | 0 | aerenchymatous | 1 |
| 77 | US-Twt | Rice | | CRO | Csa | 0 | 0 | 1 | 0 | 0 | aerenchymatous | 0 |
| 78 | US-Uaf | Bog | | ENF | Dwc | 1 | 1 | 1 | 1 | 1 | moss_sphagnum | 1 |
| 79 | US-WPT | Marsh | | WET | Dfa | 0 | 0 | 1 | 0 | 0 | aerenchymatous | 1 |

**Column Descriptions**

SITE_ID — Site identification code as assigned by regional flux data network

SITE_CLASSIFICATION — Site classification based on literature description of sites

UPLAND_CLASS — For upland sites, category of upland type

IGBP — International Geosphere–Biosphere Programme (IGBP) ecosystem surface classification

KOPPEN — Koppen climate zone abbreviation

MOSS_BROWN — Presence/absence (1/0) brown moss. Presence/absence designated by Avni Malhotra using site-literature

MOSS_SPHAGNUM — Presence/absence (1/0) sphagnum moss. Presence/absence designated by Avni Malhotra using site-literature

AERENCHYMATOUS — Presence/absence (1/0) aerenchymatous vegetation. Presence/absence designated by Avni Malhotra using site-literature





| ERI_SHRUB | Presence/absence (1/0) ericaceous shrubs. Presence/absence designated by Avni Malhotra using site-literature |
| TREE | Presence/absence (1/0) trees. Presence/absence designated by Avni Malhotra using site-literature |
| DOM_VEG | Dominant vegetation type in tower footprint. Dom_veg provided to Avni Malhotra by site personnel via survey, except 15 sites where PIs did not answer and Avni Malhotra estimated dominant vegetation type based on site-literature |
| IN_SEASONALITY_ANALYSIS | Is site in freshwater wetland seasonality analysis? 1 = yes, 0 = no. |




**Table B2-C: Site metadata, select data, and DOI links**

| | SITE_ID | Mean_Air_Temp_C | Mean_Air_Temp_stdev_C | Ann_Flux_g_CH4-C_m-2 | Ann_Flux_stdev_g_CH4-C_m-2 | JFM_flux_g_CH4-C_m-2 | JFM_flux_stdev_g_CH4-C_m-2 | AMJ_flux_g_CH4-C_m-2 | AMJ_flux_stdev_g_CH4-C_m-2 | JAS_flux_g_CH4-C_m-2 | JAS_flux_stdev_g_CH4-C_m-2 | OND_flux_g_CH4-C_m-2 | OND_flux_stdev_g_CH4-C_m-2 |
|---|---|---|---|---|---|---|---|---|---|---|---|---|---|
| 1 | AT-Neu | 6.60 | 0.51 | 0.32 | 0.09 | 0.03 | | 0.05 | 0.04 | 0.16 | 0.07 | 0.09 | 0.01 |
| 2 | BR-Npw | 25.44 | 0.73 | 19.21 | 2.45 | 9.68 | | 8.52 | 1.15 | 0.01 | 0.16 | 0.15 | 0.15 |
| 3 | BW-Gum | 22.79 | | 51.73 | | | 1.46 | 19.32 | | | | | |
| 4 | BW-Nxr | 23.06 | | 47.32 | | | | 8.88 | | 16.90 | | 18.09 | |
| 5 | CA-SCB | -0.75 | 1.92 | 10.67 | 1.34 | | | 2.96 | 0.70 | 6.58 | 0.77 | 1.16 | 0.11 |
| 6 | CA-SCC | -0.24 | 2.04 | 6.15 | 1.05 | | | 1.79 | 0.45 | 3.41 | 0.67 | | |
| 7 | CH-Cha | 9.74 | 0.54 | 2.95 | 0.88 | 0.75 | 0.18 | 0.99 | 0.37 | 0.60 | 0.28 | 0.61 | 0.25 |
| 8 | CH-Dav | 4.37 | 0.09 | 1.21 | | 0.28 | 0.05 | 0.37 | 0.21 | 0.13 | | 0.24 | |
| 9 | CH-Oe2 | 11.00 | | 0.29 | | | | | | 0.14 | 0.15 | 0.13 | |
| 10 | CN-Hgu | 3.77 | 1.31 | 0.82 | 0.01 | | | 0.23 | 0.04 | 0.28 | 0.15 | 0.15 | |
| 11 | DE-Dgw | 9.72 | 0.38 | 8.97 | 2.06 | 0.07 | 0.06 | 1.49 | 0.57 | 4.15 | 0.76 | 2.51 | 1.05 |
| 12 | DE-Hte | 10.04 | 0.54 | 48.11 | 7.41 | 2.98 | 0.72 | 17.18 | 3.66 | 24.28 | 4.07 | 6.17 | 1.46 |
| 13 | DE-SfN | 8.28 | 0.72 | 3.62 | | 0.43 | 0.10 | 0.23 | | 1.67 | 0.43 | 0.72 | 0.25 |
| 14 | DE-Zrk | 9.55 | 0.51 | 30.53 | 0.96 | 1.18 | 0.24 | 12.27 | 1.55 | 16.18 | 1.47 | 1.51 | 0.56 |
| 15 | FI-Hyy | 4.36 | | | | | | | | -0.02 | | | |
| 16 | FI-Lom | -0.35 | 0.78 | 15.58 | 1.83 | 0.93 | 0.22 | 3.75 | 0.51 | 9.49 | 1.25 | 1.68 | 0.24 |
| 17 | FI-Si2 | 5.14 | 0.84 | 9.74 | 0.67 | | | 2.71 | 0.59 | 5.83 | 1.15 | 1.19 | 0.08 |
| 18 | FI-Sii | 4.72 | 0.42 | 12.43 | 3.36 | 0.68 | 0.11 | 3.34 | 0.75 | 6.79 | 2.90 | 1.58 | 0.42 |
| 19 | FR-LGt | 11.07 | 0.37 | 2.45 | | 0.02 | | 1.09 | | 0.85 | 0.29 | 0.29 | |
| 20 | HK-MPM | 23.75 | 0.10 | 11.09 | 0.51 | 0.97 | | 2.33 | 0.52 | 5.56 | 0.34 | 2.95 | 0.26 |
| 21 | ID-Pag | 26.57 | 0.19 | 0.09 | 0.00 | 0.19 | | 0.13 | | -0.24 | | 0.05 | |
| 22 | IT-BCi | 16.69 | 0.39 | | | -5.18 | | | | | | -2.69 | |
| 23 | IT-Cas | 12.58 | 0.58 | 21.62 | 5.40 | 0.60 | | 5.59 | 3.71 | 15.31 | 2.24 | 0.42 | 0.12 |
| 24 | JP-BBY | 7.11 | 0.44 | 15.19 | 5.15 | 1.60 | 0.39 | 2.61 | 0.97 | 8.27 | 2.60 | 3.67 | 0.03 |
| 25 | JP-Mse | 13.75 | | 9.50 | | | | 1.59 | | 7.42 | | 0.45 | |
| 26 | JP-SwL | 11.67 | | 66.68 | | | | | | 39.86 | | 18.53 | |
| 27 | KR-CRK | 10.96 | 0.46 | 27.92 | 1.81 | 0.92 | 0.15 | 8.81 | 0.92 | 16.69 | 1.77 | 1.25 | 0.11 |
| 28 | MY-MLM | 27.09 | 0.11 | 9.55 | | 3.28 | | 2.60 | 0.02 | 1.62 | | 2.34 | |
| 29 | NL-Hor | 10.75 | 0.60 | | | | | | | | | | |
| 30 | NZ-Kop | 13.68 | 0.28 | 17.34 | 4.46 | 3.99 | 0.84 | 3.03 | 1.63 | 3.63 | 0.52 | 5.87 | 0.30 |





| | | | | | | | | | | | | | |
|---|---|---|---|---|---|---|---|---|---|---|---|---|---|
| 31 | PH-RiF | 26.54 | 0.15 | 12.41 | | 3.57 | 1.02 | 2.58 | 2.66 | 5.53 | 0.01 | 3.34 | 0.28 |
| 32 | RU-Ch2 | -9.88 | 1.26 | 6.43 | 0.79 | 0.29 | 0.00 | 0.87 | 0.09 | 4.65 | 0.48 | 1.44 | 0.04 |
| 33 | RU-Che | -9.77 | 1.25 | 4.09 | 0.22 | 0.37 | | 0.47 | 0.08 | 2.18 | 0.12 | 1.19 | |
| 34 | RU-Cok | -12.38 | 0.92 | 4.45 | | | | 0.74 | 0.09 | 3.42 | 0.10 | | 1.32 |
| 35 | RU-Fy2 | 5.80 | 0.53 | 3.50 | 1.88 | 1.65 | 0.58 | -0.27 | 0.04 | -0.39 | 0.78 | 2.36 | 0.09 |
| 36 | SE-Deg | 2.57 | 0.77 | 10.74 | 0.88 | 0.59 | 0.55 | 3.30 | 0.29 | 5.70 | 1.66 | 1.44 | 1.87 |
| 37 | UK-LBT | 10.62 | 0.78 | 46.54 | 5.61 | 13.70 | 1.77 | 12.52 | | 12.26 | 0.32 | 14.04 | |
| 38 | US-A03 | -7.15 | 0.66 | 5.81 | 2.06 | | | 1.27 | 0.19 | 3.25 | | | |
| 39 | US-A10 | | | | | | | | | 1.08 | | | |
| 40 | US-Atq | -10.88 | 2.23 | 1.77 | 0.03 | 0.00 | | 0.30 | 0.07 | 1.05 | 0.03 | 0.55 | 0.12 |
| 41 | US-Beo | -9.50 | 0.20 | 2.74 | | 0.09 | | 0.27 | | 1.77 | | 0.69 | |
| 42 | US-Bes | -10.46 | 0.21 | 3.19 | 0.18 | 0.09 | | 0.58 | 0.26 | 2.20 | 0.18 | 0.71 | |
| 43 | US-Bi1 | 13.87 | 1.20 | 0.69 | | 0.45 | 0.43 | -0.07 | 0.02 | -0.13 | | 0.17 | 0.05 |
| 44 | US-Bi2 | 15.01 | 0.28 | 1.28 | 0.59 | 0.66 | | 0.30 | 0.21 | 0.10 | 0.06 | 0.54 | 0.02 |
| 45 | US-BZB | -0.62 | 0.55 | 9.05 | 2.23 | | | 2.41 | 0.39 | 6.06 | 1.43 | | |
| 46 | US-BZF | -0.31 | 0.55 | 8.72 | 2.98 | | | 3.21 | 2.77 | 6.35 | 2.24 | | |
| 47 | US-BZS | 0.26 | 0.68 | 0.78 | 0.15 | 0.58 | | 0.23 | 0.01 | 0.53 | 0.11 | | |
| 48 | US-CRT | 11.32 | 0.91 | 2.21 | 0.00 | 1.53 | | | | 0.26 | | 0.59 | |
| 49 | US-DPW | 22.23 | 0.41 | 48.71 | 8.84 | | 1.84 | 11.27 | 2.75 | 27.64 | 7.55 | 12.90 | 2.54 |
| 50 | US-EDN | 14.99 | | -0.04 | | -0.03 | | -0.19 | | 0.16 | | | |
| 51 | US-EML | -1.72 | 3.76 | 0.59 | 0.39 | -0.04 | 0.27 | 0.06 | 0.17 | 0.35 | 0.12 | 0.27 | |
| 52 | US-Ho1 | 6.48 | 1.32 | -0.16 | 0.09 | | 0.01 | -0.03 | 0.02 | -0.02 | 0.05 | -0.07 | 0.02 |
| 53 | US-HRA | 19.36 | | -0.24 | | | | 1.28 | | 6.08 | | | |
| 54 | US-HRC | 20.23 | | -0.24 | | | | 3.07 | | 8.38 | | | |
| 55 | US-ICs | -6.02 | 0.48 | | | | | | | 1.23 | 0.30 | | |
| 56 | US-Ivo | -8.27 | 0.54 | 4.90 | 0.95 | 0.70 | 0.02 | 0.80 | 0.05 | 2.55 | 0.54 | 1.26 | 0.42 |
| 57 | US-LA1 | 24.12 | 0.42 | 12.68 | | 0.68 | | 2.27 | | 7.58 | | 1.39 | 1.08 |
| 58 | US-LA2 | 20.34 | 4.43 | 34.81 | 19.34 | 4.27 | | 14.50 | 2.18 | 21.72 | 2.75 | 6.96 | 0.79 |
| 59 | US-Los | 5.01 | 1.23 | 6.51 | 1.28 | 0.36 | 0.07 | 1.71 | 0.46 | 3.57 | 0.96 | 0.81 | 0.25 |
| 60 | US-MAC | 23.15 | 0.96 | 15.82 | 10.34 | 1.32 | 0.02 | 3.71 | 2.07 | 14.70 | 5.53 | 2.81 | 0.25 |
| 61 | US-MRM | 13.14 | 0.88 | 0.34 | 0.05 | 0.09 | | 0.07 | 0.01 | 0.11 | 0.01 | | |
| 62 | US-Myb | 15.53 | 0.58 | 47.88 | 14.90 | 4.51 | 1.99 | 14.12 | 5.54 | 22.04 | 7.87 | 6.52 | 3.17 |
| 63 | US-NC4 | 16.74 | 0.85 | 33.89 | 17.41 | 0.80 | 0.18 | 5.70 | 1.62 | 20.41 | 9.80 | 6.77 | 2.19 |
| 64 | US-NGB | -9.45 | 0.92 | 2.41 | 0.15 | | | 0.26 | 0.15 | 2.00 | 0.26 | | |

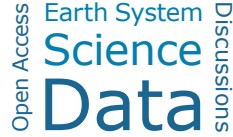

| | SITE_ID | Mean_Air_Temp_C | Mean_Air_Temp_stdev_C | Ann_Flux_g | Ann_Flux_stdev_g | JFM_flux_g | JFM_flux_stdev_g | AMJ_flux_g | AMJ_flux_stdev_g | JAS_flux_g | JAS_flux_stdev_g | OND_flux_g | OND_flux_stdev_g |
|---|---|---|---|---|---|---|---|---|---|---|---|---|---|
| 65 | US-NGC | 1.21 | 0.82 | 2.52 | 0.00 | 0.88 | 0.10 | 2.55 | 0.93 | 0.86 | 0.41 | 0.99 | 0.17 |
| 66 | US-ORv | 12.20 | 0.92 | 7.20 | 2.51 | | | 31.03 | 0.19 | 3.14 | 1.16 | 9.81 | |
| 67 | US-OWC | 13.02 | 1.72 | 113.99 | | 0.15 | 0.05 | 0.39 | 0.74 | 66.03 | 10.07 | | |
| 68 | US-PFa | 5.42 | 1.24 | 0.54 | 0.25 | 2.00 | 1.79 | 1.11 | 0.04 | 0.00 | 0.00 | 1.74 | 1.14 |
| 69 | US-Snd | 14.76 | 1.16 | 4.71 | 1.71 | 2.44 | 0.71 | 14.95 | | 1.35 | 0.61 | 4.71 | 3.54 |
| 70 | US-Sne | 15.04 | 0.45 | 42.80 | 4.48 | 0.09 | 0.06 | 0.31 | 3.74 | 12.61 | 10.68 | 0.08 | 0.07 |
| 71 | US-Srr | 15.93 | 0.45 | 0.83 | 0.08 | | | 1.40 | | 0.42 | 0.09 | 2.92 | |
| 72 | US-StJ | 13.96 | | 9.55 | | 4.92 | 1.91 | 10.13 | 4.83 | 5.24 | | 7.11 | 2.49 |
| 73 | US-Tw1 | 15.16 | 0.74 | 39.51 | 11.03 | | | | | 18.02 | 4.08 | 0.29 | |
| 74 | US-Tw3 | 16.04 | 0.87 | | | 4.39 | 1.75 | 9.49 | 0.59 | | | 5.89 | 1.83 |
| 75 | US-Tw4 | 15.52 | 0.56 | 32.54 | 11.74 | | | 21.49 | | 12.78 | 6.07 | 8.45 | |
| 76 | US-Tw5 | 15.03 | | 59.72 | | 3.07 | 1.10 | 1.05 | 3.99 | 29.78 | | 1.73 | 0.40 |
| 77 | US-Twt | 14.26 | 1.71 | 12.07 | 2.75 | | | | 0.59 | 5.49 | 3.15 | | |
| 78 | US-Uaf | -2.87 | 1.03 | 0.53 | 0.19 | 1.66 | 0.22 | 0.07 | | 0.33 | 0.12 | 3.26 | 0.12 |
| 79 | US-WPT | 11.40 | 0.99 | 49.59 | 7.48 | | | 16.31 | 3.99 | 28.75 | 3.05 | | |

## Column Descriptions

| Column | Description |
|---|---|
| SITE_ID | Site identification code as assigned by regional flux data network |
| Mean_Air_Temp_C | Mean annual air temperature (C) |
| Ann_Flux_g_CH4-C_m-2 | Mean annual methane flux (g CH4-C/m2/year) |
| JFM_flux_g_CH4-C_m-2 | Mean methane flux in January, February, March (gCH4-C/m2/year) |
| AMJ_flux_g_CH4-C_m-2 | Mean methane flux in April, May, June (gCH4-C/m2/year) |
| JAS_flux_g_CH4-C_m-2 | Mean methane flux in July, August, September (gCH4-C/m2/year) |
| OND_flux_g_CH4-C_m-2 | Mean methane flux in October, November, December (gCH4-C/m2/year) |
| Mean_Air_Temp_stdev_C | Standard deviation of annual air temperature (C) |
| Ann_Flux_stdev_g_CH4-C_m-2 | Standard deviation of annual methane flux (gCH4-C/m2/year) |
| JFM_flux_stdev_g_CH4-C_m-2 | Standard deviation of methane flux in January, February, March (gCH4-C/m2/year) |
| AMJ_flux_stdev_g_CH4-C_m-2 | Standard deviation of methane flux in April, May, June (gCH4-C/m2/year) |
| JAS_flux_stdev_g_CH4-C_m-2 | Standard deviation of methane flux in July, August, September (gCH4-C/m2/year) |
| OND_flux_stdev_g_CH4-C_m-2 | Standard deviation of methane flux in October, November, December (gCH4-C/m2/year) |

**Table B2-D: Site metadata, select data, and DOI links**

| | SITE_ID | SOIL_TEMP_PROBE_DEPTHS |
|---|---|---|
| 1 | AT-Neu | TS_1 = -0.05cm; TS_2 = -0.1cm; TS_3 = -0.2cm; |
| 2 | BR-Npw | |
| 3 | BW-Gum | |
| 4 | BW-Nxr | |
| 5 | CA-SCB | TS_1 = 0cm; TS_2 = -0.02cm; TS_3 = -0.04cm; TS_4 = -0.08cm; TS_5 = -0.16cm; TS_6 = -0.32cm; TS_7 = -0.64cm ;TS_8 = -1.28cm; |
| 6 | CA-SCC | TS_1 = -0.1cm; TS_2 = -0.15cm; TS_3 = -0.2cm; TS_4 = -0.25cm; TS_5 = -0.3cm; TS_6 = -0.5cm; TS_7 = -0.6cm; TS_8 = -0.7cm; |
| 7 | CH-Cha | TS_1 = -0.01cm; TS_2 = -0.02cm; TS_3 = -0.04cm; TS_4 = -0.07cm; TS_5 = -0.1cm; TS_6 = -0.15cm; TS_7 = -0.25cm; TS_8 = -0.4cm; TS_9 = -0.95cm; |
| 8 | CH-Dav | TS_1 = -0.05cm; TS_2 = -0.15cm; TS_3 = -0.5cm; |
| 9 | CH-Oe2 | TS_1 = -0.05cm; TS_2 = -0.1cm; TS_3 = -0.15cm; TS_5 = -0.3cm; TS_6 = -0.5cm; |
| 10 | CN-Hgu | |
| 11 | DE-Dgw | |
| 12 | DE-Hte | TS_1 = 0cm; TS_2 = -0.1cm; TS_3 = -0.2cm; |
| 13 | DE-SfN | TS_1 = -0.02cm; TS_3 = -0.1cm; TS_4 = -0.2cm; TS_5 = -0.5cm; |
| 14 | DE-Zrk | TS_1 = -0.05cm; TS_2 = -0.1cm; TS_3 = -0.2cm; TS_4 = -0.3cm; TS_5 = -0.5cm; |
| 15 | FI-Hyy | TS_1 = -0.02cm; TS_2 = -0.04cm; TS_3 = -0.12cm; TS_4 = -0.25cm; TS_5 = -0.5cm; |
| 16 | FI-Lom | TS_1 = -0.07cm; TS_2 = -0.3cm; TS_3 = -0.5cm; |
| 17 | FI-Si2 | TS_1 = -0.05cm; TS_2 = -0.2cm; TS_3 = -0.35cm; TS_4 = -0.5cm;   TS_1 = -0.05cm; TS_2 = -0.2cm; TS_3 = -0.35cm; TS_4 = -0.5cm; |
| 18 | FI-Sii | before 2016(TS_1 = -0.05cm; TS_2 = -0.2cm; TS_3 = -0.35cm; TS_4 = -0.5cm;   TS_2 = -0.2cm; TS_3 = -0.35cm; TS_4 = -0.5cm) after 2017 (TS_1 = 0cm; TS_2 = -0.5cm; TS_3 = -0.1cm; TS_4 = -0.15cm; TS_5 = -0.25cm; TS_6 = -0.45cm; TS_7 = -0.95cm) |
| 19 | FR-LGt | TS_1 = -0.02cm; TS_2 = -0.05cm; TS_3 = -0.1cm; TS_4 = -0.2cm; TS_5 = -0.4cm; |
| 20 | HK-MPM | |
| 21 | ID-Pag | TS_1 = -0.05cm; |
| 22 | IT-BCi | TS_1 = -0.05cm; TS_2 = -0.1cm; TS_3 = -0.3cm; TS_4 = -0.5cm; TS_5 = -1cm; |
| 23 | IT-Cas | TS_1 = -0.05cm; TS_2 = -0.3cm; TS_3 = -0.5cm; |
| 24 | JP-BBY | TS_1 = -0.183cm; TS_2 = -0.233cm; TS_3 = -0.283cm; TS_4 = -0.383cm; TS_5 = -0.483cm; |
| 25 | JP-Mse | TS_1 = -0.01cm; TS_2 = -0.025cm; TS_3 = -0.05cm; TS_4 = -0.1cm; TS_5 = -0.2cm; TS_6 = -0.4cm; |
| 26 | JP-SwL | |
| 27 | KR-CRK | TS_1 = -0.05cm; TS_2 = -0.15cm; |
| 28 | MY-MLM | TS_1 = -0.05cm; |
| 29 | NL-Hor | TS_1 = -0.01cm; TS_2 = -0.02cm; TS_3 = -0.04cm; TS_4 = -0.05cm; TS_5 = -0.1cm; TS_6 = -0.15cm; TS_7 = -0.25cm; TS_8 = -0.4cm; TS_9 = -0.6cm; |
| 30 | NZ-Kop | TS_1 = -0.5cm; TS_2 = -0.1cm; TS_3 = -0.2cm; |
| 31 | PH-RiF | |





| | | |
|---|---|---|
| 32 | RU-Ch2 | TS_1 = -0.04cm; TS_2 = -0.08cm; TS_3 = -0.16cm; |
| 33 | RU-Che | TS_1 = -0.04cm; TS_2 = -0.08cm; TS_3 = -0.16cm; |
| 34 | RU-Cok | |
| 35 | RU-Fy2 | |
| 36 | SE-Deg | TS_1 = -0.02cm; TS_2 = -0.05cm; TS_3 = -0.1cm; TS_4 = -0.15cm; TS_5 = -0.3cm; TS_6 = -0.5cm; |
| 37 | UK-LBT | |
| 38 | US-A03 | TS_1 = -0.025cm; TS_2 = -0.1cm; TS_3 = -0.3cm; |
| 39 | US-A10 | TS_1 = -0.025cm; TS_2 = -0.1cm; TS_3 = -0.3cm; |
| 40 | US-Atq | |
| 41 | US-Beo | |
| 42 | US-Bes | |
| 43 | US-Bi1 | TS_1 = -0.02cm; TS_2 = -0.04cm; TS_3 = -0.08cm; TS_4 = -0.16cm; TS_5 = -0.32cm; |
| 44 | US-Bi2 | TS_1 = -0.02cm; TS_2 = -0.04cm; TS_3 = -0.08cm; TS_4 = -0.16cm; TS_5 = -0.32cm; |
| 45 | US-BZB | TS_1 = -0.075cm; TS_2 = -0.05cm; |
| 46 | US-BZF | TS_1 = -0.075cm; TS_2 = -0.05cm; |
| 47 | US-BZS | |
| 48 | US-CRT | |
| 49 | US-DPW | |
| 50 | US-EDN | TS_1 = -0.25cm; TS_2 = -0.15cm; TS_3 = -0.05cm; TS_4 = 0cm; TS_5 = 0.05cm; TS_6 = 0.1cm; TS_7 = 0.2cm; TS_8 = 0.3cm; |
| 51 | US-EML | TS_1 = -0.05cm; TS_2 = -0.1cm; TS_3 = -0.2cm; TS_4 = -0.4cm; |
| 52 | US-Ho1 | TS_1 = -0.05cm; TS_2 = -0.1cm; |
| 53 | US-HRA | |
| 54 | US-HRC | |
| 55 | US-ICs | TS_1 = -0.075cm; TS_2 = -0.05cm; |
| 56 | US-Ivo | TS_1 = -0.05cm; TS_2 = -0.1cm; TS_3 = -0.15cm; TS_4 = -0.3cm; TS_5 = -0.4cm; |
| 57 | US-LA1 | TS = -0.1cm; |
| 58 | US-LA2 | TS = -0.1cm; |
| 59 | US-Los | TS_1 = 0cm; TS_2 = -0.05cm; TS_3 = -0.1cm; TS_4 = -0.2cm; TS_5 = -0.5cm; |
| 60 | US-MAC | |
| 61 | US-MRM | |
| 62 | US-Myb | TS_1 = -0.02cm; TS_2 = -0.04cm; TS_3 = -0.08cm; TS_4 = -0.16cm; TS_5 = -0.32cm; |
| 63 | US-NC4 | TS_1 = -0.05cm; TS_2 = -0.2cm; |
| 64 | US-NGB | |
| 65 | US-NGC | |





| 66 | US-ORv | TS_1 = -0.08cm; |
| 67 | US-OWC | TS_1 = -0.05cm; TS_2 = -0.3cm; |
| 68 | US-PFa | |
| 69 | US-Snd | TS_1 = -0.08cm; TS_2 = -0.16cm; TS_3 = nancm; TS_4 = nancm; TS_5 = nancm; TS_6 = nancm; |
| 70 | US-Sne | TS_1 = -0.01cm; TS_2 = -0.02cm; TS_3 = -0.08cm; TS_4 = -0.16cm; TS_5 = -0.32cm; |
| 71 | US-Srr | |
| 72 | US-StJ | TS_2 = -0.05cm; TS_3 = -0.1cm; |
| 73 | US-Tw1 | TS_1 = -0.02cm; TS_2 = -0.04cm; TS_3 = -0.08cm; TS_4 = -0.16cm; TS_5 = -0.32cm; |
| 74 | US-Tw3 | TS_1 = -0.02cm; TS_2 = -0.04cm; TS_3 = -0.08cm; TS_4 = -0.16cm; TS_5 = -0.32cm; |
| 75 | US-Tw4 | TS_1 = -0.02cm; TS_2 = -0.04cm; TS_3 = -0.08cm; TS_4 = -0.16cm; TS_5 = -0.32cm; |
| 76 | US-Tw5 | TS_1 = -0.02cm; TS_2 = -0.1cm; TS_3 = -0.02cm; TS_4 = -0.08cm; TS_5 = -0.16cm; |
| 77 | US-Twt | TS_1 = -0.02cm; TS_2 = -0.04cm; TS_3 = -0.08cm; TS_4 = -0.16cm; TS_5 = -0.32cm; |
| 78 | US-Uaf | TS_1 = -0.09cm; TS_2 = -0.183cm; TS_3 = -0.283cm; TS_4 = -0.367cm; TS_5 = -0.5cm; TS_6 = -0.6cm; TS_7 = -0.75cm; TS_8 = -0.925cm; TS_9 = -1cm; |
| 79 | US-WPT | TS_1 = -0.1cm; TS_2 = -0.3cm; |

## Column Descriptions

| SITE_ID | Site identification code as assigned by regional flux data network |
| SOIL_TEMP_PROBE_DEPTHS | Depth of soil temperature probe (m), with negative values being under the surface |



**Table B3:** Bio-climatic data for representativeness analysis.

| Bioclimatic predictor | Source | Units | Original temporal resolution |
|---|---|---|---|
| Latent Heat (LE) | FLUXCOM (Jung et al., 2019) | W m-2 | Monthly 2003-2013 |
| Wong Simple Ratio Water Index (SRWI) | MOD09A1 (Vermote 2015) | Unitless | Monthly ~2001-2018 |
| Enhanced Vegetation Index (EVI) | MOD13A3 (Didan 2015) | Unitless | Monthly 2001-2018 |
| Mean Annual Temperature (MAT) | BioClim (Fick & Hijman 2017) | Degrees Celsius | Monthly 2001-2018 |

**Supplementary References**

Didan, K. *MOD13A3 MODIS/Terra vegetation Indices Monthly L3 Global 1km SIN Grid V006* [Data set]. NASA EOSDIS Land Processes DAAC. 2015.

Fick, S.E. & R.J. Hijmans. WorldClim 2: new 1km spatial resolution climate surfaces for global land areas. International Journal of Climatology, 37(12): 4302-4315. 2017.

Jung, M., Koirala, S., Weber, U., Ichii, K., Gans, F., Camps-Valls, G., Papale, D., Schwalm, C., Tramontana, G., & Reichstein, M. The FLUXCOM ensemble of global land-atmosphere energy fluxes. Scientific Data, 6(74). doi:10.1038/s41597-019-0076-8. 2019.

Vermote. E. MOD09A1 MODIS Surface Reflectance 8-Day L3 Global 500m SIN Grid V006. NASA EOSDIS Land Processes DAAC. http://doi.org/10.5067/MODIS/MOD09A1.006 (Terra). 2015.

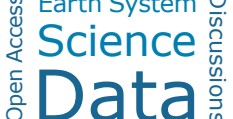

**Table B4-A** Timesat output for FCH4, GPP_DT, TA, and TS (TS from shallowest probe at each site)

| | Site | Year | Start_FCH4_(DOY) | End_FCH4_(DOY) | Base_value_FCH4_(nmolCH4/m2/s) | Ampl_FCH4_(nmolCH4/m2/s) | Peak_FCH4_(DOY) | Peak_value_FCH4_(nmolCH4/m2/s) |
|---|---|---|---|---|---|---|---|---|
| 1 | AT-Neu | 2010 | NaN | NaN | NaN | NaN | NaN | NaN |
| 2 | AT-Neu | 2011 | NaN | NaN | NaN | NaN | NaN | NaN |
| 3 | AT-Neu | 2012 | NaN | NaN | NaN | NaN | NaN | NaN |
| 4 | BR-Npw | 2014 | NaN | NaN | NaN | NaN | NaN | NaN |
| 5 | BR-Npw | 2015 | NaN | NaN | NaN | NaN | NaN | NaN |
| 6 | BR-Npw | 2016 | 192.7 | 345.8 | -2.0 | 154.8 | 270.0 | 152.8 |
| 7 | BW-Gum | 2018 | 34.1 | 151.1 | 132.9 | 186.2 | 89.0 | 319.1 |
| 8 | BW-Gum | 2019 | 230.4 | NaN | 134.2 | 202.1 | 281.9 | 336.3 |
| 9 | BW-Nxr | 2018 | 65.1 | NaN | 29.2 | 208.7 | 287.5 | 237.9 |
| 10 | CA-SCB | 2014 | 138.8 | 299.1 | 17.5 | 72.3 | 222.4 | 89.8 |
| 11 | CA-SCB | 2015 | NaN | NaN | NaN | NaN | NaN | NaN |
| 12 | CA-SCB | 2016 | 109.2 | 290.4 | 11.8 | 82.0 | 207.9 | 93.8 |
| 13 | CA-SCB | 2017 | 119.0 | 300.4 | 14.0 | 58.9 | 221.5 | 72.9 |
| 14 | CA-SCC | 2013 | NaN | NaN | NaN | NaN | 203.4 | 44.8 |
| 15 | CA-SCC | 2014 | 128.4 | 313.1 | 3.1 | 40.1 | 215.0 | 43.3 |
| 16 | CA-SCC | 2015 | 98.0 | 303.9 | 1.7 | 54.6 | 210.9 | 56.3 |
| 17 | CA-SCC | 2016 | 102.7 | NaN | 1.7 | NaN | 208.0 | 59.6 |
| 18 | DE-Dgw | 2015 | NaN | NaN | NaN | NaN | NaN | NaN |
| 19 | DE-Dgw | 2016 | NaN | NaN | NaN | NaN | NaN | NaN |
| 20 | DE-Dgw | 2017 | NaN | NaN | NaN | NaN | NaN | NaN |
| 21 | DE-Hte | 2011 | NaN | NaN | NaN | NaN | NaN | NaN |
| 22 | DE-Hte | 2012 | 82.6 | 330.1 | 20.3 | 201.8 | 205.5 | 222.1 |
| 23 | DE-Hte | 2013 | 101.9 | NaN | 29.9 | NaN | 201.1 | 378.7 |
| 24 | DE-Hte | 2014 | NaN | 338.5 | 38.3 | NaN | 204.8 | 314.1 |
| 25 | DE-Hte | 2015 | 75.1 | 322.4 | 29.2 | 277.7 | 202.0 | 306.9 |
| 26 | DE-Hte | 2016 | 83.9 | 289.7 | 21.5 | 347.8 | 202.0 | 369.3 |
| 27 | DE-Hte | 2017 | 90.0 | 304.5 | 18.3 | 272.7 | 194.0 | 290.9 |
| 28 | DE-Hte | 2018 | 85.6 | 258.1 | 21.0 | 322.0 | 196.0 | 343.0 |





| # | Site | Year | | | | | | |
|---|------|------|------|------|------|------|------|------|
| 29 | DE-SfN | 2012 | 79.6 | 340.7 | 4.3 | 10.2 | 217.0 | 14.5 |
| 30 | DE-SfN | 2013 | NaN | NaN | 2.7 | 3.0 | 301.9 | 5.7 |
| 31 | DE-SfN | 2014 | NaN | NaN | NaN | NaN | NaN | NaN |
| 32 | DE-Zrk | 2013 | NaN | NaN | NaN | NaN | NaN | NaN |
| 33 | DE-Zrk | 2014 | NaN | NaN | NaN | NaN | NaN | NaN |
| 34 | DE-Zrk | 2015 | 87.0 | 273.0 | 9.3 | 242.5 | 208.7 | 251.9 |
| 35 | DE-Zrk | 2016 | 107.9 | 274.0 | 9.9 | 224.2 | 187.3 | 234.2 |
| 36 | DE-Zrk | 2017 | 110.0 | 270.1 | 11.2 | 203.6 | 190.0 | 214.8 |
| 37 | DE-Zrk | 2018 | 88.1 | 261.0 | 8.5 | 250.8 | 196.0 | 259.2 |
| 38 | FI-Lom | 2006 | 142.8 | 288.8 | 7.8 | 111.0 | 215.1 | 118.9 |
| 39 | FI-Lom | 2007 | 139.3 | 270.6 | 10.9 | 165.0 | 214.5 | 175.9 |
| 40 | FI-Lom | 2008 | 134.0 | 284.8 | 10.8 | 127.7 | 211.5 | 138.5 |
| 41 | FI-Lom | 2009 | 121.5 | 291.0 | 12.1 | 132.3 | 215.0 | 144.4 |
| 42 | FI-Lom | 2010 | 137.1 | 282.8 | 13.4 | 101.6 | 214.0 | 115.0 |
| 43 | FI-Si2 | 2012 | NaN | NaN | NaN | NaN | 220.6 | 80.0 |
| 44 | FI-Si2 | 2013 | NaN | NaN | NaN | NaN | 211.1 | 77.4 |
| 45 | FI-Si2 | 2014 | NaN | 280.7 | 7.2 | NaN | 212.8 | 111.1 |
| 46 | FI-Si2 | 2015 | NaN | 309.5 | 9.5 | NaN | 212.0 | 72.2 |
| 47 | FI-Si2 | 2016 | NaN | NaN | NaN | NaN | NaN | NaN |
| 48 | FI-Sii | 2013 | 123.8 | 307.6 | 7.2 | 104.3 | 202.5 | 111.5 |
| 49 | FI-Sii | 2014 | 118.8 | NaN | 2.3 | NaN | 215.1 | 112.7 |
| 50 | FI-Sii | 2015 | NaN | NaN | NaN | NaN | 236.0 | 112.7 |
| 51 | FI-Sii | 2016 | 114.5 | 311.3 | 8.9 | 121.1 | 214.0 | 130.0 |
| 52 | FI-Sii | 2017 | 118.9 | 300.4 | 6.5 | 57.1 | 203.0 | 63.6 |
| 53 | FI-Sii | 2018 | 116.3 | 295.1 | 7.5 | 53.8 | 187.0 | 61.3 |
| 54 | HK-MPM | 2016 | NaN | NaN | NaN | NaN | NaN | NaN |
| 55 | HK-MPM | 2017 | NaN | NaN | NaN | NaN | NaN | NaN |
| 56 | HK-MPM | 2018 | NaN | NaN | NaN | NaN | NaN | NaN |
| 57 | ID-Pag | 2016 | 274.1 | NaN | -2.8 | 5.1 | NaN | 2.3 |
| 58 | JP-BBY | 2015 | 166.7 | NaN | 18.3 | NaN | 237.7 | 71.4 |
| 59 | JP-BBY | 2016 | NaN | 324.9 | 18.3 | 105.7 | 244.3 | 124.0 |
| 60 | JP-BBY | 2017 | 138.5 | 323.1 | 15.2 | 130.1 | 236.0 | 145.3 |
| 61 | JP-BBY | 2018 | NaN | 332.1 | 17.8 | 74.7 | 221.0 | 92.6 |
| 62 | JP-Mse | 2012 | NaN | NaN | NaN | NaN | NaN | NaN |



| # | Site | Year | | | | | | |
|---|------|------|------|------|------|------|------|------|
| 63 | KR-CRK | 2015 | NaN | NaN | NaN | NaN | NaN | NaN |
| 64 | KR-CRK | 2016 | NaN | NaN | NaN | NaN | NaN | NaN |
| 65 | KR-CRK | 2017 | NaN | NaN | NaN | NaN | NaN | NaN |
| 66 | KR-CRK | 2018 | NaN | NaN | NaN | NaN | NaN | NaN |
| 67 | MY-MLM | 2014 | 229.6 | 562.4 | 15.5 | 19.8 | 64.2 | 35.3 |
| 68 | MY-MLM | 2015 | NaN | NaN | NaN | NaN | NaN | NaN |
| 69 | NZ-Kop | 2012 | -94.5 | 227.6 | 36.9 | 28.3 | 176.2 | 65.2 |
| 70 | NZ-Kop | 2013 | 7.2 | 251.5 | 21.1 | 61.7 | 182.0 | 82.8 |
| 71 | NZ-Kop | 2014 | 10.0 | 228.4 | 22.6 | 42.7 | 161.0 | 65.2 |
| 72 | NZ-Kop | 2015 | -8.5 | NaN | 23.0 | 34.7 | 150.0 | 57.8 |
| 73 | PH-RiF | 2012 | 154.2 | 303.9 | 4.0 | 62.9 | 239.1 | 66.9 |
| 74 | PH-RiF | 2013 | 304.1 | 455.0 | 5.3 | 54.0 | 380.3 | 59.3 |
| 75 | PH-RiF | 2014 | 133.9 | 265.7 | 6.1 | 121.8 | 178.3 | 127.9 |
| 76 | PH-RiF | 2015 | NaN | NaN | 3.8 | 56.3 | NaN | 60.1 |
| 77 | RU-Ch2 | 2014 | 150.8 | 312.2 | 0.7 | 70.2 | 216.5 | 70.9 |
| 78 | RU-Ch2 | 2015 | 153.3 | NaN | 8.0 | NaN | 209.0 | 56.1 |
| 79 | RU-Ch2 | 2016 | NaN | NaN | NaN | NaN | 218.8 | 68.3 |
| 80 | RU-Che | 2014 | NaN | NaN | NaN | NaN | NaN | NaN |
| 81 | RU-Che | 2015 | NaN | NaN | NaN | NaN | NaN | NaN |
| 82 | RU-Che | 2016 | NaN | NaN | NaN | NaN | NaN | NaN |
| 83 | SE-Deg | 2014 | NaN | NaN | NaN | 80.8 | 204.2 | 91.7 |
| 84 | SE-Deg | 2015 | 103.3 | 318.7 | 5.1 | 73.7 | 211.3 | 78.8 |
| 85 | SE-Deg | 2016 | 102.5 | 324.1 | 4.3 | 74.3 | 205.3 | 78.7 |
| 86 | SE-Deg | 2017 | NaN | NaN | NaN | NaN | NaN | NaN |
| 87 | SE-Deg | 2018 | 117.2 | 327.6 | 6.9 | 50.9 | 192.0 | 57.8 |
| 88 | US-Atq | 2013 | NaN | NaN | NaN | NaN | NaN | NaN |
| 89 | US-Atq | 2014 | 145.7 | 328.7 | 0.9 | 13.2 | 215.0 | 14.1 |
| 90 | US-Atq | 2015 | 153.3 | 264.0 | 1.0 | 18.6 | 193.2 | 19.6 |
| 91 | US-Beo | 2013 | NaN | NaN | NaN | NaN | NaN | NaN |
| 92 | US-Beo | 2014 | 157.0 | 356.3 | 0.4 | 23.0 | 211.4 | 23.4 |
| 93 | US-Bes | 2013 | NaN | NaN | NaN | NaN | NaN | NaN |
| 94 | US-Bes | 2014 | 157.3 | 312.6 | 0.6 | 34.3 | 206.5 | 34.9 |
| 95 | US-Bes | 2015 | 146.8 | 283.8 | 0.6 | 35.0 | 193.1 | 35.7 |
| 96 | US-BZB | 2014 | NaN | NaN | NaN | NaN | 226.9 | 67.5 |

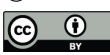



| | Site | Year | | | | | | |
|---|---|---|---|---|---|---|---|---|
| 97 | US-BZB | 2015 | NaN | NaN | NaN | NaN | 219.4 | 68.4 |
| 98 | US-BZB | 2016 | NaN | NaN | NaN | NaN | 226.1 | 98.4 |
| 99 | US-BZF | 2014 | NaN | NaN | NaN | NaN | 231.6 | 57.7 |
| 100 | US-BZF | 2015 | NaN | NaN | NaN | NaN | 179.0 | 87.0 |
| 101 | US-BZF | 2016 | NaN | NaN | NaN | NaN | 220.1 | 119.1 |
| 102 | US-BZS | 2015 | NaN | NaN | NaN | NaN | NaN | NaN |
| 103 | US-BZS | 2016 | NaN | NaN | NaN | NaN | NaN | NaN |
| 104 | US-DPW | 2013 | 151.7 | 364.0 | 16.4 | 395.0 | 240.7 | 411.4 |
| 105 | US-DPW | 2014 | 98.9 | 332.5 | 34.5 | 338.0 | 228.9 | 372.4 |
| 106 | US-DPW | 2015 | NaN | 376.3 | 25.0 | NaN | 248.6 | 247.3 |
| 107 | US-DPW | 2016 | 84.2 | 389.2 | 23.5 | 184.3 | 237.0 | 207.8 |
| 108 | US-HRA | 2017 | NaN | NaN | NaN | NaN | NaN | NaN |
| 109 | US-HRC | 2018 | NaN | NaN | NaN | NaN | NaN | NaN |
| 110 | US-ICs | 2014 | NaN | NaN | NaN | NaN | NaN | NaN |
| 111 | US-ICs | 2015 | NaN | NaN | NaN | NaN | NaN | NaN |
| 112 | US-ICs | 2016 | 138.2 | 302.1 | 0.2 | 18.0 | 200.5 | 18.2 |
| 113 | US-Ivo | 2013 | NaN | 400.0 | 1.9 | 29.9 | 238.9 | 31.9 |
| 114 | US-Ivo | 2014 | 158.5 | 301.8 | 6.7 | 30.0 | 226.8 | 36.7 |
| 115 | US-Ivo | 2015 | 156.8 | 278.0 | 6.9 | 19.4 | 231.1 | 26.3 |
| 116 | US-Ivo | 2016 | 164.7 | 352.4 | 6.1 | 32.5 | 232.0 | 38.7 |
| 117 | US-LA1 | 2012 | NaN | NaN | NaN | NaN | NaN | NaN |
| 118 | US-LA2 | 2012 | 62.8 | NaN | 38.8 | 225.7 | 229.2 | 264.5 |
| 119 | US-LA2 | 2013 | NaN | NaN | 25.1 | 193.2 | 216.2 | 218.3 |
| 120 | US-Los | 2014 | 127.1 | 309.8 | 4.0 | 35.1 | 219.3 | 39.1 |
| 121 | US-Los | 2015 | 143.4 | 324.4 | 3.2 | 34.6 | 220.7 | 37.8 |
| 122 | US-Los | 2016 | 143.8 | 310.1 | 3.3 | 75.8 | 193.6 | 79.1 |
| 123 | US-Los | 2017 | 134.1 | 255.2 | 3.6 | 58.3 | 185.0 | 61.9 |
| 124 | US-Los | 2018 | 143.0 | 288.8 | 3.0 | 52.4 | 191.0 | 55.4 |
| 125 | US-MAC | 2013 | NaN | NaN | NaN | NaN | NaN | NaN |
| 126 | US-MAC | 2014 | NaN | NaN | NaN | NaN | NaN | NaN |
| 127 | US-MAC | 2015 | NaN | NaN | NaN | NaN | NaN | NaN |
| 128 | US-Myb | 2010 | NaN | NaN | NaN | NaN | NaN | NaN |
| 129 | US-Myb | 2011 | 72.4 | 369.3 | 18.3 | 174.2 | 253.5 | 192.5 |
| 130 | US-Myb | 2012 | 97.2 | 345.3 | 18.9 | 366.6 | 214.7 | 385.5 |

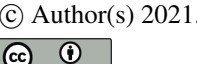



| | | | | | | | | |
|---|---|---|---|---|---|---|---|---|
| 131 | US-Myb | 2013 | 46.8 | 336.3 | 39.0 | 265.4 | 220.2 | 304.3 |
| 132 | US-Myb | 2014 | 57.4 | 334.7 | 37.1 | 276.9 | 206.0 | 314.0 |
| 133 | US-Myb | 2015 | 23.7 | 330.0 | 21.6 | 285.7 | 201.0 | 307.3 |
| 134 | US-Myb | 2016 | 36.9 | 306.0 | 21.9 | 216.0 | 191.0 | 237.9 |
| 135 | US-Myb | 2017 | 175.8 | 332.2 | 30.5 | 191.7 | 235.0 | 222.2 |
| 136 | US-Myb | 2018 | 33.1 | 322.6 | 28.8 | 99.3 | 169.0 | 128.1 |
| 137 | US-NC4 | 2012 | 132.9 | 307.9 | 9.5 | 323.8 | 232.2 | 333.3 |
| 138 | US-NC4 | 2013 | 97.2 | 365.1 | 4.3 | 113.3 | 240.7 | 117.5 |
| 139 | US-NC4 | 2014 | 110.6 | 332.3 | -0.1 | 181.8 | 253.9 | 181.6 |
| 140 | US-NC4 | 2015 | 68.4 | 414.1 | -0.8 | 122.5 | 245.0 | 121.7 |
| 141 | US-NC4 | 2016 | 128.4 | 350.6 | 2.7 | 373.6 | 259.0 | 376.2 |
| 142 | US-ORv | 2011 | NaN | 297.3 | 7.3 | 15.2 | 178.5 | 22.4 |
| 143 | US-ORv | 2012 | 120.1 | 269.1 | 8.3 | 65.2 | 189.7 | 73.4 |
| 144 | US-ORv | 2013 | 72.4 | 308.4 | 9.0 | 27.4 | 189.4 | 36.4 |
| 145 | US-ORv | 2014 | 87.6 | 292.2 | 9.1 | 32.4 | 201.0 | 41.5 |
| 146 | US-ORv | 2015 | 86.0 | NaN | 8.9 | 38.5 | 170.0 | 47.4 |
| 147 | US-OWC | 2015 | NaN | NaN | NaN | NaN | NaN | NaN |
| 148 | US-OWC | 2016 | NaN | NaN | NaN | NaN | 219.2 | 882.8 |
| 149 | US-Sne | 2016 | NaN | NaN | NaN | NaN | NaN | NaN |
| 150 | US-Sne | 2017 | 76.2 | 337.1 | 14.9 | 244.2 | 187.8 | 259.0 |
| 151 | US-Sne | 2018 | 60.6 | 341.6 | 21.4 | 168.3 | 222.6 | 189.8 |
| 152 | US-Srr | 2014 | NaN | NaN | NaN | NaN | NaN | NaN |
| 153 | US-Srr | 2015 | NaN | NaN | NaN | NaN | NaN | NaN |
| 154 | US-Srr | 2016 | NaN | NaN | NaN | NaN | NaN | NaN |
| 155 | US-Srr | 2017 | NaN | NaN | NaN | NaN | NaN | NaN |
| 156 | US-StJ | 2016 | NaN | NaN | NaN | NaN | NaN | NaN |
| 157 | US-Tw1 | 2011 | 140.5 | 352.4 | 36.3 | 104.5 | 233.8 | 140.8 |
| 158 | US-Tw1 | 2012 | NaN | 309.6 | 28.1 | 243.5 | 242.9 | 271.6 |
| 159 | US-Tw1 | 2013 | 33.4 | 307.1 | 42.2 | 114.3 | 218.0 | 156.5 |
| 160 | US-Tw1 | 2014 | 174.6 | 331.5 | 65.3 | 253.2 | 240.0 | 318.5 |
| 161 | US-Tw1 | 2015 | 62.3 | 330.3 | 63.8 | 204.1 | 207.0 | 267.9 |
| 162 | US-Tw1 | 2016 | 32.3 | 323.8 | 48.5 | 160.0 | 221.0 | 208.5 |
| 163 | US-Tw1 | 2017 | 27.8 | 305.0 | 43.1 | 138.1 | 226.0 | 181.1 |
| 164 | US-Tw1 | 2018 | 155.0 | 314.9 | 38.7 | 127.5 | 228.0 | 166.3 |





|     | Site   | Year |       |       |      |       |       |       |
| --- | ------ | ---- | ----- | ----- | ---- | ----- | ----- | ----- |
| 165 | US-Tw4 | 2014 | 93.8  | 461.3 | 27.4 | 36.5  | 226.8 | 63.8  |
| 166 | US-Tw4 | 2015 | 114.5 | 334.1 | 39.8 | 86.5  | 228.2 | 126.3 |
| 167 | US-Tw4 | 2016 | 42.8  | 357.1 | 43.2 | 101.8 | 215.6 | 144.9 |
| 168 | US-Tw4 | 2017 | 110.7 | 318.8 | 55.1 | 201.2 | 222.0 | 256.3 |
| 169 | US-Tw4 | 2018 | 63.0  | 237.3 | 53.0 | 165.1 | 173.0 | 218.1 |
| 170 | US-Tw5 | 2018 | NaN   | 331.9 | 26.5 | 339.3 | 196.9 | 365.8 |
| 171 | US-Twt | 2009 | NaN   | NaN   | NaN  | NaN   | NaN   | NaN   |
| 172 | US-Twt | 2010 | NaN   | NaN   | NaN  | NaN   | NaN   | NaN   |
| 173 | US-Twt | 2011 | NaN   | NaN   | NaN  | NaN   | NaN   | NaN   |
| 174 | US-Twt | 2012 | NaN   | NaN   | NaN  | NaN   | NaN   | NaN   |
| 175 | US-Twt | 2013 | NaN   | NaN   | NaN  | NaN   | NaN   | NaN   |
| 176 | US-Twt | 2014 | NaN   | NaN   | NaN  | NaN   | NaN   | NaN   |
| 177 | US-Twt | 2015 | NaN   | NaN   | NaN  | NaN   | NaN   | NaN   |
| 178 | US-Twt | 2016 | NaN   | NaN   | NaN  | NaN   | NaN   | NaN   |
| 179 | US-Uaf | 2011 | 157.6 | NaN   | 0.8  | 2.1   | 242.0 | 2.8   |
| 180 | US-Uaf | 2012 | 151.8 | NaN   | 0.7  | 1.6   | 265.9 | 2.3   |
| 181 | US-Uaf | 2013 | 167.0 | NaN   | 0.8  | 1.4   | 267.0 | 2.2   |
| 182 | US-Uaf | 2014 | 182.2 | NaN   | 0.9  | 3.2   | 247.0 | 4.1   |
| 183 | US-Uaf | 2015 | 176.0 | NaN   | 0.8  | 3.5   | 245.0 | 4.3   |
| 184 | US-Uaf | 2016 | 184.7 | NaN   | 0.9  | 7.3   | 248.0 | 8.2   |
| 185 | US-Uaf | 2017 | 182.0 | NaN   | 0.9  | 6.0   | 248.0 | 6.8   |
| 186 | US-Uaf | 2018 | 158.5 | NaN   | 0.9  | 4.9   | 250.0 | 5.8   |
| 187 | US-WPT | 2011 | 103.5 | 294.1 | 5.6  | 355.3 | 207.1 | 360.9 |
| 188 | US-WPT | 2012 | 90.0  | 296.5 | 9.0  | 380.5 | 195.6 | 389.5 |
| 189 | US-WPT | 2013 | 72.5  | 297.0 | 7.5  | 343.3 | 220.0 | 350.8 |

**Column Descriptions**

| Column | Description |
| --- | --- |
| Site | Site name |
| Year | Data year |
| Start_FCH4_(DOY) | Season start for elevated methane fluxes (DOY), point "f" in Figure 1 |
| End_FCH4_(DOY) | Season end for elevated methane fluxes (DOY), point "h" in Figure 1 |





| Base_value_FCH4_(nmolCH4/m2/s) | Baseline methane flux during non-elevated season (nmol CH4 /m2/ s), average of points "a" and "b" in Figure 1 |
| Ampl_FCH4_(nmolCH4/m2/s) | Amplitude of methane flux during elevated flux season (nmol CH4/m2/s), difference between point "e" in Figure 1 and Base_value_FCH4 |
| Peak_FCH4_(DOY) | Day of maximum elevated methane flux (DOY), point "g" in Figure 1 |
| Peak_value_FCH4_(nmolCH4/m2/s) | Maximum value of methane flux (nmol CH4/m2/s), point "e" in Figure 1 |

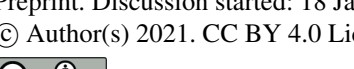



**Table B4-B Timesat output for FCH4, GPP_DT, TA, and TS (TS from shallowest probe at each site)**

| | Site | Year | Start_GPP_D T_(DOY) | End_GPP_ DT_(DOY) | Base_value_G PP_DT_(µmol CO2/m2/s) | Ampl_GPP_D T_(µmolCO2/ m2/s) | Peak_GPP_D T_(DOY) | Peak_value_G PP_DT_(µmol CO2/s) |
|---|---|---|---|---|---|---|---|---|
| 1 | AT-Neu | 2010 | 61.39 | 332.24 | -0.22 | 9.75 | 175.90 | 9.53 |
| 2 | AT-Neu | 2011 | 76.71 | 303.74 | 0.18 | 11.36 | 167.90 | 11.54 |
| 3 | AT-Neu | 2012 | 84.67 | 305.48 | 0.29 | 9.75 | 179.00 | 10.04 |
| 4 | BR-Npw | 2014 | 59.47 | 367.83 | 2.03 | 5.10 | 242.90 | 7.13 |
| 5 | BR-Npw | 2015 | 61.54 | 385.15 | 2.05 | 5.15 | 228.00 | 7.20 |
| 6 | BR-Npw | 2016 | 83.78 | 375.61 | 2.44 | 4.74 | 203.00 | 7.19 |
| 7 | BW-Gum | 2018 | NaN | NaN | NaN | NaN | NaN | NaN |
| 8 | BW-Gum | 2019 | NaN | NaN | NaN | NaN | NaN | NaN |
| 9 | BW-Nxr | 2018 | NaN | NaN | NaN | NaN | NaN | NaN |
| 10 | CA-SCB | 2014 | 127.53 | 266.07 | 0.10 | 2.69 | 210.00 | 2.79 |
| 11 | CA-SCB | 2015 | 60.15 | 275.15 | 0.10 | 3.35 | 199.40 | 3.45 |
| 12 | CA-SCB | 2016 | 123.04 | 277.07 | 0.05 | 3.66 | 191.90 | 3.71 |
| 13 | CA-SCB | 2017 | 113.78 | 274.63 | 0.04 | 2.99 | 202.00 | 3.02 |
| 14 | CA-SCC | 2013 | 126.43 | 273.72 | 0.20 | 3.21 | 198.80 | 3.41 |
| 15 | CA-SCC | 2014 | 130.06 | 269.31 | 0.30 | 3.22 | 194.30 | 3.52 |
| 16 | CA-SCC | 2015 | 104.41 | 269.97 | 0.28 | 4.54 | 196.90 | 4.82 |
| 17 | CA-SCC | 2016 | 106.89 | 284.75 | 0.09 | 3.67 | 191.00 | 3.76 |
| 18 | DE-Dgw | 2015 | 13.43 | 348.62 | 0.04 | 0.40 | 227.30 | 0.44 |
| 19 | DE-Dgw | 2016 | 31.35 | 294.27 | 0.04 | 0.49 | 167.60 | 0.52 |
| 20 | DE-Dgw | 2017 | 80.56 | 293.50 | 0.04 | 0.46 | 191.00 | 0.50 |
| 21 | DE-Hte | 2011 | 111.70 | 280.46 | 0.05 | 6.87 | 170.10 | 6.92 |
| 22 | DE-Hte | 2012 | 122.64 | 296.52 | 0.31 | 7.11 | 200.10 | 7.42 |
| 23 | DE-Hte | 2013 | 133.51 | 293.97 | 0.29 | 6.10 | 206.40 | 6.39 |
| 24 | DE-Hte | 2014 | 37.51 | 277.70 | 0.27 | 5.50 | 160.00 | 5.77 |
| 25 | DE-Hte | 2015 | 127.82 | 303.30 | 0.27 | 5.35 | 191.00 | 5.61 |
| 26 | DE-Hte | 2016 | 117.96 | 328.49 | 0.16 | 5.39 | 184.00 | 5.55 |
| 27 | DE-Hte | 2017 | 123.83 | 301.59 | 0.11 | 5.71 | 177.00 | 5.82 |
| 28 | DE-Hte | 2018 | 121.03 | 334.06 | 0.06 | 6.99 | 190.00 | 7.05 |





| # | | Year | | | | | | |
|---|--------|------|--------|--------|-------|------|--------|-------|
| 29 | DE-SfN | 2012 | -13.79 | 320.94 | 0.37 | 4.34 | 168.10 | 4.71 |
| 30 | DE-SfN | 2013 | 64.35 | 316.65 | 0.31 | 3.96 | 198.00 | 4.27 |
| 31 | DE-SfN | 2014 | 43.97 | 335.84 | 0.41 | 4.10 | 193.00 | 4.51 |
| 32 | DE-Zrk | 2013 | 110.23 | 283.50 | 0.09 | 4.30 | 186.30 | 4.39 |
| 33 | DE-Zrk | 2014 | 86.61 | 309.76 | 0.06 | 3.56 | 180.40 | 3.62 |
| 34 | DE-Zrk | 2015 | 99.44 | 264.19 | 0.10 | 3.54 | 186.60 | 3.64 |
| 35 | DE-Zrk | 2016 | 90.31 | 301.90 | 0.13 | 4.38 | 218.00 | 4.51 |
| 36 | DE-Zrk | 2017 | 92.93 | 303.86 | 0.11 | 4.07 | 180.00 | 4.18 |
| 37 | DE-Zrk | 2018 | 105.21 | 314.58 | 0.06 | 6.90 | 212.00 | 6.96 |
| 38 | FI-Lom | 2006 | 147.75 | 261.40 | 0.06 | 5.89 | 197.40 | 5.95 |
| 39 | FI-Lom | 2007 | 145.82 | 257.55 | 0.06 | 6.36 | 197.90 | 6.43 |
| 40 | FI-Lom | 2008 | 151.61 | 258.87 | 0.06 | 7.24 | 200.00 | 7.30 |
| 41 | FI-Lom | 2009 | 147.57 | 262.55 | 0.02 | 6.55 | 197.00 | 6.57 |
| 42 | FI-Lom | 2010 | 153.94 | 262.64 | 0.03 | 6.39 | 199.00 | 6.41 |
| 43 | FI-Si2 | 2012 | 33.66 | 276.85 | 0.07 | 1.52 | 209.00 | 1.59 |
| 44 | FI-Si2 | 2013 | 106.78 | 338.07 | 0.13 | 1.63 | 182.90 | 1.76 |
| 45 | FI-Si2 | 2014 | 40.93 | 290.00 | 0.13 | 2.19 | 146.00 | 2.32 |
| 46 | FI-Si2 | 2015 | 113.24 | 267.77 | 0.13 | 1.93 | 197.00 | 2.06 |
| 47 | FI-Si2 | 2016 | 43.90 | 284.85 | 0.13 | 1.85 | 166.00 | 1.98 |
| 48 | FI-Sii | 2013 | 118.98 | 282.46 | -0.03 | 3.70 | 185.70 | 3.67 |
| 49 | FI-Sii | 2014 | 100.30 | 294.41 | 0.00 | 2.43 | 199.70 | 2.44 |
| 50 | FI-Sii | 2015 | 84.64 | 321.89 | 0.06 | 2.63 | 204.30 | 2.69 |
| 51 | FI-Sii | 2016 | 118.70 | 284.09 | 0.09 | 3.43 | 200.00 | 3.52 |
| 52 | FI-Sii | 2017 | 117.46 | 290.52 | 0.05 | 3.04 | 206.00 | 3.09 |
| 53 | FI-Sii | 2018 | 113.63 | 295.59 | 0.04 | 2.32 | 185.00 | 2.36 |
| 54 | HK-MPM | 2016 | NaN | NaN | NaN | NaN | NaN | NaN |
| 55 | HK-MPM | 2017 | NaN | NaN | NaN | NaN | NaN | NaN |
| 56 | HK-MPM | 2018 | NaN | NaN | NaN | NaN | NaN | NaN |
| 57 | ID-Pag | 2016 | NaN | NaN | NaN | NaN | NaN | NaN |
| 58 | JP-BBY | 2015 | 123.64 | 304.32 | 0.23 | 5.32 | 203.40 | 5.56 |
| 59 | JP-BBY | 2016 | 114.13 | 302.43 | 0.03 | 7.94 | 203.20 | 7.98 |
| 60 | JP-BBY | 2017 | 119.90 | 300.38 | 0.03 | 7.58 | 199.80 | 7.61 |
| 61 | JP-BBY | 2018 | 96.26 | 311.58 | 0.01 | 5.44 | 217.00 | 5.45 |
| 62 | JP-Mse | 2012 | 144.63 | 266.88 | 0.63 | 9.81 | 209.70 | 10.44 |





| | | | | | | | | |
|---|---|---|---|---|---|---|---|---|
| 63 | KR-CRK | 2015 | 134.99 | 267.83 | 0.10 | 10.68 | 202.10 | 10.78 |
| 64 | KR-CRK | 2016 | 137.21 | 262.37 | 0.06 | 12.44 | 198.80 | 12.50 |
| 65 | KR-CRK | 2017 | 143.28 | 266.25 | 0.13 | 12.20 | 193.50 | 12.33 |
| 66 | KR-CRK | 2018 | 138.97 | 263.80 | 0.17 | 10.96 | 198.00 | 11.13 |
| 67 | MY-MLM | 2014 | 179.97 | 437.86 | 8.48 | 2.67 | 272.60 | 11.16 |
| 68 | MY-MLM | 2015 | 194.24 | NaN | 8.76 | 8.31 | 271.10 | 17.07 |
| 69 | NZ-Kop | 2012 | 38.67 | 334.88 | 1.33 | 2.50 | 194.80 | 3.83 |
| 70 | NZ-Kop | 2013 | 58.12 | 351.65 | 1.50 | 2.61 | 190.00 | 4.10 |
| 71 | NZ-Kop | 2014 | 42.25 | 355.00 | 1.41 | 2.78 | 209.00 | 4.18 |
| 72 | NZ-Kop | 2015 | 44.32 | 366.21 | 1.19 | 3.28 | 193.00 | 4.47 |
| 73 | PH-RiF | 2012 | NaN | NaN | NaN | NaN | NaN | NaN |
| 74 | PH-RiF | 2013 | NaN | NaN | NaN | NaN | NaN | NaN |
| 75 | PH-RiF | 2014 | NaN | NaN | NaN | NaN | NaN | NaN |
| 76 | PH-RiF | 2015 | NaN | NaN | NaN | NaN | NaN | NaN |
| 77 | RU-Ch2 | 2014 | 142.91 | 252.79 | 0.02 | 5.10 | 210.30 | 5.11 |
| 78 | RU-Ch2 | 2015 | 157.47 | 247.95 | -0.02 | 5.00 | 202.60 | 4.97 |
| 79 | RU-Ch2 | 2016 | 145.27 | 257.90 | -0.04 | 4.15 | 201.50 | 4.11 |
| 80 | RU-Che | 2014 | 161.74 | 258.77 | 0.14 | 5.51 | 206.90 | 5.65 |
| 81 | RU-Che | 2015 | 157.04 | 250.25 | 0.01 | 5.39 | 203.30 | 5.40 |
| 82 | RU-Che | 2016 | 140.55 | 258.30 | -0.10 | 6.94 | 188.60 | 6.84 |
| 83 | SE-Deg | 2014 | 115.38 | 285.94 | 0.02 | 2.78 | 196.30 | 2.79 |
| 84 | SE-Deg | 2015 | 113.50 | 278.71 | 0.02 | 2.69 | 203.40 | 2.70 |
| 85 | SE-Deg | 2016 | 118.80 | 290.27 | 0.02 | 2.32 | 195.50 | 2.35 |
| 86 | SE-Deg | 2017 | 121.86 | 276.14 | 0.02 | 2.45 | 199.00 | 2.47 |
| 87 | SE-Deg | 2018 | 118.54 | 276.63 | 0.00 | 1.72 | 188.00 | 1.72 |
| 88 | US-Atq | 2013 | 33.24 | 256.46 | 0.03 | 3.05 | 161.70 | 3.09 |
| 89 | US-Atq | 2014 | 139.11 | 244.88 | 0.09 | 1.77 | 194.30 | 1.86 |
| 90 | US-Atq | 2015 | 132.75 | 243.97 | 0.08 | 3.41 | 191.00 | 3.48 |
| 91 | US-Beo | 2013 | 39.33 | 285.28 | 0.01 | 0.88 | 159.80 | 0.88 |
| 92 | US-Beo | 2014 | 88.23 | 261.54 | 0.02 | 1.99 | 200.00 | 2.02 |
| 93 | US-Bes | 2013 | 49.45 | 269.44 | 0.04 | 0.84 | 187.90 | 0.87 |
| 94 | US-Bes | 2014 | 174.64 | 262.25 | 0.04 | 1.60 | 220.70 | 1.64 |
| 95 | US-Bes | 2015 | 160.53 | 248.82 | 0.04 | 2.53 | 198.40 | 2.57 |
| 96 | US-BZB | 2014 | NaN | NaN | NaN | NaN | NaN | NaN |

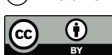



|  |  |  |  |  |  |  |  |  |
|---|---|---|---|---|---|---|---|---|
| 97 | US-BZB | 2015 | NaN | NaN | NaN | NaN | NaN | NaN |
| 98 | US-BZB | 2016 | NaN | NaN | NaN | NaN | NaN | NaN |
| 99 | US-BZF | 2014 | 132.71 | 201.27 | 0.18 | 5.84 | 167.50 | 6.01 |
| 100 | US-BZF | 2015 | 129.12 | 258.65 | 0.16 | 6.93 | 187.50 | 7.10 |
| 101 | US-BZF | 2016 | 128.63 | 227.99 | 0.18 | 9.14 | 175.00 | 9.32 |
| 102 | US-BZS | 2015 | NaN | NaN | NaN | NaN | NaN | NaN |
| 103 | US-BZS | 2016 | NaN | NaN | NaN | NaN | NaN | NaN |
| 104 | US-DPW | 2013 | NaN | NaN | NaN | NaN | NaN | NaN |
| 105 | US-DPW | 2014 | 55.72 | 332.08 | 0.65 | 4.60 | 183.50 | 5.24 |
| 106 | US-DPW | 2015 | 53.69 | 372.93 | 0.73 | 4.77 | 174.80 | 5.50 |
| 107 | US-DPW | 2016 | 71.21 | 343.87 | 0.85 | 4.83 | 197.00 | 5.68 |
| 108 | US-HRA | 2017 | 131.27 | 244.82 | 0.39 | 20.14 | 183.10 | 20.53 |
| 109 | US-HRC | 2018 | 135.69 | 237.65 | 1.34 | 18.79 | 182.20 | 20.13 |
| 110 | US-ICs | 2014 | 150.35 | 253.40 | 0.21 | 3.46 | 201.60 | 3.66 |
| 111 | US-ICs | 2015 | 142.83 | 263.16 | 0.15 | 4.42 | 189.90 | 4.57 |
| 112 | US-ICs | 2016 | 154.39 | 245.71 | 0.12 | 3.22 | 192.80 | 3.34 |
| 113 | US-Ivo | 2013 | 149.10 | 257.98 | 0.06 | 3.85 | 201.90 | 3.91 |
| 114 | US-Ivo | 2014 | 151.63 | 257.85 | 0.11 | 3.63 | 199.10 | 3.73 |
| 115 | US-Ivo | 2015 | 121.66 | 248.35 | 0.07 | 3.97 | 187.10 | 4.03 |
| 116 | US-Ivo | 2016 | 154.70 | 254.51 | 0.09 | 5.30 | 194.00 | 5.39 |
| 117 | US-LA1 | 2012 | -7.96 | 216.91 | 0.49 | 2.29 | 142.80 | 2.78 |
| 118 | US-LA2 | 2012 | 46.97 | 334.04 | 0.34 | 5.64 | 146.80 | 5.98 |
| 119 | US-LA2 | 2013 | 93.73 | 335.84 | 0.31 | 6.94 | 166.30 | 7.25 |
| 120 | US-Los | 2014 | 131.45 | 288.06 | -0.01 | 6.65 | 198.20 | 6.64 |
| 121 | US-Los | 2015 | 130.36 | 288.12 | 0.11 | 6.26 | 201.50 | 6.37 |
| 122 | US-Los | 2016 | 130.87 | 291.48 | 0.17 | 7.17 | 198.20 | 7.33 |
| 123 | US-Los | 2017 | 136.00 | 292.41 | 0.13 | 7.38 | 199.00 | 7.51 |
| 124 | US-Los | 2018 | 134.97 | 285.61 | 0.13 | 7.17 | 199.00 | 7.29 |
| 125 | US-MAC | 2013 | NaN | 378.12 | 2.50 | 5.43 | 222.70 | 9.45 |
| 126 | US-MAC | 2014 | 47.02 | 334.19 | 2.51 | 9.17 | 180.70 | 11.68 |
| 127 | US-MAC | 2015 | 46.53 | 356.65 | 2.92 | 5.92 | 154.10 | 8.84 |
| 128 | US-Myb | 2010 | 28.65 | 305.24 | 0.56 | 1.41 | 183.40 | 1.97 |
| 129 | US-Myb | 2011 | 200.41 | 367.27 | 0.28 | 3.62 | 265.50 | 3.89 |
| 130 | US-Myb | 2012 | 88.45 | 331.93 | -0.06 | 13.68 | 204.50 | 13.62 |



| | | | | | | | |
|---|---|---|---|---|---|---|---|
| **131** | US-Myb | 2013 | 47.39 | 341.89 | -0.12 | 7.95 | 200.00 | 7.83 |
| **132** | US-Myb | 2014 | 86.80 | 310.73 | 0.16 | 8.20 | 168.00 | 8.36 |
| **133** | US-Myb | 2015 | 76.10 | 323.19 | 0.17 | 7.31 | 202.00 | 7.47 |
| **134** | US-Myb | 2016 | 24.11 | 328.43 | 0.00 | 4.03 | 176.00 | 4.03 |
| **135** | US-Myb | 2017 | 67.46 | 395.16 | -0.24 | 5.79 | 202.00 | 5.55 |
| **136** | US-Myb | 2018 | 83.66 | 331.31 | -0.25 | 10.50 | 201.00 | 10.25 |
| **137** | US-NC4 | 2012 | NaN | NaN | NaN | NaN | NaN | NaN |
| **138** | US-NC4 | 2013 | 94.35 | 304.81 | 0.78 | 6.97 | 181.00 | 7.75 |
| **139** | US-NC4 | 2014 | 97.51 | 354.46 | 0.64 | 6.09 | 173.70 | 6.73 |
| **140** | US-NC4 | 2015 | 92.03 | 315.08 | 0.74 | 9.73 | 185.00 | 10.47 |
| **141** | US-NC4 | 2016 | 99.57 | 326.98 | 0.92 | 8.22 | 183.00 | 9.14 |
| **142** | US-ORv | 2011 | 88.77 | 316.45 | -0.06 | 7.39 | 180.40 | 7.33 |
| **143** | US-ORv | 2012 | 93.50 | 303.39 | 0.27 | 9.29 | 181.00 | 9.55 |
| **144** | US-ORv | 2013 | 107.62 | 305.23 | 0.36 | 10.11 | 192.40 | 10.46 |
| **145** | US-ORv | 2014 | 109.31 | 299.17 | 0.24 | 10.14 | 190.00 | 10.38 |
| **146** | US-ORv | 2015 | 105.01 | 301.56 | 0.27 | 9.64 | 194.00 | 9.91 |
| **147** | US-OWC | 2015 | NaN | 301.35 | 0.26 | 6.72 | 151.30 | 6.98 |
| **148** | US-OWC | 2016 | 116.05 | 309.20 | 0.30 | 6.44 | 204.00 | 6.73 |
| **149** | US-Sne | 2016 | -21.79 | 306.16 | 0.34 | 7.99 | 190.30 | 8.34 |
| **150** | US-Sne | 2017 | NaN | NaN | NaN | NaN | NaN | NaN |
| **151** | US-Sne | 2018 | 84.63 | 370.31 | 0.32 | 2.43 | 202.00 | 2.75 |
| **152** | US-Srr | 2014 | 47.02 | 307.53 | 0.78 | 6.04 | 175.60 | 6.83 |
| **153** | US-Srr | 2015 | 35.50 | 320.88 | 0.33 | 7.96 | 158.60 | 8.29 |
| **154** | US-Srr | 2016 | 44.76 | 318.87 | 0.38 | 8.86 | 170.80 | 9.24 |
| **155** | US-Srr | 2017 | 56.75 | 309.79 | 0.30 | 10.46 | 185.00 | 10.76 |
| **156** | US-StJ | 2016 | 120.72 | 280.75 | 1.30 | 12.01 | 193.80 | 13.31 |
| **157** | US-Tw1 | 2011 | NaN | NaN | NaN | NaN | NaN | NaN |
| **158** | US-Tw1 | 2012 | 102.12 | 325.54 | 0.00 | 12.83 | 216.10 | 12.83 |
| **159** | US-Tw1 | 2013 | 98.35 | 338.02 | -0.18 | 13.11 | 208.40 | 12.93 |
| **160** | US-Tw1 | 2014 | 95.66 | 326.27 | 0.12 | 10.46 | 208.00 | 10.58 |
| **161** | US-Tw1 | 2015 | 105.53 | 344.13 | 0.26 | 9.88 | 215.00 | 10.13 |
| **162** | US-Tw1 | 2016 | 91.82 | 313.13 | -0.01 | 10.10 | 209.00 | 10.09 |
| **163** | US-Tw1 | 2017 | 93.36 | 329.76 | -0.04 | 11.26 | 214.00 | 11.22 |
| **164** | US-Tw1 | 2018 | 119.04 | 363.78 | -0.02 | 12.73 | 217.00 | 12.70 |



| | | | | | | | |
|---|---|---|---|---|---|---|---|
| **165** | US-Tw4 | 2014 | 160.04 | 363.23 | 0.00 | 4.70 | 236.60 | 4.70 |
| **166** | US-Tw4 | 2015 | 57.22 | 335.89 | 0.01 | 8.11 | 213.00 | 8.13 |
| **167** | US-Tw4 | 2016 | 76.15 | 311.33 | 0.17 | 8.22 | 185.00 | 8.39 |
| **168** | US-Tw4 | 2017 | 100.19 | 332.90 | 0.14 | 8.76 | 214.00 | 8.90 |
| **169** | US-Tw4 | 2018 | 98.39 | 337.78 | 0.04 | 11.84 | 206.00 | 11.88 |
| **170** | US-Tw5 | 2018 | 115.94 | 321.33 | 1.77 | 6.68 | 231.10 | 8.45 |
| **171** | US-Twt | 2009 | 149.98 | 293.01 | 0.20 | 12.46 | 212.00 | 12.66 |
| **172** | US-Twt | 2010 | 141.10 | 311.91 | 0.10 | 13.71 | 224.40 | 13.81 |
| **173** | US-Twt | 2011 | 158.51 | 288.69 | 0.12 | 14.22 | 215.90 | 14.34 |
| **174** | US-Twt | 2012 | 166.84 | 308.78 | 0.21 | 12.31 | 233.00 | 12.52 |
| **175** | US-Twt | 2013 | 138.24 | 272.38 | 0.27 | 16.71 | 202.00 | 16.98 |
| **176** | US-Twt | 2014 | 148.14 | 281.40 | 0.16 | 15.01 | 205.00 | 15.17 |
| **177** | US-Twt | 2015 | 137.11 | 277.23 | 0.17 | 11.52 | 218.00 | 11.68 |
| **178** | US-Twt | 2016 | 169.14 | 289.90 | 0.29 | 13.80 | 224.00 | 14.08 |
| **179** | US-Uaf | 2011 | 114.56 | 283.49 | 0.12 | 6.04 | 196.90 | 6.17 |
| **180** | US-Uaf | 2012 | 88.03 | 271.24 | 0.18 | 6.58 | 187.30 | 6.76 |
| **181** | US-Uaf | 2013 | 124.13 | 271.61 | 0.22 | 5.79 | 192.10 | 6.01 |
| **182** | US-Uaf | 2014 | 84.63 | 269.30 | 0.14 | 5.32 | 188.00 | 5.46 |
| **183** | US-Uaf | 2015 | 90.52 | 264.29 | 0.09 | 5.17 | 196.00 | 5.25 |
| **184** | US-Uaf | 2016 | 103.06 | 270.75 | 0.14 | 4.66 | 192.00 | 4.80 |
| **185** | US-Uaf | 2017 | 102.29 | 275.58 | 0.14 | 6.10 | 198.00 | 6.25 |
| **186** | US-Uaf | 2018 | 111.65 | 291.46 | 0.04 | 5.60 | 190.00 | 5.64 |
| **187** | US-WPT | 2011 | 134.98 | 285.59 | 0.24 | 7.43 | 206.10 | 7.67 |
| **188** | US-WPT | 2012 | 129.04 | 293.67 | 0.06 | 6.97 | 205.90 | 7.03 |
| **189** | US-WPT | 2013 | 134.87 | 278.12 | 0.05 | 6.20 | 200.90 | 6.24 |

**Column Descriptions**

| | |
|---|---|
| Site | Site name |
| Year | Data year |
| Start_GPP_DT_(DOY) | Season start for elevated GPP_DT (DOY), point "f" in Figure 1 |
| End_GPP_DT_(DOY) | Season end for elevated GPP_DT fluxes (DOY), point "h" in Figure 1 |



| | |
|---|---|
| Base_value_GPP_DT_(µmolCO2/m2/s) | Baseline GPP_DT flux during non-elevated season (µmol CO2/m2/s), average of points "a" and "b" in Figure 1 |
| Ampl_GPP_DT_(µmolCO2/m2/s) | Amplitude of GPP_DT flux during elevated flux season (µmol CO2/m2/s), difference between point "e" in Figure 1 and Base_value_GPP_DT |
| Peak_GPP_DT_(DOY) | Day of maximum elevated GPP_DT flux (DOY), point "g" in Figure 1 |
| Peak_value_GPP_DT_(µmolCO2/m2/s) | Maximum value of GPP_DT flux (µmol CO2/m2/s), point "e" in Figure 1 |



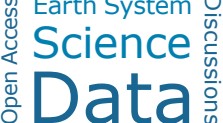

**Table B4-C Timesat output for FCH4, GPP_DT, TA, and TS (TS from shallowest probe at each site)**

| | Site | Year | Start_TA_ (DOY) | End_TA_ (DOY) | Base_value_ TA_(C) | Ampl_TA_ (C) | Peak_TA_ (DOY) | Peak_value_ TA_(C) |
|---|---|---|---|---|---|---|---|---|
| 1 | AT-Neu | 2010 | 43.17 | 351.66 | -4.84 | 20.94 | 195.90 | 16.10 |
| 2 | AT-Neu | 2011 | 18.47 | 359.91 | -5.08 | 20.55 | 198.50 | 15.47 |
| 3 | AT-Neu | 2012 | 38.03 | 366.62 | -5.57 | 22.35 | 197.20 | 16.78 |
| 4 | BR-Npw | 2014 | NaN | NaN | NaN | NaN | NaN | NaN |
| 5 | BR-Npw | 2015 | NaN | NaN | NaN | NaN | NaN | NaN |
| 6 | BR-Npw | 2016 | 7.49 | 348.67 | 19.56 | 7.53 | 211.00 | 27.10 |
| 7 | BW-Gum | 2018 | NaN | NaN | NaN | NaN | NaN | NaN |
| 8 | BW-Gum | 2019 | NaN | NaN | NaN | NaN | NaN | NaN |
| 9 | BW-Nxr | 2018 | NaN | NaN | NaN | NaN | NaN | NaN |
| 10 | CA-SCB | 2014 | 60.23 | 335.31 | -23.33 | 41.29 | 197.40 | 17.96 |
| 11 | CA-SCB | 2015 | 45.11 | 360.83 | -21.75 | 39.11 | 186.40 | 17.37 |
| 12 | CA-SCB | 2016 | 49.77 | 335.79 | -18.88 | 37.28 | 193.90 | 18.40 |
| 13 | CA-SCB | 2017 | 64.68 | 327.30 | -18.45 | 35.95 | 201.00 | 17.50 |
| 14 | CA-SCC | 2013 | 67.57 | 338.23 | -21.08 | 39.13 | 203.80 | 18.05 |
| 15 | CA-SCC | 2014 | 54.14 | 337.83 | -22.27 | 40.98 | 196.70 | 18.72 |
| 16 | CA-SCC | 2015 | 46.41 | 359.38 | -20.09 | 37.86 | 187.10 | 17.77 |
| 17 | CA-SCC | 2016 | 47.31 | 350.23 | -18.76 | 37.65 | 194.00 | 18.89 |
| 18 | DE-Dgw | 2015 | 72.47 | 347.56 | 2.18 | 16.25 | 203.80 | 18.42 |
| 19 | DE-Dgw | 2016 | 64.46 | 324.69 | 1.34 | 17.37 | 205.40 | 18.71 |
| 20 | DE-Dgw | 2017 | 43.79 | 375.32 | -0.17 | 18.47 | 202.20 | 18.30 |
| 21 | DE-Hte | 2011 | NaN | NaN | NaN | NaN | NaN | NaN |
| 22 | DE-Hte | 2012 | 50.34 | 352.49 | 0.77 | 17.03 | 207.60 | 17.80 |
| 23 | DE-Hte | 2013 | 83.53 | 365.30 | 1.64 | 17.09 | 202.50 | 18.73 |
| 24 | DE-Hte | 2014 | 48.71 | 352.47 | 2.86 | 15.82 | 213.00 | 18.68 |
| 25 | DE-Hte | 2015 | 57.62 | 366.35 | 2.59 | 15.33 | 211.00 | 17.92 |
| 26 | DE-Hte | 2016 | 67.53 | 323.10 | 2.75 | 16.01 | 211.00 | 18.76 |
| 27 | DE-Hte | 2017 | 58.94 | 370.07 | 1.81 | 16.22 | 212.00 | 18.02 |
| 28 | DE-Hte | 2018 | 74.55 | 368.02 | 0.91 | 19.14 | 203.00 | 20.05 |
| 29 | DE-SfN | 2012 | 54.86 | 355.18 | -2.09 | 20.25 | 196.30 | 18.16 |
| 30 | DE-SfN | 2013 | 64.64 | 344.04 | -0.35 | 18.57 | 202.80 | 18.22 |



 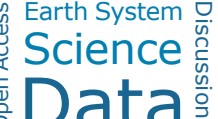

| | | | | | | | | |
|---|---|---|---|---|---|---|---|---|
| 31 | DE-SfN | 2014 | NaN | NaN | NaN | NaN | NaN | NaN |
| 32 | DE-Zrk | 2013 | NaN | NaN | NaN | NaN | NaN | NaN |
| 33 | DE-Zrk | 2014 | NaN | NaN | NaN | NaN | NaN | NaN |
| 34 | DE-Zrk | 2015 | 49.71 | 355.39 | 2.13 | 15.33 | 203.40 | 17.46 |
| 35 | DE-Zrk | 2016 | 63.30 | 323.76 | 1.85 | 16.45 | 204.00 | 18.31 |
| 36 | DE-Zrk | 2017 | 45.25 | 372.48 | 0.47 | 17.44 | 205.00 | 17.91 |
| 37 | DE-Zrk | 2018 | 75.77 | 358.90 | -0.10 | 18.78 | 190.00 | 18.68 |
| 38 | FI-Lom | 2006 | 72.49 | 355.63 | -12.19 | 26.67 | 195.00 | 14.48 |
| 39 | FI-Lom | 2007 | 56.24 | 372.50 | -11.01 | 22.60 | 193.40 | 11.59 |
| 40 | FI-Lom | 2008 | 73.45 | 348.05 | -10.36 | 22.30 | 200.40 | 11.95 |
| 41 | FI-Lom | 2009 | 56.72 | 356.29 | -13.61 | 27.09 | 205.00 | 13.48 |
| 42 | FI-Lom | 2010 | 54.81 | 346.00 | -16.10 | 27.81 | 200.00 | 11.71 |
| 43 | FI-Si2 | 2012 | 70.38 | 349.74 | -6.91 | 22.72 | 199.10 | 15.81 |
| 44 | FI-Si2 | 2013 | 88.15 | 369.48 | -5.99 | 23.74 | 188.60 | 17.76 |
| 45 | FI-Si2 | 2014 | 53.29 | 350.42 | -4.39 | 21.50 | 200.30 | 17.11 |
| 46 | FI-Si2 | 2015 | 39.64 | 359.19 | -5.00 | 19.39 | 212.00 | 14.39 |
| 47 | FI-Si2 | 2016 | 42.62 | 345.02 | -6.72 | 23.17 | 199.00 | 16.45 |
| 48 | FI-Sii | 2013 | 89.28 | 363.91 | -5.74 | 22.24 | 189.10 | 16.50 |
| 49 | FI-Sii | 2014 | 52.05 | 339.85 | -4.36 | 21.52 | 200.90 | 17.15 |
| 50 | FI-Sii | 2015 | 34.40 | 357.72 | -5.42 | 19.66 | 209.50 | 14.25 |
| 51 | FI-Sii | 2016 | 55.00 | 320.65 | -5.24 | 20.90 | 195.00 | 15.66 |
| 52 | FI-Sii | 2017 | 61.62 | 377.27 | -6.32 | 20.93 | 208.00 | 14.61 |
| 53 | FI-Sii | 2018 | 74.22 | 378.66 | -8.98 | 27.37 | 195.00 | 18.39 |
| 54 | HK-MPM | 2016 | 53.59 | 378.86 | 15.97 | 13.27 | 198.10 | 29.24 |
| 55 | HK-MPM | 2017 | 49.13 | 357.32 | 16.27 | 13.07 | 215.60 | 29.34 |
| 56 | HK-MPM | 2018 | 42.11 | 383.02 | 15.25 | 14.05 | 199.00 | 29.30 |
| 57 | ID-Pag | 2016 | NaN | NaN | NaN | NaN | NaN | NaN |
| 58 | JP-BBY | 2015 | 56.18 | 355.66 | -5.30 | 25.89 | 209.50 | 20.58 |
| 59 | JP-BBY | 2016 | 46.06 | 348.04 | -7.13 | 28.01 | 212.00 | 20.89 |
| 60 | JP-BBY | 2017 | 44.65 | 361.65 | -7.47 | 28.20 | 208.70 | 20.73 |
| 61 | JP-BBY | 2018 | 45.63 | 367.94 | -7.55 | 27.12 | 210.00 | 19.57 |
| 62 | JP-Mse | 2012 | 54.48 | 351.81 | 1.48 | 24.59 | 219.90 | 26.08 |
| 63 | KR-CRK | 2015 | NaN | NaN | NaN | NaN | NaN | NaN |
| 64 | KR-CRK | 2016 | NaN | NaN | NaN | NaN | NaN | NaN |





| | | | | | | | | |
|---|---|---|---|---|---|---|---|---|
| 65 | KR-CRK | 2017 | NaN | NaN | NaN | NaN | NaN | NaN |
| 66 | KR-CRK | 2018 | NaN | NaN | NaN | NaN | NaN | NaN |
| 67 | MY-MLM | 2014 | 17.51 | 365.80 | 25.97 | 1.95 | 179.40 | 27.91 |
| 68 | MY-MLM | 2015 | NaN | NaN | NaN | NaN | NaN | NaN |
| 69 | NZ-Kop | 2012 | 50.84 | 347.05 | 9.12 | 9.72 | 219.90 | 18.84 |
| 70 | NZ-Kop | 2013 | 39.31 | 352.93 | 9.31 | 8.28 | 209.60 | 17.60 |
| 71 | NZ-Kop | 2014 | 53.45 | 352.71 | 9.08 | 9.68 | 215.00 | 18.76 |
| 72 | NZ-Kop | 2015 | 51.01 | 357.77 | 8.55 | 10.41 | 212.00 | 18.96 |
| 73 | PH-RiF | 2012 | NaN | NaN | NaN | NaN | NaN | NaN |
| 74 | PH-RiF | 2013 | NaN | NaN | NaN | NaN | NaN | NaN |
| 75 | PH-RiF | 2014 | NaN | NaN | NaN | NaN | NaN | NaN |
| 76 | PH-RiF | 2015 | NaN | NaN | NaN | NaN | NaN | NaN |
| 77 | RU-Ch2 | 2014 | 62.07 | 339.91 | -31.58 | 46.02 | 208.40 | 14.44 |
| 78 | RU-Ch2 | 2015 | 54.09 | 340.02 | -34.37 | 47.59 | 204.40 | 13.23 |
| 79 | RU-Ch2 | 2016 | 56.44 | 373.19 | -34.38 | 48.39 | 201.90 | 14.00 |
| 80 | RU-Che | 2014 | 61.35 | 339.96 | -31.54 | 45.92 | 208.00 | 14.37 |
| 81 | RU-Che | 2015 | 53.19 | 340.10 | -34.28 | 47.55 | 204.30 | 13.26 |
| 82 | RU-Che | 2016 | 55.88 | 372.30 | -34.29 | 48.37 | 201.70 | 14.08 |
| 83 | SE-Deg | 2014 | 69.57 | 327.67 | -5.24 | 21.42 | 201.00 | 16.18 |
| 84 | SE-Deg | 2015 | 29.69 | 352.18 | -7.38 | 19.31 | 213.90 | 11.93 |
| 85 | SE-Deg | 2016 | 54.69 | 331.26 | -7.34 | 20.99 | 197.10 | 13.65 |
| 86 | SE-Deg | 2017 | 61.87 | 353.17 | -8.35 | 21.32 | 210.00 | 12.98 |
| 87 | SE-Deg | 2018 | 74.36 | 373.70 | -11.34 | 26.53 | 193.00 | 15.18 |
| 88 | US-Atq | 2013 | NaN | NaN | NaN | NaN | NaN | NaN |
| 89 | US-Atq | 2014 | 70.05 | 360.75 | -25.81 | 32.20 | 203.40 | 6.38 |
| 90 | US-Atq | 2015 | 84.07 | 347.24 | -26.07 | 34.70 | 196.70 | 8.63 |
| 91 | US-Beo | 2013 | NaN | NaN | NaN | NaN | NaN | NaN |
| 92 | US-Beo | 2014 | 63.26 | 351.27 | -22.70 | 25.61 | 203.20 | 2.91 |
| 93 | US-Bes | 2013 | NaN | NaN | NaN | NaN | NaN | NaN |
| 94 | US-Bes | 2014 | 76.62 | 357.34 | -24.74 | 26.57 | 208.30 | 1.83 |
| 95 | US-Bes | 2015 | 82.66 | 344.05 | -25.16 | 29.49 | 203.30 | 4.32 |
| 96 | US-BZB | 2014 | 65.05 | 339.25 | -17.12 | 32.08 | 189.60 | 14.95 |
| 97 | US-BZB | 2015 | 52.17 | 340.04 | -17.27 | 33.86 | 181.60 | 16.58 |
| 98 | US-BZB | 2016 | 35.70 | 321.04 | -16.88 | 33.52 | 192.50 | 16.64 |



| | | | | | | | | |
|---|---|---|---|---|---|---|---|---|
| 99 | US-BZF | 2014 | 64.65 | 341.50 | -16.78 | 31.98 | 190.50 | 15.20 |
| 100 | US-BZF | 2015 | 52.19 | 340.98 | -16.93 | 33.41 | 181.50 | 16.48 |
| 101 | US-BZF | 2016 | 34.60 | 321.31 | -16.57 | 33.21 | 194.00 | 16.63 |
| 102 | US-BZS | 2015 | 46.24 | 343.61 | -17.41 | 34.59 | 180.20 | 17.18 |
| 103 | US-BZS | 2016 | 32.55 | 320.26 | -15.91 | 33.02 | 193.80 | 17.11 |
| 104 | US-DPW | 2013 | 59.45 | 361.72 | 16.30 | 10.86 | 220.80 | 27.16 |
| 105 | US-DPW | 2014 | 38.66 | 326.75 | 15.99 | 11.79 | 210.30 | 27.78 |
| 106 | US-DPW | 2015 | 33.62 | 381.23 | 15.93 | 9.98 | 211.80 | 25.91 |
| 107 | US-DPW | 2016 | 45.53 | 367.09 | 15.72 | 12.81 | 203.00 | 28.53 |
| 108 | US-HRA | 2017 | NaN | NaN | NaN | NaN | NaN | NaN |
| 109 | US-HRC | 2018 | NaN | NaN | NaN | NaN | NaN | NaN |
| 110 | US-ICs | 2014 | NaN | NaN | NaN | NaN | NaN | NaN |
| 111 | US-ICs | 2015 | NaN | NaN | NaN | NaN | NaN | NaN |
| 112 | US-ICs | 2016 | 68.13 | 328.76 | -16.90 | 26.34 | 196.80 | 9.44 |
| 113 | US-Ivo | 2013 | 93.13 | 360.42 | -23.36 | 36.20 | 193.60 | 12.84 |
| 114 | US-Ivo | 2014 | 69.00 | 341.89 | -21.44 | 30.70 | 193.50 | 9.26 |
| 115 | US-Ivo | 2015 | 90.83 | 326.77 | -21.52 | 31.94 | 188.60 | 10.42 |
| 116 | US-Ivo | 2016 | 90.00 | 339.98 | -21.61 | 31.33 | 197.00 | 9.72 |
| 117 | US-LA1 | 2012 | 31.05 | 302.54 | 19.07 | 8.85 | 177.60 | 27.92 |
| 118 | US-LA2 | 2012 | 35.18 | 316.87 | 16.70 | 11.55 | 197.40 | 28.25 |
| 119 | US-LA2 | 2013 | 71.41 | 321.17 | 15.91 | 13.32 | 210.60 | 29.23 |
| 120 | US-Los | 2014 | 58.01 | 365.46 | -14.20 | 33.10 | 195.40 | 18.89 |
| 121 | US-Los | 2015 | 50.36 | 367.57 | -10.76 | 29.37 | 203.50 | 18.60 |
| 122 | US-Los | 2016 | 39.16 | 356.37 | -9.11 | 29.07 | 209.40 | 19.96 |
| 123 | US-Los | 2017 | 30.73 | 345.36 | -9.80 | 28.09 | 212.00 | 18.29 |
| 124 | US-Los | 2018 | 69.61 | 336.03 | -9.75 | 29.76 | 197.00 | 20.02 |
| 125 | US-MAC | 2013 | NaN | NaN | NaN | NaN | NaN | NaN |
| 126 | US-MAC | 2014 | 42.39 | 323.96 | 17.10 | 9.77 | 211.20 | 26.87 |
| 127 | US-MAC | 2015 | 39.48 | 328.01 | 16.94 | 9.62 | 200.60 | 26.56 |
| 128 | US-Myb | 2010 | NaN | NaN | NaN | NaN | NaN | NaN |
| 129 | US-Myb | 2011 | 31.10 | 331.80 | 8.15 | 12.63 | 223.60 | 20.78 |
| 130 | US-Myb | 2012 | 16.54 | 358.52 | 7.18 | 13.47 | 214.90 | 20.66 |
| 131 | US-Myb | 2013 | 24.44 | 342.29 | 7.26 | 13.44 | 197.00 | 20.70 |
| 132 | US-Myb | 2014 | 11.52 | 353.52 | 8.78 | 12.76 | 211.00 | 21.54 |



| | | | | | | | | |
|---|---|---|---|---|---|---|---|---|
| 133 | US-Myb | 2015 | -0.97 | 325.62 | 8.88 | 13.12 | 228.00 | 21.99 |
| 134 | US-Myb | 2016 | -3.34 | 351.73 | 7.96 | 12.97 | 208.00 | 20.93 |
| 135 | US-Myb | 2017 | 27.35 | 345.82 | 8.30 | 13.88 | 214.00 | 22.19 |
| 136 | US-Myb | 2018 | 44.40 | 332.57 | 9.03 | 11.46 | 218.00 | 20.49 |
| 137 | US-NC4 | 2012 | 57.32 | 339.28 | 9.01 | 17.49 | 208.70 | 26.50 |
| 138 | US-NC4 | 2013 | 69.27 | 352.37 | 6.79 | 19.01 | 204.60 | 25.79 |
| 139 | US-NC4 | 2014 | 49.17 | 367.68 | 5.41 | 18.30 | 206.30 | 23.71 |
| 140 | US-NC4 | 2015 | 64.43 | 392.18 | 6.92 | 19.42 | 202.00 | 26.33 |
| 141 | US-NC4 | 2016 | 53.97 | 350.61 | 8.14 | 19.07 | 215.00 | 27.21 |
| 142 | US-ORv | 2011 | 63.96 | 358.28 | 0.50 | 25.07 | 195.20 | 25.57 |
| 143 | US-ORv | 2012 | 32.88 | 351.31 | 0.84 | 24.92 | 195.90 | 25.76 |
| 144 | US-ORv | 2013 | 51.21 | 355.78 | -2.44 | 25.94 | 203.60 | 23.50 |
| 145 | US-ORv | 2014 | 51.30 | 370.00 | -4.35 | 27.96 | 196.00 | 23.61 |
| 146 | US-ORv | 2015 | 62.52 | 393.56 | -4.58 | 27.87 | 199.00 | 23.29 |
| 147 | US-OWC | 2015 | NaN | NaN | NaN | NaN | NaN | NaN |
| 148 | US-OWC | 2016 | 56.19 | 365.00 | 2.20 | 22.65 | 211.80 | 24.85 |
| 149 | US-Sne | 2016 | NaN | NaN | NaN | NaN | NaN | NaN |
| 150 | US-Sne | 2017 | 9.23 | 344.35 | 7.29 | 14.52 | 217.90 | 21.80 |
| 151 | US-Sne | 2018 | 49.63 | 357.43 | 7.14 | 13.97 | 220.90 | 21.11 |
| 152 | US-Srr | 2014 | 50.56 | 337.45 | 10.74 | 10.06 | 217.30 | 20.80 |
| 153 | US-Srr | 2015 | 4.98 | 323.35 | 9.49 | 12.20 | 235.30 | 21.69 |
| 154 | US-Srr | 2016 | -6.23 | 346.21 | 8.26 | 11.70 | 211.90 | 19.96 |
| 155 | US-Srr | 2017 | 17.23 | 346.39 | 7.94 | 13.42 | 216.00 | 21.36 |
| 156 | US-StJ | 2016 | 67.33 | 347.07 | 3.01 | 23.37 | 214.00 | 26.37 |
| 157 | US-Tw1 | 2011 | 59.53 | 328.03 | 7.82 | 13.46 | 222.10 | 21.28 |
| 158 | US-Tw1 | 2012 | 18.52 | 356.80 | 6.67 | 14.88 | 216.30 | 21.56 |
| 159 | US-Tw1 | 2013 | 29.27 | 344.79 | 7.09 | 14.34 | 196.80 | 21.43 |
| 160 | US-Tw1 | 2014 | 14.74 | 349.77 | 8.52 | 13.61 | 204.00 | 22.13 |
| 161 | US-Tw1 | 2015 | 7.05 | 324.07 | 8.19 | 13.45 | 222.00 | 21.63 |
| 162 | US-Tw1 | 2016 | 4.40 | 350.56 | 7.22 | 13.89 | 203.00 | 21.11 |
| 163 | US-Tw1 | 2017 | 25.14 | 343.29 | 7.24 | 15.12 | 206.00 | 22.36 |
| 164 | US-Tw1 | 2018 | 50.84 | 337.80 | 7.43 | 13.37 | 211.00 | 20.80 |
| 165 | US-Tw4 | 2014 | 15.67 | 348.81 | 8.65 | 13.44 | 206.80 | 22.09 |
| 166 | US-Tw4 | 2015 | 5.29 | 324.60 | 8.49 | 13.42 | 224.80 | 21.90 |



| | Site | Year | Start_TA_(DOY) | End_TA_(DOY) | Base_value_TA_(C) | Ampl_TA_(C) | Peak_TA_(DOY) | Peak_value_TA_(C) |
|---|---|---|---|---|---|---|---|---|
| 167 | US-Tw4 | 2016 | 2.66 | 347.62 | 7.58 | 14.06 | 201.00 | 21.64 |
| 168 | US-Tw4 | 2017 | 30.79 | 337.78 | 7.88 | 15.11 | 208.00 | 22.99 |
| 169 | US-Tw4 | 2018 | 44.48 | 331.60 | 8.26 | 13.08 | 213.00 | 21.33 |
| 170 | US-Tw5 | 2018 | 76.28 | 338.55 | 9.15 | 12.61 | 208.50 | 21.76 |
| 171 | US-Twt | 2009 | NaN | NaN | NaN | NaN | NaN | NaN |
| 172 | US-Twt | 2010 | NaN | NaN | NaN | NaN | NaN | NaN |
| 173 | US-Twt | 2011 | NaN | NaN | NaN | NaN | NaN | NaN |
| 174 | US-Twt | 2012 | NaN | NaN | NaN | NaN | NaN | NaN |
| 175 | US-Twt | 2013 | NaN | NaN | NaN | NaN | NaN | NaN |
| 176 | US-Twt | 2014 | NaN | NaN | NaN | NaN | NaN | NaN |
| 177 | US-Twt | 2015 | NaN | NaN | NaN | NaN | NaN | NaN |
| 178 | US-Twt | 2016 | NaN | NaN | NaN | NaN | NaN | NaN |
| 179 | US-Uaf | 2011 | 59.01 | 330.80 | -23.42 | 38.45 | 191.90 | 15.02 |
| 180 | US-Uaf | 2012 | 43.13 | 317.75 | -23.94 | 38.57 | 192.30 | 14.63 |
| 181 | US-Uaf | 2013 | 64.43 | 344.15 | -22.12 | 39.63 | 195.90 | 17.51 |
| 182 | US-Uaf | 2014 | 49.85 | 342.99 | -20.55 | 34.19 | 190.00 | 13.65 |
| 183 | US-Uaf | 2015 | 51.31 | 346.65 | -19.52 | 34.90 | 182.00 | 15.38 |
| 184 | US-Uaf | 2016 | 27.33 | 325.47 | -20.83 | 36.13 | 193.00 | 15.31 |
| 185 | US-Uaf | 2017 | 59.18 | 357.64 | -22.10 | 38.30 | 191.00 | 16.20 |
| 186 | US-Uaf | 2018 | 35.38 | 354.57 | -21.58 | 36.74 | 196.00 | 15.16 |
| 187 | US-WPT | 2011 | 62.29 | 362.57 | -1.13 | 26.31 | 199.10 | 25.18 |
| 188 | US-WPT | 2012 | 34.86 | 355.23 | -0.44 | 25.53 | 198.40 | 25.09 |
| 189 | US-WPT | 2013 | 64.19 | 341.07 | -1.92 | 24.49 | 205.00 | 22.57 |

**Column Description**

| | |
|---|---|
| Site | Site name |
| Year | Data year |
| Start_TA_(DOY) | Season start for elevated TA (DOY), point "f" in Figure 1 |
| End_TA_(DOY) | Season end for elevated TA (DOY), point "h" in Figure 1 |
| Base_value_TA_(C) | Baseline TA during non-elevated season (C), average of points "a" and "b" in Figure 1 |
| Ampl_TA_(C) | Amplitude of TAduring elevated temperature season (C), difference between point "e" in Figure 1 and Base_value_TA |
| Peak_TA_(DOY) | Day of maximum elevated TA (DOY), point "g" in Figure 1 |
| Peak_value_TA_(C) | Maximum value of TA (C) point "e" in Figure 1 |

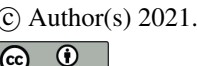



Earth System Science Data Discussions — Open Access

**Table B4-D Timesat output for FCH4, GPP_DT, TA, and TS (TS from shallowest probe at each site)**

| | Site | Year | Probe_name | Soil_temp_depth_m | Start_TS_(DOY) | End_TS_(DOY) | Base_value_TS_(C) | Ampl_TS_(C) | Peak_TS_(DOY) | Peak_value_TS_(C) |
|---|---|---|---|---|---|---|---|---|---|---|
| 1 | AT-Neu | 2010 | TS_1 | -0.05 | 61.32 | 339.44 | 0.15 | 17.54 | 200.90 | 17.70 |
| 2 | AT-Neu | 2011 | TS_1 | -0.05 | 51.04 | 328.84 | 0.40 | 16.37 | 201.00 | 16.77 |
| 3 | AT-Neu | 2012 | TS_1 | -0.05 | 61.12 | 341.88 | 0.73 | 17.57 | 202.90 | 18.30 |
| 4 | BR-Npw | 2014 | NaN | NaN | NaN | NaN | NaN | NaN | NaN | NaN |
| 5 | BR-Npw | 2015 | NaN | NaN | NaN | NaN | NaN | NaN | NaN | NaN |
| 6 | BR-Npw | 2016 | TS_1 | NaN | 18.41 | 343.22 | 22.41 | 5.98 | 188.00 | 28.40 |
| 7 | BW-Gum | 2018 | NaN | NaN | NaN | NaN | NaN | NaN | NaN | NaN |
| 8 | BW-Gum | 2019 | NaN | NaN | NaN | NaN | NaN | NaN | NaN | NaN |
| 9 | BW-Nxr | 2018 | NaN | NaN | NaN | NaN | NaN | NaN | NaN | NaN |
| 10 | CA-SCB | 2014 | TS_1 | 0.00 | 105.94 | 292.18 | -0.63 | 20.62 | 196.60 | 19.99 |
| 11 | CA-SCB | 2015 | TS_1 | 0.00 | 106.92 | 287.14 | -0.39 | 17.23 | 186.80 | 16.84 |
| 12 | CA-SCB | 2016 | TS_1 | 0.00 | 101.64 | 284.09 | -0.31 | 19.07 | 193.20 | 18.77 |
| 13 | CA-SCB | 2017 | TS_1 | 0.00 | 107.44 | 289.72 | -0.25 | 17.67 | 198.00 | 17.42 |
| 14 | CA-SCC | 2013 | NaN | NaN | NaN | NaN | NaN | NaN | NaN | NaN |
| 15 | CA-SCC | 2014 | TS_1 | -0.10 | 123.36 | 287.06 | -0.55 | 15.64 | 203.00 | 15.09 |
| 16 | CA-SCC | 2015 | TS_1 | -0.10 | 113.88 | 285.22 | -0.28 | 16.26 | 189.20 | 15.98 |
| 17 | CA-SCC | 2016 | TS_1 | -0.10 | 108.09 | 260.12 | -0.33 | 18.37 | 190.90 | 18.04 |
| 18 | DE-Dgw | 2015 | NaN | NaN | NaN | NaN | NaN | NaN | NaN | NaN |
| 19 | DE-Dgw | 2016 | NaN | NaN | NaN | NaN | NaN | NaN | NaN | NaN |
| 20 | DE-Dgw | 2017 | NaN | NaN | NaN | NaN | NaN | NaN | NaN | NaN |
| 21 | DE-Hte | 2011 | NaN | NaN | NaN | NaN | NaN | NaN | NaN | NaN |
| 22 | DE-Hte | 2012 | TS_3 | -0.20 | 76.96 | 344.01 | 4.98 | 12.26 | 215.50 | 17.23 |
| 23 | DE-Hte | 2013 | TS_3 | -0.20 | 60.92 | 377.99 | 3.96 | 11.99 | 207.90 | 15.95 |
| 24 | DE-Hte | 2014 | TS_1 | 0.00 | NaN | 327.83 | 8.52 | 8.34 | 205.60 | 16.87 |
| 25 | DE-Hte | 2015 | TS_1 | 0.00 | 62.92 | 360.55 | 5.17 | 11.67 | 187.40 | 16.84 |
| 26 | DE-Hte | 2016 | TS_1 | 0.00 | 71.87 | NaN | 4.94 | 12.36 | 175.60 | 17.30 |
| 27 | DE-Hte | 2017 | TS_1 | 0.00 | 61.55 | 343.32 | 4.50 | 11.76 | 186.00 | 16.26 |
| 28 | DE-Hte | 2018 | NaN | NaN | NaN | NaN | NaN | NaN | NaN | NaN |
| 29 | DE-SfN | 2012 | TS_1 | -0.02 | NaN | 372.59 | 0.00 | 23.55 | 206.50 | 15.29 |
| 30 | DE-SfN | 2013 | TS_1 | -0.02 | 55.84 | 381.50 | 0.92 | 13.62 | 216.40 | 14.54 |

| | | | | | | | | | | |
|---|---|---|---|---|---|---|---|---|---|---|
| 31 | DE-SfN | 2014 | NaN | NaN | NaN | NaN | NaN | NaN | NaN | NaN |
| 32 | DE-Zrk | 2013 | NaN | NaN | NaN | NaN | NaN | NaN | NaN | NaN |
| 33 | DE-Zrk | 2014 | TS_1 | -0.05 | 54.79 | 361.65 | 4.36 | 13.93 | 202.30 | 18.29 |
| 34 | DE-Zrk | 2015 | TS_1 | -0.05 | 58.29 | 359.29 | 4.28 | 13.24 | 215.50 | 17.52 |
| 35 | DE-Zrk | 2016 | TS_1 | -0.05 | 72.81 | 332.00 | 4.28 | 14.93 | 200.40 | 19.20 |
| 36 | DE-Zrk | 2017 | TS_1 | -0.05 | 69.10 | 351.44 | 4.40 | 14.72 | 199.00 | 19.12 |
| 37 | DE-Zrk | 2018 | TS_1 | -0.05 | 84.47 | 336.14 | 4.83 | 12.32 | 203.00 | 17.14 |
| 38 | FI-Lom | 2006 | TS_1 | -0.07 | 114.15 | 290.83 | -0.11 | 13.42 | 204.80 | 13.31 |
| 39 | FI-Lom | 2007 | TS_1 | -0.07 | 126.84 | 302.05 | 0.11 | 13.05 | 200.00 | 13.16 |
| 40 | FI-Lom | 2008 | TS_1 | -0.07 | 135.62 | 296.74 | 0.16 | 12.73 | 202.90 | 12.88 |
| 41 | FI-Lom | 2009 | TS_1 | -0.07 | 117.16 | 291.86 | 0.14 | 11.73 | 214.00 | 11.87 |
| 42 | FI-Lom | 2010 | TS_1 | -0.07 | 129.91 | 318.54 | 0.05 | 12.13 | 208.00 | 12.18 |
| 43 | FI-Si2 | 2012 | TS_1 | -0.05 | NaN | 323.46 | 0.00 | 19.85 | 204.60 | 16.02 |
| 44 | FI-Si2 | 2013 | TS_1 | -0.05 | 106.90 | 341.05 | -0.05 | 16.04 | 199.40 | 15.98 |
| 45 | FI-Si2 | 2014 | TS_1 | -0.05 | 104.63 | 331.10 | -0.04 | 17.07 | 208.50 | 17.03 |
| 46 | FI-Si2 | 2015 | TS_1 | -0.05 | 76.49 | 352.34 | -0.87 | 16.31 | 211.00 | 15.44 |
| 47 | FI-Si2 | 2016 | TS_1 | -0.05 | 102.64 | 329.42 | -0.88 | 16.48 | 206.00 | 15.60 |
| 48 | FI-Sii | 2013 | NaN | NaN | NaN | NaN | NaN | NaN | NaN | NaN |
| 49 | FI-Sii | 2014 | NaN | NaN | NaN | NaN | NaN | NaN | NaN | NaN |
| 50 | FI-Sii | 2015 | NaN | NaN | NaN | NaN | NaN | NaN | NaN | NaN |
| 51 | FI-Sii | 2016 | NaN | NaN | NaN | NaN | NaN | NaN | NaN | NaN |
| 52 | FI-Sii | 2017 | NaN | NaN | NaN | NaN | NaN | NaN | NaN | NaN |
| 53 | FI-Sii | 2018 | NaN | NaN | NaN | NaN | NaN | NaN | NaN | NaN |
| 54 | HK-MPM | 2016 | TS_2 | 0.00 | NaN | 566.87 | 0.00 | 7.79 | 219.80 | 28.92 |
| 55 | HK-MPM | 2017 | TS_2 | 0.00 | NaN | NaN | 0.00 | 8.73 | 218.50 | 29.13 |
| 56 | HK-MPM | 2018 | TS_2 | 0.00 | NaN | NaN | 0.00 | 7.23 | 204.90 | 28.66 |
| 57 | ID-Pag | 2016 | NaN | NaN | NaN | NaN | NaN | NaN | NaN | NaN |
| 58 | JP-BBY | 2015 | TS_1 | -0.18 | 87.83 | 340.83 | 0.94 | 21.59 | 218.10 | 22.53 |
| 59 | JP-BBY | 2016 | TS_1 | -0.18 | 80.75 | 330.60 | 0.36 | 22.19 | 217.80 | 22.55 |
| 60 | JP-BBY | 2017 | TS_1 | -0.18 | 80.38 | 347.77 | 0.20 | 21.75 | 213.80 | 21.95 |
| 61 | JP-BBY | 2018 | TS_1 | -0.18 | 78.28 | 355.38 | 0.50 | 20.38 | 222.00 | 20.88 |
| 62 | JP-Mse | 2012 | TS_1 | -0.01 | 60.38 | 348.86 | 2.15 | 23.76 | 211.80 | 25.91 |
| 63 | KR-CRK | 2015 | NaN | NaN | NaN | NaN | NaN | NaN | NaN | NaN |
| 64 | KR-CRK | 2016 | NaN | NaN | NaN | NaN | NaN | NaN | NaN | NaN |

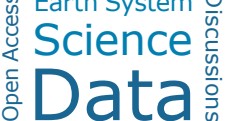

| # | Site | Year | Type | | | | | | | |
|---|---|---|---|---|---|---|---|---|---|---|
| 65 | KR-CRK | 2017 | NaN | NaN | NaN | NaN | NaN | NaN | NaN | NaN |
| 66 | KR-CRK | 2018 | NaN | NaN | NaN | NaN | NaN | NaN | NaN | NaN |
| 67 | MY-MLM | 2014 | TS | NaN | NaN | 358.37 | 25.06 | 3.97 | 194.50 | 29.03 |
| 68 | MY-MLM | 2015 | TS | NaN | 27.32 | NaN | 25.57 | 1.97 | 172.70 | 27.55 |
| 69 | NZ-Kop | 2012 | TS_1 | -0.50 | 62.54 | 360.29 | 8.30 | 8.39 | 219.80 | 16.70 |
| 70 | NZ-Kop | 2013 | TS_1 | -0.50 | 45.63 | 367.00 | 8.41 | 7.64 | 210.50 | 16.04 |
| 71 | NZ-Kop | 2014 | TS_1 | -0.50 | 56.88 | 365.68 | 8.16 | 8.79 | 219.00 | 16.95 |
| 72 | NZ-Kop | 2015 | TS_1 | -0.50 | 56.49 | 371.82 | 7.73 | 9.36 | 214.00 | 17.09 |
| 73 | PH-RiF | 2012 | NaN | NaN | NaN | NaN | NaN | NaN | NaN | NaN |
| 74 | PH-RiF | 2013 | NaN | NaN | NaN | NaN | NaN | NaN | NaN | NaN |
| 75 | PH-RiF | 2014 | NaN | NaN | NaN | NaN | NaN | NaN | NaN | NaN |
| 76 | PH-RiF | 2015 | NaN | NaN | NaN | NaN | NaN | NaN | NaN | NaN |
| 77 | RU-Ch2 | 2014 | TS_1 | -0.04 | 138.76 | 263.76 | -0.13 | 14.42 | 206.60 | 14.29 |
| 78 | RU-Ch2 | 2015 | TS_1 | -0.04 | 143.88 | 269.56 | -0.14 | 13.97 | 193.30 | 13.83 |
| 79 | RU-Ch2 | 2016 | TS_1 | -0.04 | 126.98 | 273.54 | -0.16 | 11.64 | 200.20 | 11.48 |
| 80 | RU-Che | 2014 | TS_1 | -0.04 | 138.05 | 267.57 | -0.12 | 15.04 | 208.00 | 14.92 |
| 81 | RU-Che | 2015 | TS_1 | -0.04 | 143.85 | 274.68 | -0.17 | 14.81 | 193.70 | 14.64 |
| 82 | RU-Che | 2016 | TS_1 | -0.04 | 126.72 | 274.03 | -0.19 | 12.95 | 200.40 | 12.76 |
| 83 | SE-Deg | 2014 | TS_1 | -0.02 | 111.85 | 303.55 | -0.53 | 17.22 | 201.60 | 16.69 |
| 84 | SE-Deg | 2015 | TS_1 | -0.02 | 104.25 | 310.59 | -0.28 | 15.20 | 207.20 | 14.91 |
| 85 | SE-Deg | 2016 | TS_1 | -0.02 | 108.14 | 306.39 | -0.19 | 14.88 | 200.30 | 14.68 |
| 86 | SE-Deg | 2017 | TS_1 | -0.02 | 133.38 | 326.87 | -0.20 | 12.35 | 215.00 | 12.15 |
| 87 | SE-Deg | 2018 | TS_1 | -0.02 | 111.67 | 310.21 | -0.17 | 14.70 | 198.00 | 14.52 |
| 88 | US-Atq | 2013 | NaN | NaN | NaN | NaN | NaN | NaN | NaN | NaN |
| 89 | US-Atq | 2014 | TS_1 | NaN | 10.75 | 139.53 | -0.20 | 8.07 | 61.00 | 7.86 |
| 90 | US-Atq | 2015 | NaN | NaN | NaN | NaN | NaN | NaN | NaN | NaN |
| 91 | US-Beo | 2013 | NaN | NaN | NaN | NaN | NaN | NaN | NaN | NaN |
| 92 | US-Beo | 2014 | TS_1 | NaN | 155.09 | 270.13 | -0.04 | 4.87 | 211.10 | 4.83 |
| 93 | US-Bes | 2013 | TS_1 | NaN | 143.22 | 261.99 | -0.05 | 5.65 | 201.30 | 5.60 |
| 94 | US-Bes | 2014 | TS_1 | NaN | 151.79 | 282.29 | -0.10 | 3.92 | 198.50 | 3.82 |
| 95 | US-Bes | 2015 | TS_1 | NaN | 140.45 | 270.98 | -0.11 | 4.76 | 195.30 | 4.65 |
| 96 | US-BZB | 2014 | TS_1 | -0.08 | 123.11 | 298.35 | -0.44 | 15.35 | 215.80 | 14.91 |
| 97 | US-BZB | 2015 | TS_1 | -0.08 | 107.82 | 295.60 | -0.38 | 14.04 | 210.20 | 13.67 |
| 98 | US-BZB | 2016 | TS_1 | -0.08 | 125.09 | 292.39 | -0.33 | 16.39 | 214.30 | 16.06 |

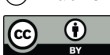



| | | | | | | | | | | |
|---|---|---|---|---|---|---|---|---|---|---|
| 99 | US-BZF | 2014 | TS_1 | -0.08 | 96.05 | 322.79 | -1.56 | 16.12 | 205.40 | 14.56 |
| 100 | US-BZF | 2015 | TS_1 | -0.08 | 108.39 | 331.07 | -1.20 | 14.88 | 197.50 | 13.68 |
| 101 | US-BZF | 2016 | TS_1 | -0.08 | 95.50 | 315.89 | -1.03 | 17.18 | 205.10 | 16.15 |
| 102 | US-BZS | 2015 | TS_1 | NaN | 116.48 | 275.22 | -0.07 | 4.87 | 202.90 | 4.79 |
| 103 | US-BZS | 2016 | TS_1 | NaN | 119.07 | 278.90 | -0.05 | 5.58 | 208.40 | 5.54 |
| 104 | US-DPW | 2013 | NaN | NaN | NaN | NaN | NaN | NaN | NaN | NaN |
| 105 | US-DPW | 2014 | NaN | NaN | NaN | NaN | NaN | NaN | NaN | NaN |
| 106 | US-DPW | 2015 | NaN | NaN | NaN | NaN | NaN | NaN | NaN | NaN |
| 107 | US-DPW | 2016 | NaN | NaN | NaN | NaN | NaN | NaN | NaN | NaN |
| 108 | US-HRA | 2017 | NaN | NaN | NaN | NaN | NaN | NaN | NaN | NaN |
| 109 | US-HRC | 2018 | NaN | NaN | NaN | NaN | NaN | NaN | NaN | NaN |
| 110 | US-ICs | 2014 | TS_1 | -0.08 | 147.27 | 263.08 | -0.08 | 5.47 | 205.60 | 5.39 |
| 111 | US-ICs | 2015 | TS_1 | -0.08 | 141.61 | 255.70 | -0.02 | 6.12 | 195.20 | 6.10 |
| 112 | US-ICs | 2016 | TS_1 | -0.08 | 146.75 | 265.86 | -0.05 | 5.67 | 206.20 | 5.62 |
| 113 | US-Ivo | 2013 | NaN | NaN | NaN | NaN | NaN | NaN | NaN | NaN |
| 114 | US-Ivo | 2014 | TS_1 | -0.05 | 136.92 | 264.34 | -0.20 | 10.84 | 195.70 | 10.64 |
| 115 | US-Ivo | 2015 | TS_1 | -0.05 | 139.42 | 257.06 | -0.14 | 14.86 | 185.60 | 14.72 |
| 116 | US-Ivo | 2016 | TS_1 | -0.05 | 133.60 | 262.43 | -0.10 | 8.90 | 197.30 | 8.80 |
| 117 | US-LA1 | 2012 | TS_1 | -0.10 | 29.15 | 331.44 | 15.59 | 13.50 | 197.20 | 29.08 |
| 118 | US-LA2 | 2012 | TS_1 | -0.10 | 36.65 | 336.05 | 15.04 | 14.35 | 193.20 | 29.39 |
| 119 | US-LA2 | 2013 | TS_1 | -0.10 | 65.79 | 377.93 | 14.70 | 16.06 | 201.50 | 30.76 |
| 120 | US-Los | 2014 | TS_1 | 0.00 | 136.18 | 417.40 | 1.95 | 8.26 | 244.30 | 10.22 |
| 121 | US-Los | 2015 | TS_1 | 0.00 | 148.23 | 422.27 | 2.43 | 7.76 | 258.70 | 10.19 |
| 122 | US-Los | 2016 | TS_1 | 0.00 | 135.20 | 415.75 | 2.47 | 8.08 | 255.10 | 10.56 |
| 123 | US-Los | 2017 | TS_1 | 0.00 | 141.15 | 414.62 | 1.89 | 7.45 | 256.00 | 9.34 |
| 124 | US-Los | 2018 | TS_1 | 0.00 | 169.83 | 421.79 | 1.46 | 7.42 | 260.00 | 8.88 |
| 125 | US-MAC | 2013 | NaN | NaN | NaN | NaN | NaN | NaN | NaN | NaN |
| 126 | US-MAC | 2014 | NaN | NaN | NaN | NaN | NaN | NaN | NaN | NaN |
| 127 | US-MAC | 2015 | NaN | NaN | NaN | NaN | NaN | NaN | NaN | NaN |
| 128 | US-Myb | 2010 | NaN | NaN | NaN | NaN | NaN | NaN | NaN | NaN |
| 129 | US-Myb | 2011 | TS_3 | -0.08 | NaN | 329.50 | 12.12 | 8.86 | 231.70 | 20.98 |
| 130 | US-Myb | 2012 | TS_3 | -0.08 | 35.60 | 372.21 | 9.39 | 10.96 | 216.10 | 20.36 |
| 131 | US-Myb | 2013 | TS_3 | -0.08 | 34.38 | 354.74 | 9.25 | 11.28 | 210.90 | 20.52 |
| 132 | US-Myb | 2014 | TS_3 | -0.08 | 26.04 | 365.88 | 9.64 | 12.28 | 210.00 | 21.93 |

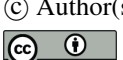



| | | | | | | | | | | |
|---|---|---|---|---|---|---|---|---|---|---|
| 133 | US-Myb | 2015 | TS_3 | -0.08 | 17.34 | 340.72 | 9.77 | 11.85 | 212.00 | 21.62 |
| 134 | US-Myb | 2016 | TS_3 | -0.08 | 5.05 | 357.82 | 9.61 | 11.19 | 201.00 | 20.80 |
| 135 | US-Myb | 2017 | TS_3 | -0.08 | 27.31 | 352.90 | 9.86 | 13.10 | 214.00 | 22.96 |
| 136 | US-Myb | 2018 | TS_3 | -0.08 | 36.27 | 326.10 | 10.20 | 10.72 | 207.00 | 20.92 |
| 137 | US-NC4 | 2012 | TS_1 | -0.05 | 42.43 | 336.03 | 7.87 | 15.75 | 215.40 | 23.62 |
| 138 | US-NC4 | 2013 | TS_1 | -0.05 | 59.96 | 368.85 | 6.92 | 16.83 | 210.50 | 23.74 |
| 139 | US-NC4 | 2014 | TS_1 | -0.05 | 54.42 | 362.34 | 6.73 | 16.81 | 208.50 | 23.54 |
| 140 | US-NC4 | 2015 | TS_1 | -0.05 | 68.13 | 387.31 | 8.27 | 16.06 | 205.00 | 24.33 |
| 141 | US-NC4 | 2016 | TS_1 | -0.05 | 52.97 | 351.38 | 9.41 | 15.04 | 220.00 | 24.45 |
| 142 | US-ORv | 2011 | NaN | NaN | NaN | NaN | NaN | NaN | NaN | NaN |
| 143 | US-ORv | 2012 | TS | NaN | 57.42 | 352.55 | 4.36 | 20.61 | 203.90 | 24.97 |
| 144 | US-ORv | 2013 | TS | NaN | 63.67 | 356.74 | 2.93 | 19.98 | 210.80 | 22.90 |
| 145 | US-ORv | 2014 | TS | NaN | 68.11 | 364.96 | 2.11 | 20.17 | 205.30 | 22.28 |
| 146 | US-ORv | 2015 | TS | NaN | 68.77 | 387.78 | 1.77 | 21.26 | 206.00 | 23.04 |
| 147 | US-OWC | 2015 | NaN | NaN | NaN | NaN | NaN | NaN | NaN | NaN |
| 148 | US-OWC | 2016 | TS_1 | -0.05 | 0.00 | 0.00 | 0.00 | 0.00 | 211.20 | 23.91 |
| 149 | US-Sne | 2016 | NaN | NaN | NaN | NaN | NaN | NaN | NaN | NaN |
| 150 | US-Sne | 2017 | TS_1 | -0.01 | 46.07 | 337.92 | 10.33 | 13.14 | 212.70 | 23.47 |
| 151 | US-Sne | 2018 | TS_1 | -0.01 | 48.41 | 325.09 | 10.28 | 12.15 | 217.30 | 22.43 |
| 152 | US-Srr | 2014 | NaN | NaN | NaN | NaN | NaN | NaN | NaN | NaN |
| 153 | US-Srr | 2015 | NaN | NaN | NaN | NaN | NaN | NaN | NaN | NaN |
| 154 | US-Srr | 2016 | TS_1 | NaN | NaN | 326.29 | 10.03 | 10.71 | 200.50 | 20.74 |
| 155 | US-Srr | 2017 | TS_1 | NaN | 11.34 | 346.85 | 7.22 | 13.71 | 199.50 | 20.93 |
| 156 | US-StJ | 2016 | TS_2 | -0.05 | 68.37 | 347.38 | 4.05 | 16.22 | 213.70 | 20.27 |
| 157 | US-Tw1 | 2011 | NaN | NaN | NaN | NaN | NaN | NaN | NaN | NaN |
| 158 | US-Tw1 | 2012 | TS_1 | -0.02 | 50.54 | 359.96 | 5.99 | 11.52 | 225.20 | 17.51 |
| 159 | US-Tw1 | 2013 | TS_1 | -0.02 | 35.80 | 337.57 | 4.39 | 14.59 | 206.60 | 18.97 |
| 160 | US-Tw1 | 2014 | TS_1 | -0.02 | 41.23 | 395.41 | 6.67 | 10.89 | 208.30 | 17.56 |
| 161 | US-Tw1 | 2015 | TS_1 | -0.02 | 50.66 | 342.55 | 9.02 | 7.83 | 235.00 | 16.85 |
| 162 | US-Tw1 | 2016 | TS_1 | -0.02 | 34.57 | 361.06 | 8.78 | 7.94 | 218.00 | 16.72 |
| 163 | US-Tw1 | 2017 | TS_1 | -0.02 | 41.17 | 343.75 | 7.55 | 10.22 | 228.00 | 17.77 |
| 164 | US-Tw1 | 2018 | TS_1 | -0.02 | 60.47 | 327.43 | 6.70 | 10.42 | 222.00 | 17.12 |
| 165 | US-Tw4 | 2014 | NaN | NaN | NaN | NaN | NaN | NaN | NaN | NaN |
| 166 | US-Tw4 | 2015 | TS_1 | -0.02 | 15.93 | 327.22 | 10.04 | 11.56 | 199.40 | 21.60 |





| | Site | Year | Probe_name | Soil_temp_depth_m | | | | | | |
|---|---|---|---|---|---|---|---|---|---|---|
| 167 | US-Tw4 | 2016 | TS_1 | -0.02 | 9.56 | 358.75 | 8.16 | 11.29 | 201.70 | 19.45 |
| 168 | US-Tw4 | 2017 | TS_1 | -0.02 | 38.35 | 347.31 | 8.07 | 11.52 | 211.90 | 19.59 |
| 169 | US-Tw4 | 2018 | TS_1 | -0.02 | 58.11 | 344.87 | 8.00 | 10.93 | 218.00 | 18.93 |
| 170 | US-Tw5 | 2018 | TS_1 | -0.02 | NaN | 414.83 | 0.00 | 8.89 | 222.20 | 18.37 |
| 171 | US-Twt | 2009 | NaN | NaN | NaN | NaN | NaN | NaN | NaN | NaN |
| 172 | US-Twt | 2010 | NaN | NaN | NaN | NaN | NaN | NaN | NaN | NaN |
| 173 | US-Twt | 2011 | NaN | NaN | NaN | NaN | NaN | NaN | NaN | NaN |
| 174 | US-Twt | 2012 | NaN | NaN | NaN | NaN | NaN | NaN | NaN | NaN |
| 175 | US-Twt | 2013 | NaN | NaN | NaN | NaN | NaN | NaN | NaN | NaN |
| 176 | US-Twt | 2014 | NaN | NaN | NaN | NaN | NaN | NaN | NaN | NaN |
| 177 | US-Twt | 2015 | NaN | NaN | NaN | NaN | NaN | NaN | NaN | NaN |
| 178 | US-Twt | 2016 | NaN | NaN | NaN | NaN | NaN | NaN | NaN | NaN |
| 179 | US-Uaf | 2011 | TS_1 | -0.09 | 86.20 | 372.46 | -12.29 | 21.95 | 199.60 | 9.67 |
| 180 | US-Uaf | 2012 | TS_1 | -0.09 | 73.77 | 338.53 | -11.83 | 20.86 | 202.40 | 9.03 |
| 181 | US-Uaf | 2013 | TS_1 | -0.09 | 109.63 | 395.51 | -10.08 | 20.52 | 200.40 | 10.44 |
| 182 | US-Uaf | 2014 | TS_1 | -0.09 | 76.07 | 365.40 | -10.94 | 19.94 | 206.00 | 9.00 |
| 183 | US-Uaf | 2015 | TS_1 | -0.09 | 80.99 | 423.19 | -9.77 | 19.76 | 190.00 | 10.00 |
| 184 | US-Uaf | 2016 | TS_1 | -0.09 | 77.38 | 315.75 | -7.74 | 19.13 | 198.00 | 11.39 |
| 185 | US-Uaf | 2017 | TS_1 | -0.09 | 84.88 | 380.17 | -7.39 | 19.08 | 196.00 | 11.69 |
| 186 | US-Uaf | 2018 | TS_1 | -0.09 | 96.04 | 333.33 | -5.60 | 17.72 | 199.00 | 12.11 |
| 187 | US-WPT | 2011 | TS_1 | -0.10 | 80.95 | 342.17 | 5.27 | 19.27 | 202.60 | 24.54 |
| 188 | US-WPT | 2012 | TS_1 | -0.10 | 40.29 | 345.57 | 3.70 | 21.60 | 197.40 | 25.30 |
| 189 | US-WPT | 2013 | TS_1 | -0.10 | 74.61 | 340.47 | 3.73 | 18.23 | 207.20 | 21.96 |

**Column Descriptions**

| | |
|---|---|
| Site | Site name |
| Year | Data year |
| Probe_name | Temperature probe name as given in data files |
| Soil_temp_depth_m | Depth of soil temperature probe (m), with negative values being under the surface |
| Start_TS_(DOY) | Season start for elevated TS (DOY), point "f" in Figure 1 |
| End_TS_(DOY) | Season end for elevated TS (DOY), point "h" in Figure 1 |
| Base_value_TS_(C) | Baseline TS during non-elevated season (C), average of points "a" and "b" in Figure 1 |





| Ampl_TS_(C) | Amplitude of TS during elevated temperature season (C), difference between point "e" in Figure 1 and Base_value_TS |
| Peak_TS_(DOY) | Day of maximum elevated TS (DOY), point "g" in Figure 1 |
| Peak_value_TS_(C) | Maximum value of TS (C) point "e" in Figure 1 |



**Table B5: Timesat output for all soil temperature probes**

| | Site | Year | Probe_name | Soil_temp_depth_m | Start_TS_(DOY) | End_TS_(DOY) | Base_value_TS_(C) | Ampl_TS_(C) | Peak_TS_(DOY) | Peak_value_TS_(C) |
|---|---|---|---|---|---|---|---|---|---|---|
| 1 | AT-Neu | 2010 | TS_1 | -0.05 | 61.32 | 339.44 | 0.15 | 17.54 | 200.9 | 17.7 |
| 2 | AT-Neu | 2011 | TS_1 | -0.05 | 51.04 | 328.84 | 0.40 | 16.37 | 201 | 16.77 |
| 3 | AT-Neu | 2012 | TS_1 | -0.05 | 61.12 | 341.88 | 0.73 | 17.57 | 202.9 | 18.3 |
| 4 | BR-Npw | 2016 | TS_1 | NaN | 18.41 | 343.22 | 22.41 | 5.982 | 188 | 28.4 |
| 5 | CA-SCB | 2014 | TS_1 | 0 | 105.94 | 292.18 | -0.63 | 20.62 | 196.6 | 19.99 |
| 6 | CA-SCB | 2014 | TS_2 | -0.02 | 105.15 | 294.06 | -0.74 | 20.42 | 197.5 | 19.68 |
| 7 | CA-SCB | 2014 | TS_3 | -0.04 | 112.00 | 294.38 | 0.05 | 19.07 | 199.6 | 19.11 |
| 8 | CA-SCB | 2014 | TS_5 | -0.16 | 123.21 | 317.72 | -1.38 | 18.6 | 205.3 | 17.23 |
| 9 | CA-SCB | 2015 | TS_1 | 0 | 106.92 | 287.14 | -0.39 | 17.23 | 186.8 | 16.84 |
| 10 | CA-SCB | 2015 | TS_2 | -0.02 | 107.06 | 287.40 | -0.42 | 17.08 | 187.4 | 16.66 |
| 11 | CA-SCB | 2015 | TS_3 | -0.04 | 107.45 | 289.83 | -0.51 | 16.81 | 188.9 | 16.3 |
| 12 | CA-SCB | 2015 | TS_5 | -0.16 | 114.95 | 305.55 | -0.39 | 15.84 | 195.7 | 15.45 |
| 13 | CA-SCB | 2016 | TS_1 | 0 | 101.64 | 284.09 | -0.31 | 19.07 | 193.2 | 18.77 |
| 14 | CA-SCB | 2016 | TS_2 | -0.02 | 101.81 | 284.11 | -0.30 | 18.96 | 193.5 | 18.66 |
| 15 | CA-SCB | 2016 | TS_3 | -0.04 | 102.22 | 285.19 | -0.30 | 18.6 | 194.3 | 18.3 |
| 16 | CA-SCB | 2016 | TS_5 | -0.16 | 101.16 | 298.99 | -0.24 | 16.99 | 201.1 | 16.74 |
| 17 | CA-SCB | 2017 | TS_1 | 0 | 107.44 | 289.72 | -0.25 | 17.67 | 198 | 17.42 |
| 18 | CA-SCB | 2017 | TS_2 | -0.02 | 107.22 | 288.88 | -0.25 | 17.59 | 198 | 17.33 |
| 19 | CA-SCB | 2017 | TS_3 | -0.04 | 108.58 | 289.29 | -0.26 | 17.28 | 199 | 17.02 |
| 20 | CA-SCB | 2017 | TS_5 | -0.16 | 116.34 | 300.28 | -0.24 | 14.95 | 214 | 14.71 |
| 21 | CA-SCC | 2014 | TS_1 | -0.1 | 123.36 | 287.06 | -0.55 | 15.64 | 203 | 15.09 |
| 22 | CA-SCC | 2014 | TS_2 | -0.15 | 114.89 | 287.71 | -0.83 | 14.49 | 200.8 | 13.66 |
| 23 | CA-SCC | 2014 | TS_3 | -0.2 | 111.36 | 288.63 | -0.69 | 11.52 | 194.9 | 10.84 |
| 24 | CA-SCC | 2014 | TS_4 | -0.25 | 129.50 | 287.24 | -0.22 | 8.612 | 207.4 | 8.391 |
| 25 | CA-SCC | 2014 | TS_5 | -0.3 | 142.36 | 287.99 | -0.10 | 6.329 | 212.1 | 6.225 |
| 26 | CA-SCC | 2015 | TS_1 | -0.1 | 113.88 | 285.22 | -0.28 | 16.26 | 189.2 | 15.98 |
| 27 | CA-SCC | 2015 | TS_2 | -0.15 | 113.05 | 284.18 | -0.24 | 14.56 | 192.8 | 14.32 |
| 28 | CA-SCC | 2015 | TS_3 | -0.2 | 111.76 | 285.45 | -0.22 | 12.71 | 199.1 | 12.48 |
| 29 | CA-SCC | 2015 | TS_4 | -0.25 | 120.81 | 287.09 | -0.16 | 10.08 | 204.8 | 9.922 |
| 30 | CA-SCC | 2015 | TS_5 | -0.3 | 131.92 | 285.42 | -0.09 | 7.705 | 209.2 | 7.616 |



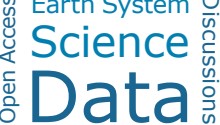

| | | | | | | | | | |
|---|---|---|---|---|---|---|---|---|---|
| 31 | CA-SCC | 2016 TS_1 | -0.1 | 108.09 | 260.12 | -0.33 | 18.37 | 190.9 | 18.04 |
| 32 | CA-SCC | 2016 TS_2 | -0.15 | 108.96 | 260.19 | -0.30 | 17.31 | 192.1 | 17.01 |
| 33 | CA-SCC | 2016 TS_3 | -0.2 | 110.49 | 260.44 | -0.26 | 15.4 | 194.1 | 15.14 |
| 34 | CA-SCC | 2016 TS_4 | -0.25 | 119.21 | 260.34 | -0.20 | 13.38 | 200.2 | 13.18 |
| 35 | CA-SCC | 2016 TS_5 | -0.3 | 130.75 | 261.73 | -0.12 | 10.03 | 202.2 | 9.906 |
| 36 | DE-Hte | 2012 TS_3 | -0.2 | 76.96 | 344.01 | 4.98 | 12.26 | 215.5 | 17.23 |
| 37 | DE-Hte | 2013 TS_3 | -0.2 | 60.92 | 377.99 | 3.96 | 11.99 | 207.9 | 15.95 |
| 38 | DE-Hte | 2014 TS_1 | 0 | NaN | 327.83 | 8.52 | 8.342 | 205.6 | 16.87 |
| 39 | DE-Hte | 2015 TS_1 | 0 | 62.92 | 360.55 | 5.17 | 11.67 | 187.4 | 16.84 |
| 40 | DE-Hte | 2016 TS_1 | 0 | 71.87 | NaN | 4.94 | 12.36 | 175.6 | 17.3 |
| 41 | DE-Hte | 2017 TS_1 | 0 | 61.55 | 343.32 | 4.50 | 11.76 | 186 | 16.26 |
| 42 | DE-SfN | 2012 TS_1 | -0.02 | NaN | 372.59 | 0.00 | 23.55 | 206.5 | 15.29 |
| 43 | DE-SfN | 2012 TS_3 | -0.1 | NaN | 366.65 | 1.64 | 12.91 | 219.7 | 14.55 |
| 44 | DE-SfN | 2012 TS_4 | -0.2 | NaN | 367.40 | 4.86 | 7.276 | 242.7 | 12.14 |
| 45 | DE-SfN | 2012 TS_5 | -0.5 | NaN | 367.40 | 4.86 | 7.276 | 242.7 | 12.14 |
| 46 | DE-SfN | 2013 TS_1 | -0.02 | 55.84 | 381.50 | 0.92 | 13.62 | 216.4 | 14.54 |
| 47 | DE-SfN | 2013 TS_3 | -0.1 | 60.45 | 384.77 | 1.56 | 12.5 | 221.1 | 14.06 |
| 48 | DE-SfN | 2013 TS_4 | -0.2 | 83.55 | 394.53 | 3.62 | 8.417 | 243.4 | 12.04 |
| 49 | DE-SfN | 2013 TS_5 | -0.5 | 83.55 | 394.53 | 3.62 | 8.417 | 243.4 | 12.04 |
| 50 | DE-Zrk | 2014 TS_1 | -0.05 | 54.79 | 361.65 | 4.36 | 13.93 | 202.3 | 18.29 |
| 51 | DE-Zrk | 2014 TS_2 | -0.1 | 59.27 | 366.51 | 4.87 | 12.65 | 207.3 | 17.52 |
| 52 | DE-Zrk | 2014 TS_3 | -0.2 | 62.95 | 371.30 | 5.53 | 11.5 | 211.7 | 17.03 |
| 53 | DE-Zrk | 2014 TS_4 | -0.3 | 67.45 | 375.14 | 6.05 | 10.4 | 216.5 | 16.45 |
| 54 | DE-Zrk | 2014 TS_5 | -0.5 | 72.50 | 378.95 | 6.57 | 9.359 | 221 | 15.93 |
| 55 | DE-Zrk | 2015 TS_1 | -0.05 | 58.29 | 359.29 | 4.28 | 13.24 | 215.5 | 17.52 |
| 56 | DE-Zrk | 2015 TS_2 | -0.1 | 62.61 | 364.99 | 4.79 | 12 | 219.8 | 16.8 |
| 57 | DE-Zrk | 2015 TS_3 | -0.2 | 66.01 | 369.78 | 5.42 | 10.87 | 223.7 | 16.29 |
| 58 | DE-Zrk | 2015 TS_4 | -0.3 | 70.47 | 374.40 | 5.93 | 9.771 | 228 | 15.7 |
| 59 | DE-Zrk | 2015 TS_5 | -0.5 | 74.71 | 378.76 | 6.43 | 8.751 | 232.2 | 15.19 |
| 60 | DE-Zrk | 2016 TS_1 | -0.05 | 72.81 | 332.00 | 4.28 | 14.93 | 200.4 | 19.2 |
| 61 | DE-Zrk | 2016 TS_2 | -0.1 | 76.31 | 337.37 | 4.79 | 13.6 | 204.4 | 18.39 |
| 62 | DE-Zrk | 2016 TS_3 | -0.2 | 79.73 | 343.16 | 5.43 | 12.33 | 208.1 | 17.77 |
| 63 | DE-Zrk | 2016 TS_4 | -0.3 | 83.58 | 347.57 | 5.94 | 11.14 | 212 | 17.09 |
| 64 | DE-Zrk | 2016 TS_5 | -0.5 | 87.15 | 354.22 | 6.40 | 10.07 | 216 | 16.47 |


Open Access — Earth System Science Data Discussions

| # | Site | Year / TS | | | | | | | |
|---|------|-----------|------|--------|--------|-------|--------|-------|-------|
| 65 | DE-Zrk | 2017 TS_1 | -0.05 | 69.10 | 351.44 | 4.40 | 14.72 | 199 | 19.12 |
| 66 | DE-Zrk | 2017 TS_2 | -0.1 | 73.29 | 356.52 | 4.91 | 13.33 | 204 | 18.23 |
| 67 | DE-Zrk | 2017 TS_3 | -0.2 | 77.19 | 362.40 | 5.56 | 12.02 | 208 | 17.58 |
| 68 | DE-Zrk | 2017 TS_4 | -0.3 | 82.20 | 367.20 | 6.04 | 10.8 | 212 | 16.84 |
| 69 | DE-Zrk | 2017 TS_5 | -0.5 | 86.22 | 372.96 | 6.48 | 9.675 | 217 | 16.15 |
| 70 | DE-Zrk | 2018 TS_1 | -0.05 | 84.47 | 336.14 | 4.83 | 12.32 | 203 | 17.14 |
| 71 | DE-Zrk | 2018 TS_2 | -0.1 | 86.60 | 342.59 | 5.30 | 11.27 | 208 | 16.57 |
| 72 | DE-Zrk | 2018 TS_3 | -0.2 | 87.68 | 348.07 | 5.89 | 10.25 | 212 | 16.14 |
| 73 | DE-Zrk | 2018 TS_4 | -0.3 | 89.82 | 354.77 | 6.31 | 9.308 | 217 | 15.61 |
| 74 | DE-Zrk | 2018 TS_5 | -0.5 | 92.01 | 360.46 | 6.69 | 8.412 | 222 | 15.11 |
| 75 | FI-Lom | 2006 TS_1 | -0.07 | 114.15 | 290.83 | -0.11 | 13.42 | 204.8 | 13.31 |
| 76 | FI-Lom | 2006 TS_2 | -0.3 | 117.21 | 307.88 | 0.27 | 12.01 | 214.1 | 12.28 |
| 77 | FI-Lom | 2006 TS_3 | -0.5 | 128.82 | 329.03 | 1.06 | 9.071 | 225.8 | 10.13 |
| 78 | FI-Lom | 2007 TS_1 | -0.07 | 126.84 | 302.05 | 0.11 | 13.05 | 200 | 13.16 |
| 79 | FI-Lom | 2007 TS_2 | -0.3 | 134.00 | 321.03 | 0.42 | 11.5 | 207.5 | 11.92 |
| 80 | FI-Lom | 2007 TS_3 | -0.5 | 138.37 | 348.03 | 1.06 | 8.873 | 221.1 | 9.936 |
| 81 | FI-Lom | 2008 TS_1 | -0.07 | 135.62 | 296.74 | 0.16 | 12.73 | 202.9 | 12.88 |
| 82 | FI-Lom | 2008 TS_2 | -0.3 | 141.46 | 318.74 | 0.58 | 10.62 | 209.6 | 11.2 |
| 83 | FI-Lom | 2008 TS_3 | -0.5 | 146.70 | 349.21 | 1.17 | 8.214 | 221.2 | 9.382 |
| 84 | FI-Lom | 2009 TS_1 | -0.07 | 117.16 | 291.86 | 0.14 | 11.73 | 214 | 11.87 |
| 85 | FI-Lom | 2009 TS_2 | -0.3 | 123.51 | 314.51 | 0.67 | 9.692 | 221 | 10.36 |
| 86 | FI-Lom | 2009 TS_3 | -0.5 | 133.69 | 336.65 | 1.30 | 7.896 | 233 | 9.193 |
| 87 | FI-Lom | 2010 TS_1 | -0.07 | 129.91 | 318.54 | 0.05 | 12.13 | 208 | 12.18 |
| 88 | FI-Lom | 2010 TS_2 | -0.3 | 138.09 | 338.24 | 0.52 | 9.962 | 218 | 10.48 |
| 89 | FI-Lom | 2010 TS_3 | -0.5 | 147.34 | 359.95 | 1.19 | 7.344 | 231 | 8.532 |
| 90 | FI-Si2 | 2012 TS_1 | -0.05 | NaN | 323.46 | 0.00 | 19.85 | 204.6 | 16.02 |
| 91 | FI-Si2 | 2012 TS_2 | -0.2 | 103.64 | 333.52 | -0.04 | 15.75 | 217.5 | 15.71 |
| 92 | FI-Si2 | 2012 TS_3 | -0.35 | 105.57 | NaN | 0.00 | 19.38 | 230.6 | 15.09 |
| 93 | FI-Si2 | 2012 TS_4 | -0.5 | 110.87 | NaN | 0.00 | 17.26 | 237.5 | 14.66 |
| 94 | FI-Si2 | 2013 TS_1 | -0.05 | 106.90 | 341.05 | -0.05 | 16.04 | 199.4 | 15.98 |
| 95 | FI-Si2 | 2013 TS_2 | -0.2 | 102.57 | 356.13 | 0.23 | 14.91 | 207.3 | 15.14 |
| 96 | FI-Si2 | 2013 TS_3 | -0.35 | NaN | 376.47 | 0.00 | 18.26 | 209.6 | 14.23 |
| 97 | FI-Si2 | 2013 TS_4 | -0.5 | NaN | 392.35 | 0.00 | 16.71 | 216.4 | 13.48 |
| 98 | FI-Si2 | 2014 TS_1 | -0.05 | 104.63 | 331.10 | -0.04 | 17.07 | 208.5 | 17.03 |



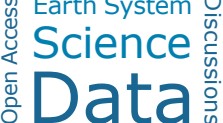

| # | Site | Year TS | | | | | | | |
|---|---|---|---|---|---|---|---|---|---|
| 99 | FI-Si2 | 2014 TS_2 | -0.2 | 107.82 | 359.78 | 0.59 | 15.33 | 215.3 | 15.92 |
| 100 | FI-Si2 | 2014 TS_3 | -0.35 | 112.02 | 385.94 | 0.99 | 13.61 | 222.2 | 14.61 |
| 101 | FI-Si2 | 2014 TS_4 | -0.5 | 118.24 | 400.15 | 1.59 | 12.01 | 229.1 | 13.59 |
| 102 | FI-Si2 | 2015 TS_1 | -0.05 | 76.49 | 352.34 | -0.87 | 16.31 | 211 | 15.44 |
| 103 | FI-Si2 | 2015 TS_2 | -0.2 | 80.08 | 364.72 | -0.41 | 14.83 | 218 | 14.42 |
| 104 | FI-Si2 | 2015 TS_3 | -0.35 | 82.01 | 374.79 | 0.12 | 13.27 | 225 | 13.39 |
| 105 | FI-Si2 | 2015 TS_4 | -0.5 | 88.25 | 382.37 | 0.84 | 11.7 | 233 | 12.53 |
| 106 | FI-Si2 | 2016 TS_1 | -0.05 | 102.64 | 329.42 | -0.88 | 16.48 | 206 | 15.6 |
| 107 | FI-Si2 | 2016 TS_2 | -0.2 | 102.16 | 361.04 | -0.77 | 16.02 | 212 | 15.25 |
| 108 | FI-Si2 | 2016 TS_3 | -0.35 | 103.15 | 383.82 | -0.63 | 14.7 | 219 | 14.07 |
| 109 | FI-Si2 | 2016 TS_4 | -0.5 | 104.76 | 399.01 | -0.20 | 13.36 | 227 | 13.16 |
| 110 | HK-MPM | 2016 TS_2 | NaN | NaN | 566.87 | 0.00 | 7.789 | 219.8 | 28.92 |
| 111 | HK-MPM | 2016 TS_3 | NaN | NaN | 373.06 | 20.56 | 7.386 | 227.3 | 27.95 |
| 112 | HK-MPM | 2017 TS_2 | NaN | NaN | NaN | 0.00 | 8.726 | 218.5 | 29.13 |
| 113 | HK-MPM | 2017 TS_3 | NaN | 69.55 | 364.53 | 19.53 | 8.572 | 233.6 | 28.1 |
| 114 | HK-MPM | 2018 TS_2 | NaN | NaN | NaN | 0.00 | 7.231 | 204.9 | 28.66 |
| 115 | HK-MPM | 2018 TS_3 | NaN | 64.82 | 383.39 | 19.17 | 8.406 | 221.4 | 27.58 |
| 116 | JP-BBY | 2015 TS_1 | -0.183 | 87.83 | 340.83 | 0.94 | 21.59 | 218.1 | 22.53 |
| 117 | JP-BBY | 2015 TS_2 | -0.233 | 90.59 | 340.99 | 1.34 | 20.9 | 219.9 | 22.25 |
| 118 | JP-BBY | 2015 TS_3 | -0.283 | 90.34 | 341.60 | 1.58 | 20.42 | 221.4 | 22 |
| 119 | JP-BBY | 2015 TS_4 | -0.383 | 96.01 | 341.35 | 2.39 | 19.09 | 225.1 | 21.48 |
| 120 | JP-BBY | 2015 TS_5 | -0.483 | 95.83 | 341.49 | 2.91 | 18.09 | 228.9 | 21 |
| 121 | JP-BBY | 2016 TS_1 | -0.183 | 80.75 | 330.60 | 0.36 | 22.19 | 217.8 | 22.55 |
| 122 | JP-BBY | 2016 TS_2 | -0.233 | 82.30 | 335.40 | 0.67 | 21.64 | 220.8 | 22.3 |
| 123 | JP-BBY | 2016 TS_3 | -0.283 | 84.28 | 332.64 | 0.99 | 21.1 | 222 | 22.09 |
| 124 | JP-BBY | 2016 TS_4 | -0.383 | 89.00 | 332.43 | 1.76 | 19.92 | 225.6 | 21.68 |
| 125 | JP-BBY | 2016 TS_5 | -0.483 | 94.29 | 331.91 | 2.44 | 18.83 | 228.9 | 21.27 |
| 126 | JP-BBY | 2017 TS_1 | -0.183 | 80.38 | 347.77 | 0.20 | 21.75 | 213.8 | 21.95 |
| 127 | JP-BBY | 2017 TS_2 | -0.233 | 84.54 | 347.79 | 0.82 | 21.02 | 214.6 | 21.83 |
| 128 | JP-BBY | 2017 TS_3 | -0.283 | 86.19 | 347.10 | 1.06 | 20.56 | 216.3 | 21.62 |
| 129 | JP-BBY | 2017 TS_4 | -0.383 | 92.20 | 346.01 | 1.97 | 19.34 | 218.5 | 21.31 |
| 130 | JP-BBY | 2017 TS_5 | -0.483 | 98.15 | 345.18 | 2.70 | 18.34 | 221 | 21.03 |
| 131 | JP-BBY | 2018 TS_1 | -0.183 | 78.28 | 355.38 | 0.50 | 20.38 | 222 | 20.88 |
| 132 | JP-BBY | 2018 TS_2 | -0.233 | 83.55 | 357.60 | 1.48 | 19.23 | 224 | 20.7 |

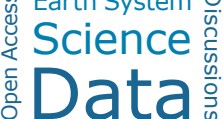



| # | | | | | | | | | |
|---|---|---|---|---|---|---|---|---|---|
| 133 | JP-BBY | 2018 TS_3 | -0.283 | 85.93 | 355.90 | 1.69 | 18.78 | 225 | 20.47 |
| 134 | JP-BBY | 2018 TS_4 | -0.383 | 95.89 | 351.34 | 2.81 | 17.25 | 229 | 20.07 |
| 135 | JP-BBY | 2018 TS_5 | -0.483 | 103.83 | 349.39 | 3.63 | 16.1 | 232 | 19.73 |
| 136 | JP-Mse | 2012 TS_1 | -0.01 | 60.38 | 348.86 | 2.15 | 23.76 | 211.8 | 25.91 |
| 137 | MY-MLM | 2014 TS | NaN | NaN | 358.37 | 25.06 | 3.968 | 194.5 | 29.03 |
| 138 | MY-MLM | 2015 TS | NaN | 27.32 | NaN | 25.57 | 1.973 | 172.7 | 27.55 |
| 139 | NZ-Kop | 2012 TS_1 | -0.5 | 62.54 | 360.29 | 8.30 | 8.394 | 219.8 | 16.7 |
| 140 | NZ-Kop | 2012 TS_2 | -0.1 | 65.53 | 362.35 | 8.45 | 8.093 | 222.1 | 16.54 |
| 141 | NZ-Kop | 2012 TS_3 | -0.2 | 68.77 | 365.64 | 8.73 | 7.243 | 228.2 | 15.98 |
| 142 | NZ-Kop | 2013 TS_1 | -0.5 | 45.63 | 367.00 | 8.41 | 7.635 | 210.5 | 16.04 |
| 143 | NZ-Kop | 2013 TS_2 | -0.1 | 47.60 | 370.98 | 8.54 | 7.486 | 212.3 | 16.03 |
| 144 | NZ-Kop | 2013 TS_3 | -0.2 | 52.74 | 377.20 | 8.82 | 6.87 | 217.8 | 15.69 |
| 145 | NZ-Kop | 2014 TS_1 | -0.5 | 56.88 | 365.68 | 8.16 | 8.79 | 219 | 16.95 |
| 146 | NZ-Kop | 2014 TS_2 | -0.1 | 59.45 | 367.47 | 8.29 | 8.512 | 221 | 16.8 |
| 147 | NZ-Kop | 2014 TS_3 | -0.2 | 62.93 | 372.48 | 8.55 | 7.792 | 226 | 16.34 |
| 148 | NZ-Kop | 2015 TS_1 | -0.5 | 56.49 | 371.82 | 7.73 | 9.355 | 214 | 17.09 |
| 149 | NZ-Kop | 2015 TS_2 | -0.1 | 58.32 | 374.74 | 7.87 | 9.063 | 217 | 16.93 |
| 150 | NZ-Kop | 2015 TS_3 | -0.2 | 62.67 | 378.44 | 8.19 | 8.217 | 222 | 16.4 |
| 151 | RU-Ch2 | 2014 TS_1 | -0.04 | 138.76 | 263.76 | -0.13 | 14.42 | 206.6 | 14.29 |
| 152 | RU-Ch2 | 2014 TS_2 | -0.08 | 146.08 | 262.11 | -0.14 | 13.03 | 206.5 | 12.9 |
| 153 | RU-Ch2 | 2014 TS_3 | -0.16 | 155.95 | 264.51 | -0.05 | 4.581 | 210.5 | 4.535 |
| 154 | RU-Ch2 | 2015 TS_1 | -0.04 | 143.88 | 269.56 | -0.14 | 13.97 | 193.3 | 13.83 |
| 155 | RU-Ch2 | 2015 TS_2 | -0.08 | 147.34 | 265.69 | -0.10 | 12.3 | 195.7 | 12.2 |
| 156 | RU-Ch2 | 2015 TS_3 | -0.16 | 159.93 | 266.60 | -0.04 | 3.995 | 205.2 | 3.96 |
| 157 | RU-Ch2 | 2016 TS_1 | -0.04 | 126.98 | 273.54 | -0.16 | 11.64 | 200.2 | 11.48 |
| 158 | RU-Ch2 | 2016 TS_2 | -0.08 | 133.92 | 272.58 | -0.10 | 10.06 | 203.2 | 9.964 |
| 159 | RU-Ch2 | 2016 TS_3 | -0.16 | 147.99 | 275.45 | -0.04 | 4.042 | 217.7 | 4.001 |
| 160 | RU-Che | 2014 TS_1 | -0.04 | 138.05 | 267.57 | -0.12 | 15.04 | 208 | 14.92 |
| 161 | RU-Che | 2014 TS_2 | -0.08 | 149.72 | 263.67 | -0.09 | 8.959 | 206.6 | 8.873 |
| 162 | RU-Che | 2014 TS_3 | -0.16 | 154.97 | 265.49 | -0.07 | 7.006 | 210.3 | 6.938 |
| 163 | RU-Che | 2015 TS_1 | -0.04 | 143.85 | 274.68 | -0.17 | 14.81 | 193.7 | 14.64 |
| 164 | RU-Che | 2015 TS_2 | -0.08 | 149.49 | 267.08 | -0.06 | 8.336 | 197.9 | 8.273 |
| 165 | RU-Che | 2015 TS_3 | -0.16 | 154.48 | 271.03 | -0.04 | 5.942 | 202.4 | 5.9 |
| 166 | RU-Che | 2016 TS_1 | -0.04 | 126.72 | 274.03 | -0.19 | 12.95 | 200.4 | 12.76 |



| # | Site | Year TS | | | | | | | |
|---|------|---------|------|--------|--------|------|-------|-------|-------|
| 167 | RU-Che | 2016 TS_2 | -0.08 | 137.01 | 273.70 | -0.07 | 7.076 | 205.4 | 7.01 |
| 168 | RU-Che | 2016 TS_3 | -0.16 | 142.51 | 275.62 | -0.05 | 5.498 | 211.8 | 5.451 |
| 169 | SE-Deg | 2014 TS_1 | -0.02 | 111.85 | 303.55 | -0.53 | 17.22 | 201.6 | 16.69 |
| 170 | SE-Deg | 2014 TS_2 | -0.05 | 119.11 | 308.12 | -0.31 | 13.23 | 207.4 | 12.93 |
| 171 | SE-Deg | 2014 TS_3 | -0.1 | 125.46 | 315.55 | -0.10 | 12.54 | 212 | 12.44 |
| 172 | SE-Deg | 2014 TS_4 | -0.15 | 134.61 | 321.20 | 0.29 | 11.63 | 215.6 | 11.93 |
| 173 | SE-Deg | 2014 TS_5 | -0.3 | 126.75 | 330.85 | 0.52 | 11.61 | 220 | 12.13 |
| 174 | SE-Deg | 2014 TS_6 | -0.5 | 130.62 | 341.66 | 0.89 | 11.33 | 223.1 | 12.21 |
| 175 | SE-Deg | 2015 TS_1 | -0.02 | 104.25 | 310.59 | -0.28 | 15.2 | 207.2 | 14.91 |
| 176 | SE-Deg | 2015 TS_2 | -0.05 | 110.64 | 312.57 | 0.09 | 13.86 | 209.5 | 13.95 |
| 177 | SE-Deg | 2015 TS_3 | -0.1 | 112.94 | 321.41 | 0.41 | 12.94 | 212.9 | 13.36 |
| 178 | SE-Deg | 2015 TS_4 | -0.15 | 115.72 | 329.21 | 0.60 | 11.94 | 216.6 | 12.54 |
| 179 | SE-Deg | 2015 TS_5 | -0.3 | 118.31 | 339.17 | 0.90 | 11.08 | 220.5 | 11.98 |
| 180 | SE-Deg | 2015 TS_6 | -0.5 | 121.80 | 347.97 | 1.30 | 10.19 | 224.6 | 11.48 |
| 181 | SE-Deg | 2016 TS_1 | -0.02 | 108.14 | 306.39 | -0.19 | 14.88 | 200.3 | 14.68 |
| 182 | SE-Deg | 2017 TS_1 | -0.02 | 133.38 | 326.87 | -0.20 | 12.35 | 215 | 12.15 |
| 183 | SE-Deg | 2018 TS_1 | -0.02 | 111.67 | 310.21 | -0.17 | 14.7 | 198 | 14.52 |
| 184 | US-Atq | 2014 TS_1 | NaN | 10.75 | 139.53 | -0.20 | 8.068 | 61 | 7.864 |
| 185 | US-Atq | 2014 TS_2 | NaN | 18.00 | 137.32 | -0.08 | 4.191 | 74.3 | 4.109 |
| 186 | US-Atq | 2014 TS_3 | NaN | 28.49 | 138.48 | -0.03 | 2.366 | 83.5 | 2.334 |
| 187 | US-Beo | 2014 TS_1 | NaN | 155.09 | 270.13 | -0.04 | 4.874 | 211.1 | 4.829 |
| 188 | US-Beo | 2014 TS_2 | NaN | 168.62 | 270.60 | -0.02 | 2.94 | 219.6 | 2.922 |
| 189 | US-Beo | 2014 TS_3 | NaN | 170.53 | 269.49 | -0.02 | 3.128 | 219.9 | 3.104 |
| 190 | US-Bes | 2013 TS_1 | NaN | 143.22 | 261.99 | -0.05 | 5.649 | 201.3 | 5.602 |
| 191 | US-Bes | 2013 TS_2 | NaN | 150.57 | 267.49 | -0.04 | 6.744 | 205.7 | 6.706 |
| 192 | US-Bes | 2013 TS_3 | NaN | 146.03 | 267.28 | -0.06 | 8.248 | 202.3 | 8.187 |
| 193 | US-Bes | 2014 TS_1 | NaN | 151.79 | 282.29 | -0.10 | 3.924 | 198.5 | 3.822 |
| 194 | US-Bes | 2015 TS_1 | NaN | 140.45 | 270.98 | -0.11 | 4.763 | 195.3 | 4.65 |
| 195 | US-Bes | 2015 TS_2 | NaN | 148.76 | 269.64 | -0.11 | 5.512 | 202.8 | 5.401 |
| 196 | US-Bes | 2015 TS_3 | NaN | 146.77 | 271.72 | -0.16 | 7.182 | 197.8 | 7.027 |
| 197 | US-BZB | 2014 TS_1 | -0.075 | 123.11 | 298.35 | -0.44 | 15.35 | 215.8 | 14.91 |
| 198 | US-BZB | 2014 TS_2 | -0.05 | 115.15 | 292.07 | -0.59 | 15.49 | 209.9 | 14.9 |
| 199 | US-BZB | 2015 TS_1 | -0.075 | 107.82 | 295.60 | -0.38 | 14.04 | 210.2 | 13.67 |
| 200 | US-BZB | 2015 TS_2 | -0.05 | 98.63 | 293.17 | -0.67 | 14.56 | 203.5 | 13.9 |

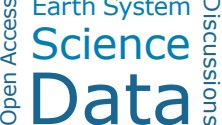

| # | Site | Year/TS | | | | | | | |
|---|------|---------|---|---|---|---|---|---|---|
| 201 | US-BZB | 2016 TS_1 | -0.075 | 125.09 | 292.39 | -0.33 | 16.39 | 214.3 | 16.06 |
| 202 | US-BZB | 2016 TS_2 | -0.05 | 109.89 | 290.95 | -0.74 | 16.89 | 211.7 | 16.15 |
| 203 | US-BZF | 2014 TS_1 | -0.075 | 96.05 | 322.79 | -1.56 | 16.12 | 205.4 | 14.56 |
| 204 | US-BZF | 2014 TS_2 | -0.05 | 112.68 | 322.83 | -1.22 | 15.9 | 208.3 | 14.68 |
| 205 | US-BZF | 2015 TS_1 | -0.075 | 108.39 | 331.07 | -1.20 | 14.88 | 197.5 | 13.68 |
| 206 | US-BZF | 2015 TS_2 | -0.05 | 111.46 | 336.09 | -1.12 | 14.83 | 200.2 | 13.72 |
| 207 | US-BZF | 2016 TS_1 | -0.075 | 95.50 | 315.89 | -1.03 | 17.18 | 205.1 | 16.15 |
| 208 | US-BZF | 2016 TS_2 | -0.05 | 100.12 | 317.57 | -0.90 | 17.72 | 206 | 16.81 |
| 209 | US-BZS | 2015 TS_1 | NaN | 116.48 | 275.22 | -0.07 | 4.866 | 202.9 | 4.792 |
| 210 | US-BZS | 2015 TS_2 | NaN | 97.29 | 283.75 | -0.27 | 8.135 | 187.7 | 7.862 |
| 211 | US-BZS | 2015 TS_3 | NaN | 105.93 | 272.47 | -0.13 | 10.61 | 193.1 | 10.47 |
| 212 | US-BZS | 2016 TS_1 | NaN | 119.07 | 278.90 | -0.05 | 5.584 | 208.4 | 5.535 |
| 213 | US-BZS | 2016 TS_2 | NaN | 87.59 | 278.96 | -0.25 | 12.09 | 203.1 | 11.84 |
| 214 | US-BZS | 2016 TS_3 | NaN | 98.29 | 277.92 | -0.16 | 11.67 | 198.8 | 11.51 |
| 215 | US-ICs | 2014 TS_1 | -0.075 | 147.27 | 263.08 | -0.08 | 5.467 | 205.6 | 5.39 |
| 216 | US-ICs | 2014 TS_2 | -0.05 | 147.13 | 262.66 | -0.01 | 6.41 | 205.1 | 6.402 |
| 217 | US-ICs | 2015 TS_1 | -0.075 | 141.61 | 255.70 | -0.02 | 6.12 | 195.2 | 6.098 |
| 218 | US-ICs | 2015 TS_2 | -0.05 | 140.38 | 256.88 | -0.02 | 7.07 | 193.1 | 7.047 |
| 219 | US-ICs | 2016 TS_1 | -0.075 | 146.75 | 265.86 | -0.05 | 5.67 | 206.2 | 5.623 |
| 220 | US-ICs | 2016 TS_2 | -0.05 | 146.79 | 265.67 | -0.05 | 6.654 | 206.1 | 6.599 |
| 221 | US-Ivo | 2014 TS_1 | -0.05 | 136.92 | 264.34 | -0.20 | 10.84 | 195.7 | 10.64 |
| 222 | US-Ivo | 2014 TS_1 | -0.4 | 136.92 | 264.34 | -0.20 | 10.84 | 195.7 | 10.64 |
| 223 | US-Ivo | 2014 TS_2 | -0.1 | 142.52 | 266.53 | -0.16 | 9.908 | 195.5 | 9.744 |
| 224 | US-Ivo | 2014 TS_3 | -0.15 | 144.98 | 262.74 | -0.19 | 6.761 | 204.8 | 6.574 |
| 225 | US-Ivo | 2014 TS_4 | -0.3 | 166.58 | 262.05 | -0.03 | 4.068 | 214.3 | 4.034 |
| 226 | US-Ivo | 2015 TS_1 | -0.05 | 139.42 | 257.06 | -0.14 | 14.86 | 185.6 | 14.72 |
| 227 | US-Ivo | 2015 TS_1 | -0.4 | 139.42 | 257.06 | -0.14 | 14.86 | 185.6 | 14.72 |
| 228 | US-Ivo | 2015 TS_2 | -0.1 | 141.20 | 256.70 | -0.09 | 11.99 | 189.1 | 11.9 |
| 229 | US-Ivo | 2015 TS_3 | -0.15 | 145.76 | 260.99 | -0.07 | 6.85 | 199.1 | 6.785 |
| 230 | US-Ivo | 2015 TS_4 | -0.3 | 158.88 | 259.63 | -0.03 | 4.032 | 208.2 | 3.997 |
| 231 | US-Ivo | 2016 TS_1 | -0.05 | 133.60 | 262.43 | -0.10 | 8.895 | 197.3 | 8.796 |
| 232 | US-Ivo | 2016 TS_1 | -0.4 | 133.60 | 262.43 | -0.10 | 8.895 | 197.3 | 8.796 |
| 233 | US-Ivo | 2016 TS_2 | -0.1 | 139.70 | 264.19 | -0.04 | 6.76 | 202.2 | 6.719 |
| 234 | US-Ivo | 2016 TS_3 | -0.15 | 153.78 | 269.47 | -0.05 | 4.305 | 211.5 | 4.253 |




| | Site | Year TS | | | | | | | |
|---|---|---|---|---|---|---|---|---|---|
| 235 | US-Ivo | 2016 TS_4 | -0.3 | 171.24 | 271.03 | -0.01 | 2.644 | 221.2 | 2.631 |
| 236 | US-LA1 | 2012 TS_1 | -0.1 | 29.15 | 331.44 | 15.59 | 13.5 | 197.2 | 29.08 |
| 237 | US-LA2 | 2012 TS_1 | -0.1 | 36.65 | 336.05 | 15.04 | 14.35 | 193.2 | 29.39 |
| 238 | US-LA2 | 2013 TS_1 | -0.1 | 65.79 | 377.93 | 14.70 | 16.06 | 201.5 | 30.76 |
| 239 | US-Los | 2014 TS_1 | 0 | 136.18 | 417.40 | 1.95 | 8.263 | 244.3 | 10.22 |
| 240 | US-Los | 2015 TS_1 | 0 | 148.23 | 422.27 | 2.43 | 7.761 | 258.7 | 10.19 |
| 241 | US-Los | 2016 TS_1 | 0 | 135.20 | 415.75 | 2.47 | 8.083 | 255.1 | 10.56 |
| 242 | US-Los | 2017 TS_1 | 0 | 141.15 | 414.62 | 1.89 | 7.451 | 256 | 9.343 |
| 243 | US-Los | 2018 TS_1 | 0 | 169.83 | 421.79 | 1.46 | 7.419 | 260 | 8.883 |
| 244 | US-Myb | 2011 TS_3 | -0.08 | NaN | 329.50 | 12.12 | 8.859 | 231.7 | 20.98 |
| 245 | US-Myb | 2011 TS_4 | -0.16 | NaN | 333.87 | 12.36 | 8.288 | 235.8 | 20.65 |
| 246 | US-Myb | 2011 TS_5 | -0.32 | NaN | 338.98 | 12.72 | 7.56 | 241.4 | 20.28 |
| 247 | US-Myb | 2011 TS_1 | -0.02 | NaN | 326.62 | 11.96 | 9.156 | 229.3 | 21.12 |
| 248 | US-Myb | 2012 TS_3 | -0.08 | 35.60 | 372.21 | 9.39 | 10.96 | 216.1 | 20.36 |
| 249 | US-Myb | 2012 TS_4 | -0.16 | 40.39 | 375.48 | 9.90 | 10.23 | 220.7 | 20.13 |
| 250 | US-Myb | 2012 TS_5 | -0.32 | 47.39 | 379.29 | 10.56 | 9.212 | 227.2 | 19.78 |
| 251 | US-Myb | 2012 TS_1 | -0.02 | 29.91 | 372.10 | 8.98 | 11.77 | 214.4 | 20.74 |
| 252 | US-Myb | 2013 TS_3 | -0.08 | 34.38 | 354.74 | 9.25 | 11.28 | 210.9 | 20.52 |
| 253 | US-Myb | 2013 TS_4 | -0.16 | 40.07 | 359.02 | 9.87 | 10.41 | 215 | 20.28 |
| 254 | US-Myb | 2013 TS_5 | -0.32 | 45.10 | 363.81 | 10.53 | 9.441 | 221.9 | 19.97 |
| 255 | US-Myb | 2013 TS_1 | -0.02 | NaN | 355.11 | 8.47 | 12.24 | 208.2 | 20.7 |
| 256 | US-Myb | 2014 TS_3 | -0.08 | 26.04 | 365.88 | 9.64 | 12.28 | 210 | 21.93 |
| 257 | US-Myb | 2014 TS_4 | -0.16 | 38.09 | 364.15 | 10.74 | 11.23 | 211 | 21.97 |
| 258 | US-Myb | 2014 TS_5 | -0.32 | 44.34 | 366.12 | 11.25 | 10.25 | 218 | 21.5 |
| 259 | US-Myb | 2015 TS_3 | -0.08 | 17.34 | 340.72 | 9.77 | 11.85 | 212 | 21.62 |
| 260 | US-Myb | 2015 TS_4 | -0.16 | 12.76 | 339.04 | 10.79 | 10.84 | 221 | 21.63 |
| 261 | US-Myb | 2015 TS_5 | -0.32 | 18.83 | 343.94 | 11.35 | 9.93 | 226 | 21.28 |
| 262 | US-Myb | 2016 TS_3 | -0.08 | 5.05 | 357.82 | 9.61 | 11.19 | 201 | 20.8 |
| 263 | US-Myb | 2016 TS_4 | -0.16 | 3.39 | 356.20 | 9.84 | 10.87 | 208 | 20.72 |
| 264 | US-Myb | 2016 TS_5 | -0.32 | 12.02 | 360.46 | 10.73 | 9.627 | 214 | 20.36 |
| 265 | US-Myb | 2017 TS_3 | -0.08 | 27.31 | 352.90 | 9.86 | 13.1 | 214 | 22.96 |
| 266 | US-Myb | 2017 TS_4 | -0.16 | 28.29 | 357.41 | 9.92 | 12.74 | 215 | 22.66 |
| 267 | US-Myb | 2017 TS_5 | -0.32 | 36.79 | 360.41 | 10.85 | 11.23 | 223 | 22.08 |
| 268 | US-Myb | 2017 TS_1 | -0.02 | 62.79 | 325.72 | 12.70 | 10.71 | 216.8 | 23.41 |





| | | | | | | | | | | |
|---|---|---|---|---|---|---|---|---|---|---|
| 269 | US-Myb | 2018 | TS_3 | -0.08 | 36.27 | 326.10 | 10.20 | 10.72 | 207 | 20.92 |
| 270 | US-Myb | 2018 | TS_4 | -0.16 | 41.30 | 332.83 | 10.47 | 10.35 | 209 | 20.82 |
| 271 | US-Myb | 2018 | TS_5 | -0.32 | 48.43 | 336.09 | 11.18 | 9.293 | 215 | 20.47 |
| 272 | US-Myb | 2018 | TS_1 | -0.02 | 38.38 | 344.86 | 10.05 | 11.93 | 200.3 | 21.98 |
| 273 | US-NC4 | 2012 | TS_1 | -0.05 | 42.43 | 336.03 | 7.87 | 15.75 | 215.4 | 23.62 |
| 274 | US-NC4 | 2013 | TS_1 | -0.05 | 59.96 | 368.85 | 6.92 | 16.83 | 210.5 | 23.74 |
| 275 | US-NC4 | 2014 | TS_1 | -0.05 | 54.42 | 362.34 | 6.73 | 16.81 | 208.5 | 23.54 |
| 276 | US-NC4 | 2015 | TS_1 | -0.05 | 68.13 | 387.31 | 8.27 | 16.06 | 205 | 24.33 |
| 277 | US-NC4 | 2016 | TS_1 | -0.05 | 52.97 | 351.38 | 9.41 | 15.04 | 220 | 24.45 |
| 278 | US-ORv | 2012 | TS | NaN | 57.42 | 352.55 | 4.36 | 20.61 | 203.9 | 24.97 |
| 279 | US-ORv | 2013 | TS | NaN | 63.67 | 356.74 | 2.93 | 19.98 | 210.8 | 22.9 |
| 280 | US-ORv | 2014 | TS | NaN | 68.11 | 364.96 | 2.11 | 20.17 | 205.3 | 22.28 |
| 281 | US-ORv | 2015 | TS | NaN | 68.77 | 387.78 | 1.77 | 21.26 | 206 | 23.04 |
| 282 | US-OWC | 2016 | TS_1 | -0.05 | 0.00 | 0.00 | 0.00 | 0 | 211.2 | 23.91 |
| 283 | US-Sne | 2017 | TS_1 | -0.01 | 46.07 | 337.92 | 10.33 | 13.14 | 212.7 | 23.47 |
| 284 | US-Sne | 2017 | TS_2 | -0.02 | 41.06 | 341.82 | 10.76 | 12.66 | 205.7 | 23.41 |
| 285 | US-Sne | 2017 | TS_3 | -0.08 | 42.89 | 343.85 | 11.04 | 12.17 | 208.5 | 23.22 |
| 286 | US-Sne | 2017 | TS_4 | -0.16 | 46.36 | 346.18 | 11.39 | 11.79 | 211 | 23.18 |
| 287 | US-Sne | 2017 | TS_5 | -0.32 | 50.59 | 350.46 | 11.98 | 10.7 | 216 | 22.67 |
| 288 | US-Sne | 2018 | TS_1 | -0.01 | 48.41 | 325.09 | 10.28 | 12.15 | 217.3 | 22.43 |
| 289 | US-Sne | 2018 | TS_2 | -0.02 | 33.91 | 331.01 | 10.55 | 11.34 | 210.2 | 21.89 |
| 290 | US-Sne | 2018 | TS_3 | -0.08 | 36.76 | 331.50 | 10.87 | 10.63 | 212.1 | 21.5 |
| 291 | US-Sne | 2018 | TS_4 | -0.16 | 35.64 | 335.45 | 11.23 | 10.09 | 220.1 | 21.32 |
| 292 | US-Sne | 2018 | TS_5 | -0.32 | 49.09 | 335.94 | 11.88 | 9.045 | 218 | 20.92 |
| 293 | US-Srr | 2016 | TS_1 | NaN | NaN | 326.29 | 10.03 | 10.71 | 200.5 | 20.74 |
| 294 | US-Srr | 2017 | TS_1 | NaN | 11.34 | 346.85 | 7.22 | 13.71 | 199.5 | 20.93 |
| 295 | US-StJ | 2016 | TS_2 | -0.05 | 68.37 | 347.38 | 4.05 | 16.22 | 213.7 | 20.27 |
| 296 | US-StJ | 2016 | TS_3 | -0.1 | 68.38 | 347.38 | 5.84 | 14.1 | 213.7 | 19.94 |
| 297 | US-Tw1 | 2012 | TS_1 | -0.02 | 50.54 | 359.96 | 5.99 | 11.52 | 225.2 | 17.51 |
| 298 | US-Tw1 | 2012 | TS_2 | -0.04 | 48.02 | 358.76 | 6.14 | 11.32 | 227 | 17.46 |
| 299 | US-Tw1 | 2012 | TS_3 | -0.08 | 50.07 | 367.92 | 5.18 | 12.31 | 222.1 | 17.49 |
| 300 | US-Tw1 | 2012 | TS_4 | -0.16 | 49.05 | 367.52 | 5.31 | 12.19 | 224.5 | 17.5 |
| 301 | US-Tw1 | 2012 | TS_5 | -0.32 | -79.10 | 347.23 | 7.89 | 9.513 | 225.3 | 17.4 |
| 302 | US-Tw1 | 2013 | TS_1 | -0.02 | 35.80 | 337.57 | 4.39 | 14.59 | 206.6 | 18.97 |





| | | | | | | | | | |
|---|---|---|---|---|---|---|---|---|---|
| 303 | US-Tw1 | 2013 TS_2 | -0.04 | 36.08 | 337.66 | 4.39 | 14.57 | 206.8 | 18.96 |
| 304 | US-Tw1 | 2013 TS_3 | -0.08 | 36.81 | 338.09 | 4.40 | 14.54 | 207.5 | 18.94 |
| 305 | US-Tw1 | 2013 TS_4 | -0.16 | 37.57 | 338.82 | 4.41 | 14.59 | 208.4 | 19.01 |
| 306 | US-Tw1 | 2013 TS_5 | -0.32 | 38.64 | 340.06 | 4.46 | 14.62 | 209.6 | 19.07 |
| 307 | US-Tw1 | 2014 TS_1 | -0.02 | 41.23 | 395.41 | 6.67 | 10.89 | 208.3 | 17.56 |
| 308 | US-Tw1 | 2014 TS_2 | -0.04 | 41.86 | 397.14 | 6.70 | 10.85 | 208.6 | 17.55 |
| 309 | US-Tw1 | 2014 TS_3 | -0.08 | 43.55 | 400.01 | 6.78 | 10.72 | 209.8 | 17.49 |
| 310 | US-Tw1 | 2014 TS_4 | -0.16 | 45.61 | 404.82 | 6.88 | 10.56 | 211.3 | 17.44 |
| 311 | US-Tw1 | 2014 TS_5 | -0.32 | 49.62 | 416.29 | 7.09 | 10.24 | 213.9 | 17.33 |
| 312 | US-Tw1 | 2015 TS_1 | -0.02 | 50.66 | 342.55 | 9.02 | 7.831 | 235 | 16.85 |
| 313 | US-Tw1 | 2015 TS_2 | -0.04 | 52.00 | 342.38 | 9.09 | 7.706 | 235 | 16.8 |
| 314 | US-Tw1 | 2015 TS_3 | -0.08 | 55.61 | 341.98 | 9.26 | 7.385 | 238 | 16.64 |
| 315 | US-Tw1 | 2015 TS_4 | -0.16 | 59.57 | 342.95 | 9.51 | 6.972 | 240 | 16.48 |
| 316 | US-Tw1 | 2015 TS_5 | -0.32 | 69.39 | 345.18 | 10.00 | 6.208 | 246 | 16.21 |
| 317 | US-Tw1 | 2016 TS_1 | -0.02 | 34.57 | 361.06 | 8.78 | 7.943 | 218 | 16.72 |
| 318 | US-Tw1 | 2016 TS_2 | -0.04 | 35.49 | 362.19 | 8.86 | 7.837 | 219 | 16.69 |
| 319 | US-Tw1 | 2016 TS_3 | -0.08 | 38.58 | 363.10 | 9.04 | 7.546 | 221 | 16.59 |
| 320 | US-Tw1 | 2016 TS_4 | -0.16 | 44.73 | 365.88 | 9.33 | 7.14 | 223 | 16.47 |
| 321 | US-Tw1 | 2016 TS_5 | -0.32 | 56.22 | 370.54 | 9.86 | 6.35 | 229 | 16.21 |
| 322 | US-Tw1 | 2017 TS_1 | -0.02 | 41.17 | 343.75 | 7.55 | 10.22 | 228 | 17.77 |
| 323 | US-Tw1 | 2017 TS_2 | -0.04 | 41.75 | 344.56 | 7.61 | 10.11 | 229 | 17.72 |
| 324 | US-Tw1 | 2017 TS_3 | -0.08 | 42.61 | 345.02 | 7.75 | 9.803 | 231 | 17.55 |
| 325 | US-Tw1 | 2017 TS_4 | -0.16 | 45.05 | 347.17 | 7.97 | 9.368 | 234 | 17.34 |
| 326 | US-Tw1 | 2017 TS_5 | -0.32 | 48.82 | 349.23 | 8.38 | 8.5 | 240 | 16.88 |
| 327 | US-Tw1 | 2018 TS_1 | -0.02 | 60.47 | 327.43 | 6.70 | 10.42 | 222 | 17.12 |
| 328 | US-Tw1 | 2018 TS_2 | -0.04 | 61.54 | 327.44 | 6.75 | 10.36 | 223 | 17.1 |
| 329 | US-Tw1 | 2018 TS_3 | -0.08 | 64.60 | 328.44 | 6.85 | 10.18 | 225 | 17.03 |
| 330 | US-Tw1 | 2018 TS_4 | -0.16 | 67.67 | 329.40 | 7.01 | 9.925 | 227 | 16.93 |
| 331 | US-Tw1 | 2018 TS_5 | -0.32 | 75.09 | 331.40 | 7.31 | 9.459 | 230 | 16.77 |
| 332 | US-Tw4 | 2015 TS_1 | -0.02 | 15.93 | 327.22 | 10.04 | 11.56 | 199.4 | 21.6 |
| 333 | US-Tw4 | 2015 TS_3 | -0.08 | 20.21 | 329.78 | 10.48 | 10.96 | 202.3 | 21.44 |
| 334 | US-Tw4 | 2015 TS_4 | -0.16 | 23.88 | 332.52 | 10.86 | 10.42 | 205.7 | 21.28 |
| 335 | US-Tw4 | 2015 TS_5 | -0.32 | 29.16 | 338.59 | 11.49 | 9.449 | 212.4 | 20.94 |
| 336 | US-Tw4 | 2016 TS_1 | -0.02 | 9.56 | 358.75 | 8.16 | 11.29 | 201.7 | 19.45 |





| | Site | Year | TS | | | | | | | |
|---|---|---|---|---|---|---|---|---|---|---|
| 337 | US-Tw4 | 2016 | TS_3 | -0.08 | 13.17 | 360.06 | 8.67 | 10.58 | 205.7 | 19.25 |
| 338 | US-Tw4 | 2016 | TS_4 | -0.16 | 15.94 | 362.56 | 9.11 | 9.991 | 209.2 | 19.11 |
| 339 | US-Tw4 | 2016 | TS_5 | -0.32 | 21.05 | 367.55 | 9.91 | 8.876 | 216.2 | 18.78 |
| 340 | US-Tw4 | 2017 | TS_1 | -0.02 | 38.35 | 347.31 | 8.07 | 11.52 | 211.9 | 19.59 |
| 341 | US-Tw4 | 2017 | TS_3 | -0.08 | 42.12 | 351.81 | 8.45 | 10.91 | 215.3 | 19.36 |
| 342 | US-Tw4 | 2017 | TS_4 | -0.16 | 46.03 | 354.70 | 8.83 | 10.35 | 218.6 | 19.18 |
| 343 | US-Tw4 | 2017 | TS_5 | -0.32 | 53.54 | 361.37 | 9.52 | 9.275 | 224.9 | 18.79 |
| 344 | US-Tw4 | 2018 | TS_1 | -0.02 | 58.11 | 344.87 | 8.00 | 10.93 | 218 | 18.93 |
| 345 | US-Tw4 | 2018 | TS_3 | -0.08 | 63.91 | 349.15 | 8.38 | 10.41 | 222 | 18.79 |
| 346 | US-Tw4 | 2018 | TS_4 | -0.16 | 67.95 | 352.10 | 8.71 | 9.862 | 225 | 18.57 |
| 347 | US-Tw4 | 2018 | TS_5 | -0.32 | 75.31 | 357.88 | 9.33 | 8.8 | 231 | 18.13 |
| 348 | US-Tw5 | 2018 | TS_1 | -0.02 | NaN | 414.83 | 0.00 | 8.894 | 222.2 | 18.37 |
| 349 | US-Tw5 | 2018 | TS_2 | -0.1 | NaN | 401.00 | 0.00 | 12.32 | 204.4 | 22.24 |
| 350 | US-Tw5 | 2018 | TS_3 | -0.02 | NaN | 414.83 | 0.00 | 8.894 | 222.2 | 18.37 |
| 351 | US-Tw5 | 2018 | TS_4 | -0.08 | NaN | 423.43 | 0.00 | 7.898 | 227.3 | 18.14 |
| 352 | US-Tw5 | 2018 | TS_5 | -0.16 | NaN | 430.15 | 0.00 | 7.531 | 230 | 17.94 |
| 353 | US-Uaf | 2011 | TS_1 | -0.09 | 86.20 | 372.46 | -12.29 | 21.95 | 199.6 | 9.667 |
| 354 | US-Uaf | 2012 | TS_1 | -0.09 | 73.77 | 338.53 | -11.83 | 20.86 | 202.4 | 9.028 |
| 355 | US-Uaf | 2013 | TS_1 | -0.09 | 109.63 | 395.51 | -10.08 | 20.52 | 200.4 | 10.44 |
| 356 | US-Uaf | 2014 | TS_1 | -0.09 | 76.07 | 365.40 | -10.94 | 19.94 | 206 | 8.999 |
| 357 | US-Uaf | 2015 | TS_1 | -0.09 | 80.99 | 423.19 | -9.77 | 19.76 | 190 | 9.998 |
| 358 | US-Uaf | 2016 | TS_1 | -0.09 | 77.38 | 315.75 | -7.74 | 19.13 | 198 | 11.39 |
| 359 | US-Uaf | 2017 | TS_1 | -0.09 | 84.88 | 380.17 | -7.39 | 19.08 | 196 | 11.69 |
| 360 | US-Uaf | 2018 | TS_1 | -0.09 | 96.04 | 333.33 | -5.60 | 17.72 | 199 | 12.11 |
| 361 | US-WPT | 2011 | TS_1 | -0.1 | 80.95 | 342.17 | 5.27 | 19.27 | 202.6 | 24.54 |
| 362 | US-WPT | 2011 | TS_2 | -0.3 | NaN | 347.04 | 6.38 | 16.62 | 209.7 | 23.01 |
| 363 | US-WPT | 2012 | TS_1 | -0.1 | 40.29 | 345.57 | 3.70 | 21.6 | 197.4 | 25.3 |
| 364 | US-WPT | 2012 | TS_2 | -0.3 | 44.91 | 354.96 | 4.52 | 19.21 | 203.8 | 23.72 |
| 365 | US-WPT | 2013 | TS_1 | -0.1 | 74.61 | 340.47 | 3.73 | 18.23 | 207.2 | 21.96 |
| 366 | US-WPT | 2013 | TS_2 | -0.3 | 77.62 | 352.35 | 4.32 | 16.69 | 211.7 | 21.01 |

**Column Descriptions**

| | |
|---|---|
| Site | Site name |

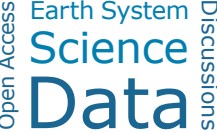

| Field | Description |
| --- | --- |
| Year | Data year |
| Probe_name | Temperature probe name as given in data files |
| Soil_temp_depth_m | Depth of soil temperature probe (m), with negative values |
| Start_TS_(DOY) | Season start for elevated TS (DOY), point "f" in Figure 1 |
| End_TS_(DOY) | Season end for elevated TS (DOY), point "h" in Figure 1 |
| Base_value_TS_(C) | Baseline TS during non-elevated season (C), average of points |
| Ampl_TS_(C) | Amplitude of TS during elevated temperature season (C), difference between point "e" in Figure 1 and Base_value_TS |
| Peak_TS_(DOY) | Day of maximum elevated TS (DOY), point "g" in Figure 1 |
| Peak_value_TS_(C) | Maximum value of TS (C) point "e" in Figure 1 |



**Table B6 - Soil temperature probe depths (m)**

| | Site | Year | Probe name | Soil_temp_depth_m | Additional_notes |
|---|---|---|---|---|---|
| 1 | AT-NEU | | TS_1 | -0.05 | |
| 2 | AT-NEU | | TS_2 | -0.1 | |
| 3 | AT-NEU | | TS_3 | -0.2 | |
| 4 | BR-NPW | | TS_1 | | |
| 5 | BR-NPW | | TS_2 | | |
| 6 | BW-GUM | | no data | | |
| 7 | BW-NXR | | no data | | |
| 8 | CA-SCB | | TS_1 | 0 | |
| 9 | CA-SCB | | TS_2 | -0.02 | |
| 10 | CA-SCB | | TS_3 | -0.04 | |
| 11 | CA-SCB | | TS_4 | -0.08 | |
| 12 | CA-SCB | | TS_5 | -0.16 | |
| 13 | CA-SCB | | TS_6 | -0.32 | |
| 14 | CA-SCB | | TS_7 | -0.64 | |
| 15 | CA-SCB | | TS_8 | -1.28 | |
| 16 | CA-SCC | | TS_1 | -0.1 | |
| 17 | CA-SCC | | TS_2 | -0.15 | |
| 18 | CA-SCC | | TS_3 | -0.2 | |
| 19 | CA-SCC | | TS_4 | -0.25 | |
| 20 | CA-SCC | | TS_5 | -0.3 | |
| 21 | CA-SCC | | TS_6 | -0.5 | |
| 22 | CA-SCC | | TS_7 | -0.6 | |
| 23 | CA-SCC | | TS_8 | -0.7 | |
| 24 | CH-CHA | | TS_1 | -0.01 | |
| 25 | CH-CHA | | TS_2 | -0.02 | |
| 26 | CH-CHA | | TS_3 | -0.04 | |
| 27 | CH-CHA | | TS_4 | -0.07 | |
| 28 | CH-CHA | | TS_5 | -0.1 | |
| 29 | CH-CHA | | TS_6 | -0.15 | |
| 30 | CH-CHA | | TS_7 | -0.25 | |
| 31 | CH-CHA | | TS_8 | -0.4 | |
| 32 | CH-CHA | | TS_9 | -0.95 | |
| 33 | CH-DAV | | TS_1 | -0.05 | |
| 34 | CH-DAV | | TS_2 | -0.15 | |
| 35 | CH-DAV | | TS_3 | -0.5 | |
| 36 | CH-DAV | | TS_4 | - | |
| 37 | CH-DAV | | TS_5 | - | |
| 38 | CH-DAV | | TS_6 | - | |
| 39 | CH-OE2 | | TS_1 | -0.05 | |
| 40 | CH-OE2 | | TS_2 | -0.1 | |
| 41 | CH-OE2 | | TS_3 | -0.15 | |
| 42 | CH-OE2 | | TS_4 | - | |





| | | | |
|---|---|---|---|
| 43 | CH-OE2 | TS_5 | -0.3 |
| 44 | CH-OE2 | TS_6 | -0.5 |
| 45 | CH-OE2 | TS_7 | - |
| 46 | CN-HGU | TS | |
| 47 | DE-DGW | no data | |
| 48 | DE-HTE | TS_1 | 0 |
| 49 | DE-HTE | TS_2 | -0.1 |
| 50 | DE-HTE | TS_3 | -0.2 |
| 51 | DE-SFN | TS_1 | -0.02 |
| 52 | DE-SFN | TS_3 | -0.1 |
| 53 | DE-SFN | TS_4 | -0.2 |
| 54 | DE-SFN | TS_5 | -0.5 |
| 55 | DE-ZRK | TS_1 | -0.05 |
| 56 | DE-ZRK | TS_2 | -0.1 |
| 57 | DE-ZRK | TS_3 | -0.2 |
| 58 | DE-ZRK | TS_4 | -0.3 |
| 59 | DE-ZRK | TS_5 | -0.5 |
| 60 | FI-HYY | TS_1 | -0.02 |
| 61 | FI-HYY | TS_2 | -0.04 |
| 62 | FI-HYY | TS_3 | -0.12 |
| 63 | FI-HYY | TS_4 | -0.25 |
| 64 | FI-HYY | TS_5 | -0.5 |
| 65 | FI-LOM | TS_1 | -0.07 |
| 66 | FI-LOM | TS_2 | -0.3 |
| 67 | FI-LOM | TS_3 | -0.5 |
| 68 | FI-SI2 | TS_1 | -0.05 |
| 69 | FI-SI2 | TS_2 | -0.2 |
| 70 | FI-SI2 | TS_3 | -0.35 |
| 71 | FI-SI2 | TS_4 | -0.5 |
| 72 | FI-SII | pre 2016 | TS_1 | -0.05 |
| 73 | FI-SII | pre 2016 | TS_2 | -0.2 |
| 74 | FI-SII | pre 2016 | TS_3 | -0.35 |
| 75 | FI-SII | pre 2016 | TS_4 | -0.5 |
| 76 | FI-SII | after 2017 | TS_1 | 0 |
| 77 | FI-SII | after 2017 | TS_2 | -0.5 |
| 78 | FI-SII | after 2017 | TS_3 | -0.1 |
| 79 | FI-SII | after 2017 | TS_4 | -0.15 |
| 80 | FI-SII | after 2017 | TS_5 | -0.25 |
| 81 | FI-SII | after 2017 | TS_6 | -0.45 |
| 82 | FI-SII | after 2017 | TS_7 | -0.95 |
| 83 | FR-LGT | TS_1 | -0.02 |
| 84 | FR-LGT | TS_2 | -0.05 |
| 85 | FR-LGT | TS_3 | -0.1 |
| 86 | FR-LGT | TS_4 | -0.2 |



| | | | |
|---|---|---|---|
| 87 | FR-LGT | TS_5 | -0.4 |
| 88 | HK-MPM | TS_1 | |
| 89 | HK-MPM | TS_2 | |
| 90 | HK-MPM | TS_3 | |
| 91 | ID-PAG | TS_1 | -0.05 |
| 92 | IT-BCI | TS_1 | -0.05 |
| 93 | IT-BCI | TS_2 | -0.1 |
| 94 | IT-BCI | TS_3 | -0.3 |
| 95 | IT-BCI | TS_4 | -0.5 |
| 96 | IT-BCI | TS_5 | -1 |
| 97 | IT-CAS | TS_1 | -0.035 |
| 98 | IT-CAS | TS_2 | -0.075 |
| 99 | IT-CAS | TS_3 | -0.15 |
| 100 | JP-BBY | TS_1 | -0.183 |
| 101 | JP-BBY | TS_2 | -0.233 |
| 102 | JP-BBY | TS_3 | -0.283 |
| 103 | JP-BBY | TS_4 | -0.383 |
| 104 | JP-BBY | TS_5 | -0.483 |
| 105 | JP-MSE | TS_1 | -0.01 |
| 106 | JP-MSE | TS_2 | -0.025 |
| 107 | JP-MSE | TS_3 | -0.05 |
| 108 | JP-MSE | TS_4 | -0.1 |
| 109 | JP-MSE | TS_5 | -0.2 |
| 110 | JP-MSE | TS_6 | -0.4 |
| 111 | JP-SWL | no data | |
| 112 | KR-CRK | TS_1 | -0.05 |
| 113 | KR-CRK | TS_2 | -0.15 |
| 114 | MAERC | TS | |
| 115 | MY-MLM | TS_1 | -0.05 |
| 116 | NL-HOR | TS_1 | -0.01 |
| 117 | NL-HOR | TS_2 | -0.02 |
| 118 | NL-HOR | TS_3 | -0.04 |
| 119 | NL-HOR | TS_4 | -0.05 |
| 120 | NL-HOR | TS_5 | -0.1 |
| 121 | NL-HOR | TS_6 | -0.15 |
| 122 | NL-HOR | TS_7 | -0.25 |
| 123 | NL-HOR | TS_8 | -0.4 |
| 124 | NL-HOR | TS_9 | -0.6 |
| 125 | NZ-KOP | TS_1 | -0.5 |
| 126 | NZ-KOP | TS_2 | -0.1 |
| 127 | NZ-KOP | TS_3 | -0.2 |
| 128 | PH-RIF | TS_1 | |
| 129 | RU-CH2 | TS_1 | -0.04 |
| 130 | RU-CH2 | TS_2 | -0.08 |



| | | | |
|---|---|---|---|
| 131 | RU-CH2 | TS_3 | -0.16 |
| 132 | RU-CHE | TS_1 | -0.04 |
| 133 | RU-CHE | TS_2 | -0.08 |
| 134 | RU-CHE | TS_3 | -0.16 |
| 135 | RU-COK | no data | |
| 136 | RU-FY2 | TS_1 | |
| 137 | RU-FY2 | TS_2 | |
| 138 | RU-FY2 | TS_3 | |
| 139 | RU-FY2 | TS_4 | |
| 140 | RU-FY2 | TS_5 | |
| 141 | SE-DEG | TS_1 | -0.02 |
| 142 | SE-DEG | TS_2 | -0.05 |
| 143 | SE-DEG | TS_3 | -0.1 |
| 144 | SE-DEG | TS_4 | -0.15 |
| 145 | SE-DEG | TS_5 | -0.3 |
| 146 | SE-DEG | TS_6 | -0.5 |
| 147 | UK-LBT | no data | |
| 148 | US-A03 | TS_1 | -0.025 |
| 149 | US-A03 | TS_2 | -0.1 |
| 150 | US-A03 | TS_3 | -0.3 |
| 151 | US-A10 | TS_1 | -0.025 |
| 152 | US-A10 | TS_2 | -0.1 |
| 153 | US-A10 | TS_3 | -0.3 |
| 154 | US-ATQ | TS_1 | |
| 155 | US-ATQ | TS_2 | |
| 156 | US-ATQ | TS_3 | |
| 157 | US-BEO | TS_1 | |
| 158 | US-BEO | TS_2 | |
| 159 | US-BEO | TS_3 | |
| 160 | US-BES | TS_1 | |
| 161 | US-BES | TS_2 | |
| 162 | US-BES | TS_3 | |
| 163 | US-BI1 | TS_1 | -0.02 |
| 164 | US-BI1 | TS_2 | -0.04 |
| 165 | US-BI1 | TS_3 | -0.08 |
| 166 | US-BI1 | TS_4 | -0.16 |
| 167 | US-BI1 | TS_5 | -0.32 |
| 168 | US-BI2 | TS_1 | -0.02 |
| 169 | US-BI2 | TS_2 | -0.04 |
| 170 | US-BI2 | TS_3 | -0.08 |
| 171 | US-BI2 | TS_4 | -0.16 |
| 172 | US-BI2 | TS_5 | -0.32 |
| 173 | US-BRW | | |
| 174 | US-BZB | TS_1 | -0.075 |





| | | | |
|---|---|---|---|
| **175** | US-BZB | TS_2 | -0.05 |
| **176** | US-BZF | TS_1 | -0.075 |
| **177** | US-BZF | TS_2 | -0.05 |
| **178** | US-BZS | TS_1 | |
| **179** | US-BZS | TS_2 | |
| **180** | US-BZS | TS_3 | |
| **181** | US-CRT | TS_1 | |
| **182** | US-DPW | no data | |
| **183** | US-EDN | TS_1 | -0.25 |
| **184** | US-EDN | TS_2 | -0.15 |
| **185** | US-EDN | TS_3 | -0.05 |
| **186** | US-EDN | TS_4 | 0 |
| **187** | US-EDN | TS_5 | 0.05 |
| **188** | US-EDN | TS_6 | 0.1 |
| **189** | US-EDN | TS_7 | 0.2 |
| **190** | US-EDN | TS_8 | 0.3 |
| **191** | US-EML | TS_1 | -0.05 |
| **192** | US-EML | TS_2 | -0.1 |
| **193** | US-EML | TS_3 | -0.2 |
| **194** | US-EML | TS_4 | -0.4 |
| **195** | US-HO1 | TS_1 | -0.05 |
| **196** | US-HO1 | TS_2 | -0.1 |
| **197** | US-HRA | no data | -0.02 |
| **198** | US-HRC | no data | -0.02 |
| **199** | US-ICS | TS_1 | -0.075 |
| **200** | US-ICS | TS_2 | -0.05 |
| **201** | US-IVO | TS_1 | -0.05 |
| **202** | US-IVO | TS_2 | -0.1 |
| **203** | US-IVO | TS_3 | -0.15 |
| **204** | US-IVO | TS_4 | -0.3 |
| **205** | US-IVO | TS_5 | -0.4 |
| **206** | US-LA1 | TS | -0.1 |
| **207** | US-LA2 | TS | -0.1 |
| **208** | US-LOS | TS_1 | 0 |
| **209** | US-LOS | TS_2 | -0.05 |
| **210** | US-LOS | TS_3 | -0.1 |
| **211** | US-LOS | TS_4 | -0.2 |
| **212** | US-LOS | TS_5 | -0.5 |
| **213** | US-MRM | TS_1 | |
| **214** | US-MRM | TS_2 | |
| **215** | US-MYB | TS_1 | -0.02 |
| **216** | US-MYB | TS_2 | -0.04 |
| **217** | US-MYB | TS_3 | -0.08 |
| **218** | US-MYB | TS_4 | -0.16 |





| | | | |
|---|---|---|---|
| **219** | US-MYB | TS_5 | -0.32 |
| **220** | US-NC4 | TS_1 | -0.05 |
| **221** | US-NC4 | TS_2 | -0.2 |
| **222** | US-NGB | no data | |
| **223** | US-NGC | no data | |
| **224** | US-ORV | TS_1 | -0.08 |
| **225** | US-OWC | TS_1 | -0.05 |
| **226** | US-OWC | TS_2 | -0.3 |
| **227** | US-PFA | | |
| **228** | US-SND | TS_1 | -0.08 |
| **229** | US-SND | TS_2 | -0.16 |
| **230** | US-SND | TS_3 | |
| **231** | US-SND | TS_4 | |
| **232** | US-SND | TS_5 | |
| **233** | US-SND | TS_6 | |
| **234** | US-SNE | TS_1 | -0.01 |
| **235** | US-SNE | TS_2 | -0.02 |
| **236** | US-SNE | TS_3 | -0.08 |
| **237** | US-SNE | TS_4 | -0.16 |
| **238** | US-SNE | TS_5 | -0.32 |
| **239** | US-SRR | TS_1 | |
| **240** | US-SRR | TS_2 | |
| **241** | US-SRR | TS_3 | |
| **242** | US-SRR | TS_4 | |
| **243** | US-SRR | TS_5 | |
| **244** | US-STJ | TS_2 | -0.05 |
| **245** | US-STJ | TS_3 | -0.1 |
| **246** | US-TW1 | TS_1 | -0.02 |
| **247** | US-TW1 | TS_2 | -0.04 |
| **248** | US-TW1 | TS_3 | -0.08 |
| **249** | US-TW1 | TS_4 | -0.16 |
| **250** | US-TW1 | TS_5 | -0.32 |
| **251** | US-TW3 | TS_1 | -0.02 |
| **252** | US-TW3 | TS_2 | -0.04 |
| **253** | US-TW3 | TS_3 | -0.08 |
| **254** | US-TW3 | TS_4 | -0.16 |
| **255** | US-TW3 | TS_5 | -0.32 |
| **256** | US-TW4 | TS_1 | -0.02 |
| **257** | US-TW4 | TS_2 | -0.04 |
| **258** | US-TW4 | TS_3 | -0.08 |
| **259** | US-TW4 | TS_4 | -0.16 |
| **260** | US-TW4 | TS_5 | -0.32 |
| **261** | US-TW5 | TS_1 | -0.02 |
| **262** | US-TW5 | TS_2 | -0.1 |





| | Site | Probe_name | Soil_temp_depth_m | Additional_notes |
|---|---|---|---|---|
| **263** | US-TW5 | TS_3 | -0.02 | |
| **264** | US-TW5 | TS_4 | -0.08 | |
| **265** | US-TW5 | TS_5 | -0.16 | |
| **266** | US-TWT | TS_1 | -0.02 | |
| **267** | US-TWT | TS_2 | -0.04 | |
| **268** | US-TWT | TS_3 | -0.08 | |
| **269** | US-TWT | TS_4 | -0.16 | |
| **270** | US-TWT | TS_5 | -0.32 | |
| **271** | US-UAF | TS_1 | -0.09 | average of 3 depths: -0.15, -0.02, -0.1 |
| **272** | US-UAF | TS_2 | -0.1833333 | average of 3 depths: -0.3, -0.05, -0.2 |
| **273** | US-UAF | TS_3 | -0.2833333 | average of 3 depths: -0.45, -0.1, -0.3 |
| **274** | US-UAF | TS_4 | -0.3666667 | average of 3 depths: -0.5, -0.2, -0.4 |
| **275** | US-UAF | TS_5 | -0.5 | average of 2 depths: -0.7, -0.3 |
| **276** | US-UAF | TS_6 | -0.6 | average of 2 depths: -0.8, -0.4 |
| **277** | US-UAF | TS_7 | -0.75 | average of 2 depths: -1, -0.5 |
| **278** | US-UAF | TS_8 | -0.925 | average of 2 depths: -1.25, 0.6, |
| **279** | US-UAF | TS_9 | -1 | |
| **280** | US-WPT | TS_1 | -0.1 | |
| **281** | US-WPT | TS_2 | -0.3 | |

**Column Descriptions**

| | |
|---|---|
| Site | Site name |
| Year | When relevant, information about time-span of probe location. If blank, assume probe depth was constant |
| Probe_name | Temperature probe name as given in data files |
| Soil_temp_depth_m | Depth of soil temperature probe (m), with negative values being under the surface |
| Additional_notes | When relevant, additional information about site |



**Table B7 Site year annual sums and uncertainties**

| | Site | Year | Ann_Flux_g_C_m-2 | Ann_Flux_Uncertainty_g_C_m-2 | Mean_Soil_Temp_C | Mean_Water_Table_Depth_m |
|---|---|---|---|---|---|---|
| 1 | ATNEU | 2010 | 0.38 | 0.03 | 8.65 | NaN |
| 2 | ATNEU | 2011 | 0.25 | 0.02 | 8.61 | NaN |
| 3 | ATNEU | 2012 | NaN | NaN | 9.39 | NaN |
| 4 | BRNPW | 2013 | NaN | NaN | NaN | NaN |
| 5 | BRNPW | 2014 | NaN | NaN | 25.95 | NaN |
| 6 | BRNPW | 2015 | 20.95 | 1.18 | 26.2 | -0.47 |
| 7 | BRNPW | 2016 | 17.48 | 1.14 | 25.31 | -0.41 |
| 8 | BWGUM | 2018 | 51.73 | 10.59 | NaN | NaN |
| 9 | BWNXR | 2018 | 47.32 | 3.70 | NaN | NaN |
| 10 | CASCB | 2014 | 10.42 | 0.66 | 9.6 | -0.15 |
| 11 | CASCB | 2015 | NaN | NaN | 5.58 | -0.1 |
| 12 | CASCB | 2016 | 12.12 | 0.31 | 5.38 | -0.15 |
| 13 | CASCB | 2017 | 9.48 | 0.27 | 6.32 | -0.21 |
| 14 | CASCC | 2013 | NaN | NaN | 7.2 | NaN |
| 15 | CASCC | 2014 | 4.94 | 0.12 | 4.38 | NaN |
| 16 | CASCC | 2015 | 6.76 | 0.15 | 3.15 | NaN |
| 17 | CASCC | 2016 | 6.76 | 0.12 | NaN | NaN |
| 18 | CHCHA | 2012 | 2.13 | 0.38 | 11.88 | NaN |
| 19 | CHCHA | 2013 | 2.30 | 0.36 | 10.89 | NaN |
| 20 | CHCHA | 2014 | 3.46 | 0.40 | 12.2 | NaN |
| 21 | CHCHA | 2015 | 3.93 | 0.68 | 11.93 | NaN |
| 22 | CHCHA | 2016 | NaN | NaN | 12.28 | NaN |
| 23 | CHDAV | 2016 | 1.21 | 0.40 | 4.33 | NaN |
| 24 | CHDAV | 2017 | NaN | NaN | 4.41 | NaN |
| 25 | CHOE2 | 2018 | 0.29 | 0.13 | 12.32 | NaN |
| 26 | CNHGU | 2015 | NaN | NaN | NaN | NaN |
| 27 | CNHGU | 2016 | 0.81 | 0.16 | 7.26 | NaN |
| 28 | CNHGU | 2017 | 0.82 | 0.45 | 7.66 | NaN |
| 29 | DEDGW | 2015 | NaN | NaN | NaN | NaN |
| 30 | DEDGW | 2016 | 7.51 | 0.22 | NaN | NaN |
| 31 | DEDGW | 2017 | 10.42 | 0.16 | NaN | NaN |
| 32 | DEDGW | 2018 | NaN | NaN | NaN | NaN |
| 33 | DEHTE | 2011 | 59.85 | 6.39 | NaN | -0.41 |
| 34 | DEHTE | 2012 | 36.83 | 3.46 | NaN | -0.21 |
| 35 | DEHTE | 2013 | 49.72 | 2.34 | NaN | -0.25 |
| 36 | DEHTE | 2014 | NaN | NaN | 13.26 | -0.19 |
| 37 | DEHTE | 2015 | 51.37 | 1.75 | 10.78 | -0.26 |
| 38 | DEHTE | 2016 | 50.77 | 2.09 | 9.8 | -0.25 |
| 39 | DEHTE | 2017 | 46.61 | 1.40 | 10.39 | -0.4 |
| 40 | DEHTE | 2018 | 41.62 | 2.52 | 6.12 | -0.22 |
| 41 | DESFN | 2012 | NaN | NaN | NaN | -0.08 |





| | | | | | | |
|---|---|---|---|---|---|---|
| 42 | DESFN | 2013 | 3.62 | 0.93 | 10.32 | -0.05 |
| 43 | DESFN | 2014 | NaN | NaN | 8.16 | NaN |
| 44 | DEZRK | 2013 | NaN | NaN | 13.03 | NaN |
| 45 | DEZRK | 2014 | NaN | NaN | 11.67 | NaN |
| 46 | DEZRK | 2015 | 30.76 | 1.00 | 10.85 | NaN |
| 47 | DEZRK | 2016 | 31.14 | 1.23 | 11.28 | 0.12 |
| 48 | DEZRK | 2017 | 29.10 | 0.87 | 10.84 | 0.31 |
| 49 | DEZRK | 2018 | 31.10 | 1.20 | 10.54 | 0.25 |
| 50 | FIHYY | 2016 | NaN | NaN | 5.41 | NaN |
| 51 | FILOM | 2006 | 13.77 | 0.76 | 4.47 | 0 |
| 52 | FILOM | 2007 | 17.22 | 0.25 | 4.33 | 0.04 |
| 53 | FILOM | 2008 | 15.52 | 0.22 | 3.79 | 0.06 |
| 54 | FILOM | 2009 | 17.63 | 0.27 | 3.98 | 0.02 |
| 55 | FILOM | 2010 | 13.78 | 0.29 | 3.71 | 0.03 |
| 56 | FISI2 | 2012 | 9.27 | 1.17 | 9.4 | 0.06 |
| 57 | FISI2 | 2013 | 10.22 | 1.17 | 10.47 | 0.13 |
| 58 | FISI2 | 2014 | NaN | NaN | 7.7 | 0.1 |
| 59 | FISI2 | 2015 | NaN | NaN | 8.18 | 0.09 |
| 60 | FISI2 | 2016 | NaN | NaN | 7.67 | 0.09 |
| 61 | FISII | 2013 | 14.58 | 0.32 | 6.45 | 0.04 |
| 62 | FISII | 2014 | 12.93 | 0.78 | 6.42 | 0.03 |
| 63 | FISII | 2015 | NaN | NaN | 6.92 | -0.02 |
| 64 | FISII | 2016 | 16.56 | 0.68 | 5.87 | -0.01 |
| 65 | FISII | 2017 | 8.63 | 0.23 | 8.4 | 0.06 |
| 66 | FISII | 2018 | 9.46 | 1.10 | 6.68 | 0.11 |
| 67 | FRLGT | 2017 | NaN | NaN | 10.45 | -0.24 |
| 68 | FRLGT | 2018 | 2.45 | 0.60 | 10.87 | -0.22 |
| 69 | HKMPM | 2016 | 11.62 | 0.61 | 25.06 | -0.61 |
| 70 | HKMPM | 2017 | 10.60 | 0.30 | 23.14 | -0.64 |
| 71 | HKMPM | 2018 | 11.04 | 0.59 | NaN | -0.8 |
| 72 | IDPAG | 2016 | 0.09 | 0.07 | NaN | NaN |
| 73 | IDPAG | 2017 | 0.09 | 0.09 | NaN | NaN |
| 74 | ITBCI | 2017 | NaN | NaN | 17.16 | NaN |
| 75 | ITBCI | 2018 | NaN | NaN | 17.36 | NaN |
| 76 | ITCAS | 2009 | 25.44 | 1.46 | 9.62 | NaN |
| 77 | ITCAS | 2010 | 17.80 | 1.26 | 12.37 | NaN |
| 78 | JPBBY | 2015 | 9.53 | 0.29 | 10.12 | 0 |
| 79 | JPBBY | 2016 | 16.42 | 0.45 | 10.02 | 0 |
| 80 | JPBBY | 2017 | 19.61 | 0.65 | 9.33 | -0.03 |
| 81 | JPBBY | 2018 | NaN | NaN | 9.79 | -0.04 |
| 82 | JPMSE | 2012 | 9.50 | 1.97 | 14.52 | 0.03 |
| 83 | JPSWL | 2016 | 66.68 | 4.29 | NaN | 1.91 |
| 84 | KRCRK | 2015 | NaN | NaN | 14.41 | 0.02 |
| 85 | KRCRK | 2016 | 29.12 | 0.91 | 12.48 | 0.03 |



| | | | | | | |
|---|---|---|---|---|---|---|
| 86 | KRCRK | 2017 | 25.84 | 0.86 | 13.94 | 0.02 |
| 87 | KRCRK | 2018 | 28.82 | 1.15 | 11.32 | 0.02 |
| 88 | MYMLM | 2014 | 9.55 | 0.59 | 26.8 | -0.09 |
| 89 | MYMLM | 2015 | NaN | NaN | 26.9 | -0.01 |
| 90 | NLHOR | 2007 | NaN | NaN | 12.4 | NaN |
| 91 | NLHOR | 2008 | NaN | NaN | 10.37 | NaN |
| 92 | NLHOR | 2009 | NaN | NaN | 11.61 | NaN |
| 93 | NZKOP | 2012 | 23.98 | 1.38 | 12.17 | -0.08 |
| 94 | NZKOP | 2013 | 15.33 | 0.43 | 12.68 | -0.13 |
| 95 | NZKOP | 2014 | 15.67 | 0.39 | 12.38 | -0.11 |
| 96 | NZKOP | 2015 | 14.37 | 2.66 | 12.46 | -0.1 |
| 97 | PHRIF | 2012 | NaN | NaN | 27.78 | NaN |
| 98 | PHRIF | 2013 | 12.41 | 0.99 | 28.17 | NaN |
| 99 | PHRIF | 2014 | NaN | NaN | 27.47 | NaN |
| 100 | RUCH2 | 2014 | 6.99 | 0.14 | -4.21 | NaN |
| 101 | RUCH2 | 2015 | 5.86 | 0.14 | -4.87 | NaN |
| 102 | RUCH2 | 2016 | NaN | NaN | -2.88 | NaN |
| 103 | RUCHE | 2014 | 3.84 | 0.14 | -3.31 | NaN |
| 104 | RUCHE | 2015 | 4.19 | 0.22 | -3.28 | NaN |
| 105 | RUCHE | 2016 | 4.24 | 0.19 | -1.65 | NaN |
| 106 | RUCOK | 2008 | NaN | NaN | NaN | NaN |
| 107 | RUCOK | 2009 | NaN | NaN | NaN | NaN |
| 108 | RUCOK | 2010 | NaN | NaN | NaN | NaN |
| 109 | RUCOK | 2011 | NaN | NaN | NaN | NaN |
| 110 | RUCOK | 2012 | NaN | NaN | -0.46 | NaN |
| 111 | RUCOK | 2013 | NaN | NaN | -5.73 | NaN |
| 112 | RUCOK | 2014 | NaN | NaN | -4.82 | NaN |
| 113 | RUCOK | 2015 | 4.45 | 0.15 | -4.4 | NaN |
| 114 | RUCOK | 2016 | NaN | NaN | -11.1 | NaN |
| 115 | RUFY2 | 2015 | NaN | NaN | 9.83 | NaN |
| 116 | RUFY2 | 2016 | 2.69 | 0.59 | 6.88 | 0.68 |
| 117 | RUFY2 | 2017 | 2.17 | 0.52 | 6.1 | 0.19 |
| 118 | RUFY2 | 2018 | 5.66 | 1.37 | 6.48 | 0.79 |
| 119 | SEDEG | 2014 | 11.24 | 1.98 | 5.02 | -0.02 |
| 120 | SEDEG | 2015 | 11.11 | 0.08 | 5.04 | 0.02 |
| 121 | SEDEG | 2016 | 11.19 | 0.15 | 5.19 | -0.01 |
| 122 | SEDEG | 2017 | NaN | NaN | 4.19 | 0 |
| 123 | SEDEG | 2018 | 9.42 | 0.09 | 5.49 | -0.03 |
| 124 | UKLBT | 2011 | NaN | NaN | NaN | NaN |
| 125 | UKLBT | 2012 | NaN | NaN | NaN | NaN |
| 126 | UKLBT | 2013 | 50.50 | 0.97 | NaN | NaN |
| 127 | UKLBT | 2014 | 42.57 | 2.25 | NaN | NaN |
| 128 | USA03 | 2015 | NaN | NaN | -6.65 | NaN |
| 129 | USA03 | 2016 | NaN | NaN | -6.14 | NaN |





| | | | | | | |
|---|---|---|---|---|---|---|
| **130** | USA03 | 2017 | 7.26 | 2.58 | -4.48 | NaN |
| **131** | USA03 | 2018 | 4.35 | 0.62 | -4.93 | NaN |
| **132** | USA10 | 2012 | NaN | NaN | NaN | NaN |
| **133** | USA10 | 2013 | NaN | NaN | NaN | NaN |
| **134** | USA10 | 2014 | NaN | NaN | NaN | NaN |
| **135** | USA10 | 2015 | NaN | NaN | NaN | NaN |
| **136** | USA10 | 2016 | NaN | NaN | NaN | NaN |
| **137** | USA10 | 2017 | NaN | NaN | NaN | NaN |
| **138** | USA10 | 2018 | NaN | NaN | NaN | NaN |
| **139** | USATQ | 2013 | NaN | NaN | -5.65 | NaN |
| **140** | USATQ | 2014 | 1.80 | 0.19 | -4.48 | NaN |
| **141** | USATQ | 2015 | 1.75 | 0.11 | -0.43 | NaN |
| **142** | USATQ | 2016 | 1.75 | 0.00 | NaN | NaN |
| **143** | USBEO | 2013 | NaN | NaN | -2.67 | NaN |
| **144** | USBEO | 2014 | 2.74 | 0.05 | -4.95 | NaN |
| **145** | USBES | 2013 | NaN | NaN | -6.01 | NaN |
| **146** | USBES | 2014 | 3.32 | 0.04 | -5.69 | NaN |
| **147** | USBES | 2015 | 3.06 | 0.54 | -6.24 | NaN |
| **148** | USBI1 | 2016 | NaN | NaN | 15.62 | NaN |
| **149** | USBI1 | 2017 | NaN | NaN | 17.17 | NaN |
| **150** | USBI1 | 2018 | 0.69 | 0.29 | 16.82 | NaN |
| **151** | USBI2 | 2017 | 0.86 | 0.20 | 20.42 | NaN |
| **152** | USBI2 | 2018 | 1.69 | 0.29 | 17.12 | NaN |
| **153** | USBZB | 2014 | 8.02 | 4.61 | 4.03 | NaN |
| **154** | USBZB | 2015 | 7.52 | 0.82 | 3.9 | NaN |
| **155** | USBZB | 2016 | 11.61 | 2.25 | 4.89 | NaN |
| **156** | USBZF | 2014 | 6.61 | 0.63 | 4.32 | NaN |
| **157** | USBZF | 2015 | 10.82 | 0.90 | 3.99 | NaN |
| **158** | USBZF | 2016 | NaN | NaN | 5.93 | NaN |
| **159** | USBZS | 2015 | 0.68 | 0.68 | 0.48 | NaN |
| **160** | USBZS | 2016 | 0.89 | 0.27 | 0.67 | NaN |
| **161** | USCRT | 2011 | 2.21 | 0.15 | 11.49 | -0.92 |
| **162** | USCRT | 2012 | 2.21 | 0.11 | 12.38 | -1.45 |
| **163** | USDPW | 2013 | NaN | NaN | NaN | NaN |
| **164** | USDPW | 2014 | 58.91 | 0.69 | NaN | NaN |
| **165** | USDPW | 2015 | NaN | NaN | NaN | NaN |
| **166** | USDPW | 2016 | 43.60 | 1.29 | NaN | NaN |
| **167** | USDPW | 2017 | 43.60 | 0.06 | NaN | NaN |
| **168** | USEDN | 2018 | -0.04 | 0.06 | NaN | NaN |
| **169** | USEML | 2015 | NaN | NaN | 5.71 | NaN |
| **170** | USEML | 2016 | 1.04 | 0.08 | 3.07 | NaN |
| **171** | USEML | 2017 | 0.36 | 0.27 | 3.8 | NaN |
| **172** | USEML | 2018 | 0.36 | 0.07 | NaN | NaN |
| **173** | USHO1 | 2012 | NaN | NaN | NaN | -0.43 |





| | | | | | | |
|---|---|---|---|---|---|---|
| 174 | USHO1 | 2013 | -0.05 | 0.02 | NaN | -0.33 |
| 175 | USHO1 | 2014 | -0.04 | 0.02 | NaN | -0.38 |
| 176 | USHO1 | 2015 | -0.16 | 0.01 | NaN | -0.48 |
| 177 | USHO1 | 2016 | -0.22 | 0.01 | NaN | -0.57 |
| 178 | USHO1 | 2017 | -0.24 | 0.01 | NaN | -0.56 |
| 179 | USHO1 | 2018 | -0.24 | 0.01 | NaN | NaN |
| 180 | USHRA | 2017 | -0.24 | 0.56 | NaN | NaN |
| 181 | USHRC | 2017 | -0.24 | 0.81 | NaN | NaN |
| 182 | USICS | 2014 | NaN | NaN | -1.55 | NaN |
| 183 | USICS | 2015 | NaN | NaN | -0.62 | NaN |
| 184 | USICS | 2016 | NaN | NaN | -1.48 | NaN |
| 185 | USIVO | 2013 | NaN | NaN | 3.19 | NaN |
| 186 | USIVO | 2014 | 5.05 | 0.22 | 0.02 | NaN |
| 187 | USIVO | 2015 | 3.89 | 0.27 | 0.47 | NaN |
| 188 | USIVO | 2016 | 5.77 | 0.55 | -1.01 | NaN |
| 189 | USLA1 | 2011 | NaN | NaN | 18.92 | NaN |
| 190 | USLA1 | 2012 | 12.68 | 0.63 | 24.23 | NaN |
| 191 | USLA2 | 2011 | 12.68 | 0.19 | NaN | NaN |
| 192 | USLA2 | 2012 | 48.42 | 1.57 | 23.09 | NaN |
| 193 | USLA2 | 2013 | 43.34 | 1.32 | 23.19 | NaN |
| 194 | USLOS | 2014 | 6.66 | 1.48 | 8.3 | -0.06 |
| 195 | USLOS | 2015 | 5.51 | 0.40 | 5.65 | -0.1 |
| 196 | USLOS | 2016 | 8.67 | 0.35 | 6.3 | -0.07 |
| 197 | USLOS | 2017 | 6.00 | 0.33 | 5.5 | -0.09 |
| 198 | USLOS | 2018 | 5.71 | 0.37 | 4.29 | -0.19 |
| 199 | USMAC | 2013 | 5.71 | 2.68 | NaN | NaN |
| 200 | USMAC | 2014 | 26.37 | 1.69 | 23.18 | -0.71 |
| 201 | USMAC | 2015 | 15.40 | 0.85 | 23.29 | -0.55 |
| 202 | USMRM | 2012 | 0.30 | 0.19 | 11.16 | NaN |
| 203 | USMRM | 2013 | 0.37 | 0.14 | 8.99 | NaN |
| 204 | USMYB | 2010 | NaN | NaN | NaN | 0.95 |
| 205 | USMYB | 2011 | 33.83 | 0.72 | 17.18 | 1.23 |
| 206 | USMYB | 2012 | 64.20 | 0.58 | 16.25 | 1.12 |
| 207 | USMYB | 2013 | 59.81 | 0.92 | 15.7 | 1.19 |
| 208 | USMYB | 2014 | 58.97 | 0.68 | 11.27 | 1.24 |
| 209 | USMYB | 2015 | 60.85 | 0.55 | NaN | 1.3 |
| 210 | USMYB | 2016 | 45.72 | 0.48 | NaN | 1.22 |
| 211 | USMYB | 2017 | 30.32 | 0.84 | 18.5 | 1.35 |
| 212 | USMYB | 2018 | 29.33 | 0.55 | 17.05 | 1.19 |
| 213 | USNC4 | 2012 | 38.28 | 1.70 | 17.12 | NaN |
| 214 | USNC4 | 2013 | 18.60 | 3.88 | NaN | NaN |
| 215 | USNC4 | 2014 | 26.98 | 0.60 | 18.02 | NaN |
| 216 | USNC4 | 2015 | 23.37 | 2.30 | 16.27 | NaN |
| 217 | USNC4 | 2016 | 62.20 | 2.78 | 16.35 | NaN |





| | | | | | |
|---|---|---|---|---|---|
| **218** | USNGB | 2012 | NaN | NaN | NaN | NaN |
| **219** | USNGB | 2013 | NaN | NaN | NaN | NaN |
| **220** | USNGB | 2014 | NaN | NaN | NaN | NaN |
| **221** | USNGB | 2015 | NaN | NaN | NaN | NaN |
| **222** | USNGB | 2016 | NaN | NaN | NaN | NaN |
| **223** | USNGB | 2017 | 2.31 | 0.11 | NaN | NaN |
| **224** | USNGB | 2018 | 2.52 | 0.22 | NaN | NaN |
| **225** | USNGC | 2017 | 2.52 | 0.06 | NaN | NaN |
| **226** | USNGC | 2018 | 2.52 | 0.05 | NaN | NaN |
| **227** | USORV | 2011 | 3.53 | 0.54 | 16.64 | NaN |
| **228** | USORV | 2012 | 9.11 | 0.45 | 14.23 | NaN |
| **229** | USORV | 2013 | 7.70 | 0.41 | 13.19 | NaN |
| **230** | USORV | 2014 | 8.46 | 0.26 | 12 | NaN |
| **231** | USORV | 2015 | NaN | NaN | 13.36 | NaN |
| **232** | USOWC | 2015 | NaN | NaN | 22.11 | 0.9 |
| **233** | USOWC | 2016 | 113.99 | 3.25 | 21.19 | 0.54 |
| **234** | USPFA | 2010 | NaN | NaN | NaN | NaN |
| **235** | USPFA | 2011 | 0.34 | 0.05 | NaN | NaN |
| **236** | USPFA | 2012 | 0.30 | 0.04 | NaN | NaN |
| **237** | USPFA | 2013 | 0.31 | 0.05 | NaN | NaN |
| **238** | USPFA | 2014 | NaN | NaN | NaN | NaN |
| **239** | USPFA | 2015 | 0.63 | 0.03 | NaN | NaN |
| **240** | USPFA | 2016 | 0.85 | 0.02 | NaN | NaN |
| **241** | USPFA | 2017 | 0.80 | 0.06 | NaN | NaN |
| **242** | USPFA | 2018 | NaN | NaN | NaN | NaN |
| **243** | USSND | 2010 | NaN | NaN | 16.85 | NaN |
| **244** | USSND | 2011 | NaN | NaN | 14.96 | NaN |
| **245** | USSND | 2012 | 6.34 | 0.25 | 16.06 | NaN |
| **246** | USSND | 2013 | 6.04 | 0.48 | 16.59 | -0.65 |
| **247** | USSND | 2014 | 3.23 | 0.36 | 17.52 | -0.78 |
| **248** | USSND | 2015 | 3.23 | 0.21 | NaN | NaN |
| **249** | USSNE | 2016 | NaN | NaN | 17.85 | -0.2 |
| **250** | USSNE | 2017 | 45.96 | 0.40 | 17.05 | 0.16 |
| **251** | USSNE | 2018 | 39.63 | 0.66 | 16.83 | 0.09 |
| **252** | USSRR | 2014 | 0.71 | 0.10 | NaN | NaN |
| **253** | USSRR | 2015 | 0.88 | 0.11 | NaN | NaN |
| **254** | USSRR | 2016 | 0.86 | 0.10 | 16.3 | -0.18 |
| **255** | USSRR | 2017 | 0.86 | 0.11 | NaN | NaN |
| **256** | USSTJ | 2016 | 9.55 | 1.04 | 11.66 | -0.26 |
| **257** | USTW1 | 2011 | 26.09 | 2.70 | 14.01 | NaN |
| **258** | USTW1 | 2012 | NaN | NaN | 11.58 | 0.24 |
| **259** | USTW1 | 2013 | 33.93 | 1.78 | 11.92 | 0.25 |
| **260** | USTW1 | 2014 | 49.60 | 1.67 | 13.14 | 0.25 |
| **261** | USTW1 | 2015 | 54.80 | 2.58 | 12.79 | 0.33 |



| | Site | Year | Ann_Flux_g_C_m-2 | Ann_Flux_Uncertainty_g_C_m-2 | Mean_Soil_Temp_C | Mean_Water_Table_Depth_m |
|---|---|---|---|---|---|---|
| 262 | USTW1 | 2016 | 45.93 | 1.90 | 12.91 | 0.41 |
| 263 | USTW1 | 2017 | 38.66 | 2.09 | 12.53 | 0.38 |
| 264 | USTW1 | 2018 | 27.60 | 1.64 | 12.1 | 0.24 |
| 265 | USTW3 | 2013 | NaN | NaN | 19.63 | NaN |
| 266 | USTW3 | 2014 | NaN | NaN | 17.91 | NaN |
| 267 | USTW4 | 2013 | NaN | NaN | NaN | NaN |
| 268 | USTW4 | 2014 | 16.26 | 0.39 | NaN | 0.48 |
| 269 | USTW4 | 2015 | 27.61 | 0.43 | 17.2 | 0.36 |
| 270 | USTW4 | 2016 | 33.49 | 0.37 | 14.8 | 0.18 |
| 271 | USTW4 | 2017 | 47.95 | 0.58 | 13.78 | 0.07 |
| 272 | USTW4 | 2018 | 37.41 | 0.48 | 13.02 | 0.08 |
| 273 | USTW5 | 2018 | 59.72 | 1.15 | 16.67 | 0.69 |
| 274 | USTWT | 2009 | NaN | NaN | 17.66 | -0.01 |
| 275 | USTWT | 2010 | 9.87 | 1.15 | 15.67 | -0.18 |
| 276 | USTWT | 2011 | 12.32 | 4.92 | 14.95 | -0.11 |
| 277 | USTWT | 2012 | 8.12 | 0.51 | 16.05 | -0.04 |
| 278 | USTWT | 2013 | 12.64 | 0.48 | 15.98 | -0.11 |
| 279 | USTWT | 2014 | 17.02 | 0.97 | 17.44 | -0.09 |
| 280 | USTWT | 2015 | 14.43 | 0.38 | 17.04 | -0.14 |
| 281 | USTWT | 2016 | 11.07 | 0.59 | 16.44 | -0.29 |
| 282 | USTWT | 2017 | 11.07 | 0.31 | NaN | NaN |
| 283 | USUAF | 2011 | 0.32 | 0.04 | -2.14 | -0.17 |
| 284 | USUAF | 2012 | NaN | NaN | -2.43 | -0.18 |
| 285 | USUAF | 2013 | NaN | NaN | -1.15 | -0.18 |
| 286 | USUAF | 2014 | NaN | NaN | -1.18 | -0.13 |
| 287 | USUAF | 2015 | NaN | NaN | -0.49 | -0.12 |
| 288 | USUAF | 2016 | 0.68 | 0.05 | -0.05 | -0.1 |
| 289 | USUAF | 2017 | 0.58 | 0.06 | 1.09 | -0.13 |
| 290 | USUAF | 2018 | NaN | NaN | 0.87 | -0.13 |
| 291 | USWPT | 2011 | 41.05 | 1.57 | 17.22 | 0.43 |
| 292 | USWPT | 2012 | 54.96 | 1.71 | 14.27 | 0.28 |
| 293 | USWPT | 2013 | 52.76 | 1.29 | 12.89 | 0.44 |

**Column Descriptions**

| | |
|---|---|
| Site | Site name |
| Year | Data year |
| Ann_Flux_g_C_m-2 | Total annual methane flux (gC/m^2) |
| Ann_Flux_Uncertainty_g_C_m-2 | Gap-filling and rancom uncertainty associated with annual flux (gC/m^2) |
| Mean_Soil_Temp_C | Annual mean soil temperature (degree C). For sites with multiple probes, we use the probe closest to the surface |
| Mean_Water_Table_Depth_m | Annual mean water table depth (m) |