# Peer review of "FLUXNET-CH4: A global, multi-ecosystem dataset and analysis 1 of methane seasonality from freshwater wetlands 2"

_Earth System Science Data, 2020_

## Author Response (AR1)

Response to comments on manuscript ESSD-2020-307

**RC1: ['Comment on essd-2020-307'](), Anonymous Referee #1, 18 Feb 2021**

**The FLUXNET-CH4 dataset includes eddy covariance methane fluxes along with CO2 fluxes and associated meteorological and site factors from 79 globally-distributed field sites. The manuscript is a well written and detailed description of the dataset and also includes preliminary analyses of seasonal patterns, site representativeness, and important summary information about the dataset. The methods used to develop and process the dataset are considered reliable and well tested within the eddy covariance field and are clearly described. The data appears to be high quality. This dataset will be highly useful for future modeling, synthesis, and process-oriented studies focused on ecosystem methane fluxes and how they vary over space and time. The dataset is freely accessible, although users must register on the FLUXNET web site. The data access interface is somewhat inefficient when downloading data for a larger number of sites (it requires clicking a link for each site individually, or installing a third-party downloading plugin). The option to download all sites as a single compressed file might facilitate easier access for projects that use many sites such as cross-site syntheses.**

**The manuscript includes quite a bit of interesting analysis of seasonal patterns across sites and how they vary with latitude and mean annual temperature. While these analyses are certainly interesting and valuable, I'm not sure if they fit entirely within the aims and scope of ESSD, which states "Any interpretation of data is outside the scope of regular articles." ([https://www.earth-system-science-data.net/about/aims_and_scope.html]()).**

Response to general comments:

Thank you for your positive review of our manuscript, we are happy that you think it will be useful for the research community.  We believe that ESSD is a good home for this work.  Regarding interpretation of the data, we recognize that our detailed description of patterns did involve some basic analyses and interpretations, and we appreciate your comment that they were 'interesting and valuable'. Our objectives were to not only provide the FLUXNET-CH4 data, but also include detailed descriptions of the patterns in the data so that a data user may understand the overall representativeness of the various wetland sites, and the seasonality patterns of methane flux from wetlands.  This necessarily involved some coarse analyses and summarizing of the raw data. Even so, there is considerable 'room' for additional, more complex analyses, some of which we recommend in the text. We hope that by describing site representativeness and patterns within the data, we are laying the foundation for future studies to focus on interpretation and mechanistic relationships.

Regarding the difficulties in data acquisition, we acknowledge that having to individually download site data can be cumbersome, and that using a third-party download plugin is not ideal.  However, there are practical reasons why the dataset does not currently include a "Download All" feature.  FLUXNET-CH4 was built on the existing Ameriflux framework,

which does not support a "Download All" feature because data files change often as users upload new data, and file sizes would be too large (which would be particularly problematic for users with slower or inconsistent connections) . While FLUXNET-CH4 is more static than the entire suite of Ameriflux sites, and a "Download All" feature may be a possibility in future FLUXNET-CH4 releases, the database does not currently support this. For now, third-party plugins such as DownThemAll can be used to download all data at once.

**Overall, I think this is a very valuable dataset and a high quality description paper. I have some specific comments about the manuscript that could help improve the clarity of some aspects:**

**Line 304-305: From the download interface, it appears that some sites are available as Fluxnet Tier 2, not as CC BY 4.0. So the statement that all site data are available under CC BY 4.0 is not completely true.**

The reviewer makes a valid point and we failed to clarify this in the manuscript. Our analysis uses data from 79 sites, all of which are available under the CC BY 4.0 data license. However, the FLUXNET-CH4 database includes an additional two sites (RU-Vrk and SE-St1) that are available under the more restrictive Tier 2 data license. To clarify this point, we have added the following text:

At updated Line 224: "FLUXNET-CH4 includes an additional 2 wetland sites (RU-Vrk and SE-St1), but they are not available under the CC BY 4.0 data policy and thus are excluded from this analysis."

At updated Line 302: "FLUXNET-CH4 has an additional 2 sites available under the FLUXNET Tier 2 license ([https://fluxnet.org/data/data-policy/](https://fluxnet.org/data/data-policy/)), though these sites are not included in our analysis."

At updated Line 794: "(2 additional sites in FLUXNET-CH4 are available under the more restrictive Tier 2 data policy, https://fluxnet.org/data/data-policy/; these sites are not used in our analysis)."

**Line 312-313: Is there a more precise definition for "relatively shallow water table"? Was a specific cutoff depth used?**

We agree this line is currently unclear. Many of these drained wetland sites likely have shallow water tables, but we did not actually use water table depth as a criterion for classification. Still, we think it is important to note that drained sites may have relatively shallow water tables that can contribute to methane emissions, and thus these sites do not necessarily behave like other "dry" sites. Therefore, we have updated the text to read:

At updated line 311: "Drained systems are former wetlands that have subsequently been drained but may maintain a relatively shallow water table, which can contribute to occasional methane emissions, although we do not have specific water table depth information at all drained sites."

**Line 316: Is there a more precise taxonomic or ecological description for "brown mosses"? This seems like a vague term and is not described in the cited Treat et al (2018) paper.**

Thank you for noting this fair point. We double checked Treat et al.'s datasets metadata information and the reviewer is correct, details are not provided for Treat et al.'s classification information. We have now included details of our classification criteria in our text to make it more transparent and repeatable. We needed a category for mosses that are not *Sphagnum* so that we could differentiate *Sphagnum* presence (primarily because *Sphagnum* often has characteristic acidic microenvironments relevant for $CH_4$ production and consumption). Thus, we labeled any moss from the division Bryophyta that is not in the class Sphagnopsida as brown moss. Revised text is as follows:

At updated Line 316: "For all sites, vegetation was classified for presence or absence of brown mosses (all species from the division Bryophyta except those in the class Sphagnopsida), Sphagnum mosses (any species from class Sphagnopsida), …"

**Line 323: To be precise, Table B3 includes citations to the climatic data, not the data itself.**

Thank you for pointing this out. We decided to include the climatic data in this table (and please note that the table number has changed to B4).

**Line 327: Table B7 includes annual methane flux and uncertainty, specifically. Referring to "flux" is ambiguous because the dataset includes methane, CO2, and energy fluxes.**

We have updated this sentence to say "$CH_4$ flux" instead of just "flux".

**Line 330: Section 2.1.4 should specify that it refers to annual CH4 fluxes to avoid confusion since the dataset also includes CO2 and energy fluxes.**

This is a good point. We have updated the Section 2.1.4 title to "Annual $CH_4$ fluxes". We also went through the manuscript and converted all references to "annual flux" to "annual $CH_4$ flux".

**Line 340-341: Some more explanation would be helpful. It wasn't immediately clear to me what this meant. Specifically, the site has one year of data that goes across two calendar years, so both years were listed separately but with the same annual flux value in the table.**

We agree this is currently unclear. We added an asterisk to ID-Pag and a footnote to Table 7, stating "*Data from ID-Pag spans 365 days from June 2016 to June 2017. Annual methane flux for each year is the sum of these 365 days, with uncertainty being calculated separately for each year."

**Line 354: It's not clear which global gridded datasets are being referred to here. Datasets of what? Salinity? Wetland area? Or something else?**

Agreed, we modified the text to specify that gridded salinity data is what is limiting our assessments of coastal wetlands. The passage now reads as:

At updated Line 353: "Coastal sites were excluded because salinity, an important control on CH4 production, could not be evaluated across the tower network due to a lack of global gridded salinity data (Bartlett et al., 1987; Poffenbarger et al., 2011).".

**Line 422: It's not clear what the "range" is referring to. Does this mean annual averages? Is the range referring to the different variables that were used, or to different values?**

Thank you for pointing out this unclear sentence. The word 'range' was not actually needed in the original ms, and we have now updated the text with more details on the regression:

At updated line 421: "We regressed the $CH_4$ seasonality parameters from Timesat against annual temperature, annual water table depth, and Timesat seasonality parameters for air temperature, soil temperature, and GPP (proxy for recent carbon input available as substrate) using linear mixed-effect modeling with the *lmer* command (with site as a random effect) from the R (R Core Team 2018, version 3.6.2) package lmerTest. "

**Figure 3: Many of the dots overlap. It would be easier to distinguish sites if the dots were smaller or transparent.**

We agree that the original figure made it difficult to distinguish many of the dots. Given how closely some sites cluster, reducing dot size or making dots transparent did not make it easier to distinguish sites. Instead, we re-did the figure to include four insets showing zoomed-in areas where sites cluster together. We also added site labels to all site locations.

**Line 559: "a site in Botswana": The site code should be provided here**

Agreed - we added the site code in question: BW-Npw.

**Line 565: "The size of wetland points are made larger": All the points are the same size so it's not clear what this means.**

Thank you for noticing this disconnect between this version of the figure and caption. Point size does not vary and is no longer described in the Fig. 6 caption.

**Line 566: Not all points are labeled with site codes. Was this just for ease of visualization? Or did some other factor go into the choice of which to label?**

Correct - for visual clarity, we only labelled selected sites that were distributed sufficiently sparsely.  The Fig.6 caption now describes this as follows: "Sites codes are labeled in blue text for selected sites deviating from average conditions."

**Does density of land pixels (gray colors) have meaningful units that can be provided for this figure? Or is it purely qualitative? If it is quantitative, a color bar should be provided for the gray shading. Is the amount of area covered by gray shaded regions quantitatively meaningful?**

We agree that the gray polygons representing land pixel density could be described further in the caption. The density of land pixels is certainly a quantitative measurement, but we intend for a qualitative interpretation of wetland hotspots as its primary use.  Because the plot represents a PCA, the density units of these polygons would be: "*number of wetland pixels per unitless 1x1 PCA unit*" which would not contribute much to the interpretation of the figure.  The density breaks determining the area occupied by each density polygon  were chosen to visually identify the major wetland hotspots (and the EC tower sites within them).

To add clarity on this issue, Fig. 6 caption was updated to the following: "The background shades of gray are a qualitative representation of the density of global wetland pixels and their distribution in the PCA climate-space, with darker color representing higher densities (excluding Greenland and Antarctica). Only grid cells with wetland that have a >5% average wetland fraction according to the WAD2M over 2000-2018 are included (Zhang et al., In Review)."

**Line 599-600: The suggestion of regions that could improve data coverage is useful. Can a citation be provided to support the statement that these regions are high CH4 emitting? Since they are not included in this dataset, there must be some outside data or publications estimating fluxes from those regions that this statement is referring to.**

We have added a reference to the Saunois et al., 2020 publication here (citation below), which presents global maps of methane emissions from wetlands.

Saunois, M., Stavert, A. R., Poulter, B., Bousquet, P., Canadell, J. G., Jackson, R. B., Raymond, P. A., Dlugokencky, E. J., Houweling, S., Patra, P. K., Ciais, P., Arora, V. K., Bastviken, D., Bergamaschi, P., Blake, D. R., Brailsford, G., Bruhwiler, L., Carlson,

K. M., Carrol, M., & Others. The Global Methane Budget 2000–2017. Earth Syst. Sci. Data, 12, 1561-1623. https://doi.org/10.5194/essd-12-1561-2020. 2020.

**Figure 8: A legend should be added to the figure labeling the different line colors. Also, it is best to avoid using red and green colors as the only distinguishing factor in graphics because red/green colorblindness is quite common and would make this figure difficult to interpret. Use of red/green colors is an issue on several of the figures (9, 10, 11, 12). This could be addressed by using a colorblind-friendly color scheme, or by using different symbols or line styles in addition to different colors.**

Thank you for pointing out the issue with our color scheme. We have updated the colors on Figures 8, 9, 10, 11, and 12 and added different marker shapes and line styles. We modified the legend to Figure 8 and updated the caption accordingly.

**Line 652: What does the yellow line show?**

We have modified the line colors and legend in Figure 8; there is no longer a yellow.

**Line 665-666: This phrasing is confusing. What are these months being added to? I guess this refers to integrating over the time period from September-May instead of October-March. But isn't it obvious that including more months would give higher total fluxes? This would always be true unless fluxes were zero or negative in some months.**

We agree that this phrasing is confusing. We have decided to eliminate this sentence since, as you point out, including more months will obviously lead to higher flux, and the following sentence (at updated lines 671-674) also references the Zona et al., 2016 work.

**Line 706: Does the confidence interval 31 +/- 40 days mean that the lag was not significantly different from zero?**

Even though the CI is high, statistically it was still significantly different from zero. To clarify, we have edited the sentence to read:

At updated Line 710: "Although the spring onset of increasing $CH_4$ emissions correlates with mean annual air temperature, on average it lags the spring increase in the shallowest soil temperatures by $31 \pm 40$ days (Fig. 11, **lag is significantly different than zero with p< 0.001)**..."

**RC2:** **'Reply on RC1'**, Anonymous Referee #2, 27 Feb 2021
**The FLUXNET-CH4 database, presented by the manuscript, is a valuable asset to the global flux and modeling community. The dataset is openly accessible through the FLUXNET website.**

**I found the manuscript to be well written, though another round of editing may be helpful to make some sections more succinct (I noticed repetitive wording in a few places).**

**The methodologies are described clearly and are well tested. I appreciate that the authors offer gap-filled, in addition to non-gap-filled, data.**

**I found the data representative analyses (i.e., the dissimilarity and tower constituency maps) to be very useful in highlighting gaps in the placement of eddy covariance towers across bioclimatic space. I also appreciate the discussion of potential biases and uncertainty in the data.**

We are happy to hear that you found this manuscript useful.  Thank you for your suggestion to remove repetitive wording.  We have re-read the manuscript and deleted unnecessary words when possible.
* * *
**CC1:** **'Comment on essd-2020-307'**, Andreas Heinemeyer, 28 Jan 2021

**This is a fantastic dataset!**

**I have a comment on the Figure 5. Whilst it seems OK to just plot a linear trendline through these fluxes (on a logarithmic scale), this is questionable when looking at the individual biomes, which seem to indicate an even stronger (possibly even some exponential) increase with temperature. I would like to see individual regression lines fitted to the individual biomes (fen, bog, ...). I know there are few points per biome, but it seems important to see these biomes in their individual light - as underpinning conditions and processes are likely very different. This is of particularly importance when considering previous work by Abdalla et al., 2016 (temperature & moisture impacts on methane fluxes), highlighting those differences - which I think might look similar when plotting these data for biomes individually.**

Thank you for your comment, we agree that it would be interesting to see if/how the temperature relationship differs between biome types.  We tried regressing the data for individual biomes against temperature.  However, only fen annual $CH_4$ flux had a significant correlation with mean

air temperature (on a log scale), and the slope of the regression line was very similar to the overall regression line.  We believe the lack of significant relationship for individual biomes is due, at least in part, to having relatively few data points per biome, and this could be an interesting area to explore when more flux data becomes available.

To acknowledge the potential differences in biome response to mean annual temperature, we have added the following sentences:

At updated line 492:  " We also note that annual $CH_4$ flux from individual biomes may have different relationships with temperature, as previous work has shown biome-specific trends in $CH_4$ flux with environmental drivers (Abdalla et al., 2016).  However, there currently are not enough data points in each biome category to compare relationships between mean annual $CH_4$ flux and temperature."

---

## Author Response (AR2)

Comments to the Author:
Well-organized valuable data product; thank for using ESSD.

Thank you for considering our manuscript for ESSD.

Two small inconsistencies:

1) Line 382: "Localized fits are minimized" I think you mean 'optimized'? Or offsets or discrepancies are minimized.

Thank you for catching this. We have updated this sentence to read "Localized fits are determined by minimizing a merit function with the Levenberg-Marquardt method".

2) Residual uncertainty - for this reader at least - about salt-water-influenced sites: Line 470 and following, including Table 1 and section 3.1.4: data set excludes coastal sites for valid reasons but includes salt marshes?k Seawater influence in both?

We agree that we could be more clear about the transition from discussing all FLUXNET-CH4 sites, regardless of type, to only considering the freshwater wetlands. To address this, we have changed the Section 3.2 heading from "Wetland Representativeness" to "Freshwater wetland representativeness". We have also added the following sentence in Section 3.2: "We exclude wetlands classified as "Salt Marsh" in this representativeness analysis and the seasonality analysis below because of the unique $CH_4$ --flux dynamics in saltwater ecosystems (as discussed in section 3.1.4), though we note that some of the coastal wetlands included in the freshwater analysis periodically experience brackish water (i.e.: US-Myb, US-Sne).".